**Investigation**

# Performance of *qpAdm*-based screens for genetic admixture on graph–shaped histories and stepping stone landscapes

Olga Flegontova,[1,2,†] Ulaş Işıldak [ID],[1,3,†] Eren Yüncü [ID],[1,4,†] Matthew P. Williams,[5] Christian D. Huber [ID],[5] Jan Kočí,[1] Leonid A. Vyazov [ID],[1] Piya Changmai,[1] Pavel Flegontov [ID] [1,6,*]

[1]Department of Biology and Ecology, Faculty of Science, University of Ostrava, Ostrava 710 00, Czechia
[2]Institute of Parasitology, Biology Centre of the Czech Academy of Sciences, České Budějovice 370 05, Czechia
[3]Leibniz Institute on Aging, Fritz Lipmann Institute, Jena 07745, Germany
[4]Department of Biological Sciences, Middle East Technical University, Üniversiteler Mahallesi, Ankara 06800, Türkiye
[5]Department of Biology, Eberly College of Science, The Pennsylvania State University, University Park, PA 16802, USA
[6]Department of Human Evolutionary Biology, Harvard University, Cambridge, MA 02138, USA

*Corresponding author: Department of Biology and Ecology, Faculty of Science, University of Ostrava, Chittussiho 1077/10, Ostrava 710 00, Czechia.
Email: pavel.flegontov@osu.cz, pavel_flegontov@fas.harvard.edu
[†]These authors contributed equally to the study.

*qpAdm* is a statistical tool that is often used for testing large sets of alternative admixture models for a target population. Despite its popularity, *qpAdm* remains untested on 2D stepping stone landscapes and in situations with low prestudy odds (low ratio of true to false models). We tested high-throughput *qpAdm* protocols with typical properties such as number of source combinations per target, model complexity, model feasibility criteria, etc. Those protocols were applied to admixture graph–shaped and stepping stone simulated histories sampled randomly or systematically. We demonstrate that false discovery rates of high-throughput *qpAdm* protocols exceed 50% for many parameter combinations since: (1) prestudy odds are low and fall rapidly with increasing model complexity; (2) complex migration networks violate the assumptions of the method; hence, there is poor correlation between *qpAdm* *P*-values and model optimality, contributing to low but nonzero false-positive rate and low power; and (3) although admixture fraction estimates between 0 and 1 are largely restricted to symmetric configurations of sources around a target, a small fraction of asymmetric highly nonoptimal models have estimates in the same interval, contributing to the false-positive rate. We also reinterpret large sets of *qpAdm* models from 2 studies in terms of source–target distance and symmetry and suggest improvements to *qpAdm* protocols: (1) temporal stratification of targets and proxy sources in the case of admixture graph–shaped histories, (2) focused exploration of few models for increasing prestudy odds; and (3) dense landscape sampling for increasing power and stringent conditions on estimated admixture fractions for decreasing the false-positive rate.

**Keywords:** archaeogenetics; genetic admixture; *qpAdm*; simulation; admixture graphs; stepping stone models

## Introduction

Although a broad range of methods exists for reconstructing population history from genome-wide autosomal single nucleotide polymorphism (SNP) data, just a few methods became the cornerstone of archaeogenetic studies: principal component analysis (PCA) (Patterson *et al.* 2006); an unsupervised or supervised algorithm for admixture inference in individuals, *ADMIXTURE* (Alexander *et al.* 2009); formal tests for admixture such as $f_3$-statistics (Patterson *et al.* 2012; Peter 2016, 2022; Soraggi and Wiuf 2019) and *D*-statistics (Green *et al.* 2010; Durand *et al.* 2011); and a tool for fitting 2-component and more complex admixture models to populations, *qpAdm* (Haak *et al.* 2015; Harney *et al.* 2021). The popularity of these methods is explained by their relatively modest computational requirements and versatility since they can analyze unphased biallelic genotype data of various types (pseudohaploid or diploid), generated using either targeted

enrichment on a panel of sites or shotgun sequencing technologies and low-coverage ancient genomes with high proportions of missing data (Harney *et al.* 2021). However, only a few studies have been devoted to testing the performance of these diverse methods on simulated genetic data (Alexander *et al.* 2009; McVean 2009; Martin *et al.* 2015; Lazaridis *et al.* 2017; Moreno-Mayar *et al.* 2018; Soraggi and Wiuf 2019; Ning, Fernandes, *et al.* 2020; Harney *et al.* 2021; Speidel *et al.* 2025), and realistically complex population histories remain virtually unexplored in this respect: both those approximated best by admixture graphs and isolation by distance (stepping stone models; Novembre and Stephens 2008; Duforet-Frebourg and Slatkin 2016).

PCA is often used by archaeogenetic studies as a 1st line of analysis, providing an overview of population structure and helping to propose hypotheses about migration and admixture.

Distribution of individual genomes in spaces of principal components (PCs) does not have unambiguous interpretation since even under ideal conditions (in the absence of missing data, batch artifacts, and selection signals), it is affected by both genetic drift and admixture (McVean 2009). Formal tests for genetic admixture such as $f_3$-statistics and $D/f_4$-statistics are often used to test for the possibility that a certain cline spotted in PC space might be a result of migration and admixture of previously isolated ancestries and does not reflect isolation by distance or recurrent bottlenecks (Novembre and Stephens 2008; Frichot *et al.* 2012; Duforet-Frebourg and Slatkin 2016; Estavoyer and François 2022). $D$- and $f_4$-statistics, which are identical except for the denominator and are not affected by genetic drift, test if an unrooted 4-population tree fits the data (Reich *et al.* 2009; Green *et al.* 2010; Durand *et al.* 2011; Patterson *et al.* 2012). A statistically significant deviation of the statistic from 0 (estimated using jackknife or bootstrap resampling) means that either the assumed tree topology is wrong or gene flow occurred between a pair of branches in the tree, assuming that at most one mutation per locus occurred in the history of the 4 populations (Durand *et al.* 2011; Patterson *et al.* 2012) and SNP ascertainment bias (Flegontov *et al.* 2023) is absent. However, interpretation of these statistics is ambiguous since gene flow directionality remains unknown, and 2 pairs of branches can be responsible for a deviation of the statistic from 0 (Lipson 2020). Since gene flow may be mediated by ghost groups related only distantly to the sampled groups at the branch tips (Tricou *et al.* 2022), excluding 1 pair of branches due to geographical and temporal plausibility of gene flow is also difficult. And, interpretating deviations of $D$- and $f_4$-statistics from 0 (or lack thereof) becomes almost impossible if both branch pairs are connected by detectable gene flows, a typical situation on isolation-by-distance landscapes. In general, unambiguous interpretation of nonrejection of the null hypothesis (no deviation from 0) should be avoided.

"Admixture" $f_3$-statistics of the form $f_3$(target; proxy source$_1$, proxy source$_2$) constitute another formal test for admixture (Patterson *et al.* 2012). Significant deviation of such a statistic from 0 in the negative direction ($Z$-score below $-3$) is considered proof of admixture since allele frequencies at most sites are intermediate in the target group between those in the proxy sources (Patterson *et al.* 2012). However, "admixture" $f_3$-statistics are usually only applicable for detection of recent admixture events since they become positive when postadmixture genetic drift on the target lineage moves allele frequencies away from these intermediate values (Patterson *et al.* 2012; Peter 2016, 2022). When the sources are more distant genetically, $f_3$-statistics remain negative for longer periods of time.

Considering these complications, more sophisticated tests for genetic admixture are needed. The *qpAdm* method introduced by Haak *et al.* (2015) was developed with the admixture graph paradigm in mind. It is based on matrices of $f_4$-statistics and does not require detailed knowledge of population phylogeny beyond a few assumptions (Lazaridis *et al.* 2016; Harney *et al.* 2021). Recently, *qpAdm* was updated to accommodate more accurate $f_4$-statistics derived from genome-wide genealogical trees for haplotypes rather than from allele frequencies (Speidel *et al.* 2025). The *qpAdm* method tests admixture models in the form of combinations of proxy ancestral groups ("sources" or "references"; Lazaridis *et al.* 2016) for a "target" (or "test") population, given a genetically diverse set of "outgroup" populations and infers ancestry proportions in the target group contributed by the lineages represented by the proxy sources ("outgroups" are often termed "right" populations for convenience since they are usually not outgroups in the phylogenetic sense, and they were termed "references" by Harney *et al.*

---

**Box 1. Terminology used in this study for describing simulations, admixture models, results of screens for admixture, and *qpAdm* protocols.**

| | |
|---|---|
| AGS history | A history represented as a (usually bifurcating) directed acyclic graph, composed of episodes of population divergence and of mass migration and admixture of 2 source populations (pulse admixture events) |
| SSL | An approximation for 2D isolation-by-distance or isolation-by-resistance landscapes (genetic landscapes); here, we describe an implementation used in this study. An SSL is originally "unfolded" from a single founder deme via a serial founder process that can be represented by a bifurcating tree (Estavoyer and François 2022) or via a multifurcation, i.e. star radiation (in this study). This stage is followed by a gene flow era represented as an undirected graph connecting demes, where each edge represents a bidirectional gene flow, and gene flow intensities are allowed to differ in the forward and reverse directions. Gene flows on this undirected graph happen in one or more epochs, with unidirectional gene flow intensities sampled randomly from uniform or nonuniform distributions. Thus, gene flow intensities are nonuniform, and our SSL approximates isolation-by-resistance landscapes (McRae 2006). In this study, all SSLs are based on finite hexagonal lattices where demes are located at vertices of triangles tiling the plane (not to be confused with honeycomb tiling), and node degree varies from 3 to 6. We note that rapid long-distance migrations (gene flow edges connecting nonneighboring demes directly) are absent in such simulations |
| Gene flow epoch | A period of unchanging gene flow intensities on the landscape (they are assigned randomly at the beginning of the epoch) |
| Gene flow era | A collection of gene flow epochs characterized by the same topology of the undirected gene flow graph, but not necessarily by stable gene flow intensities |
| "Right" (or outgroup) populations/groups | In a *qpAdm* protocol, these are reference populations needed for testing admixture models composed of a target and 1 or several proxy source groups |
| Target (or test) population/group | In an admixture model, this is a population/group whose genetic history is being modeled |
| true (ancestry) source | In the context of simulated AGS histories, this is a population directly participating in an admixture event(s) giving rise to a target group and its ancestral (unsampled) population before merging with other populations deeper in time. In contrast to AGS histories, the concept of true and false ancestry sources is vague in the case of SSLs where all neighboring demes continuously exchange migrants; we can just rank sources into more and less appropriate |

**Box 1. (Continued)**

| | |
|---|---|
| Proxy (ancestry) source | In the context of simulated admixture graphs, this is a sampled population included in an admixture model as a potential source and assumed to be cladal with one of the true ancestry sources. In the context of SSL, this is any deme included in an admixture model as a potential source. |
| Model complexity | Number of proxy ancestry sources in an admixture model "target = proxy source$_1$ + proxy source$_i$ + … proxy source$_i$" |
| "Left" populations/groups | In a *qpAdm* model, these are proxy sources and a target |
| reference populations/ groups | "Right" and proxy source populations combined |
| *qpAdm* experiment | A set of all possible (and progressively more complex) models for a given target and a set of reference groups. An experiment is stopped as soon as at least one feasible model for the target is encountered |
| "Trailing" models | For an *n*-way admixture model, these are all possible simpler models composed of the target and/or proxy sources from that *n*-way model (but not from other *n*-way models in the experiment): "target = proxy source$_1$ + proxy source$_i$ + …" and "proxy source$_1$ = proxy source$_i$ + …" |
| (Composite) model feasibility criterion | A set of criteria for identifying fitting (feasible) *qpAdm* models that rely on both estimated admixture proportions (or EAFs) and *P*-values. In general terms, a composite criterion is composed of up to 3 conditions: (1) on admixture proportion estimates in an *n*-way model, (2) on the *P*-value of the *n*-way model, and (3) on *P*-values of the "trailing" simpler models. The *P*-value thresholds nos. 2 and 3 can be the same or different. Here is an example applied in this study to AGS histories: (1) EAF $\pm$ 2 SE $\in$ (0, 1); (2) the *P*-value for an *n*-way model $\geq$0.01; and (3) *P*-values for "trailing" simpler models are <0.01. Other versions of this criterion are also found in the literature: with different values of the *P*-value cutoff (usually 0.05, 0.01, or 0.001); with limits on EAF, but not on their SEs; and with or without conditions on simpler models. In the case of SSL histories, 36 composite feasibility criteria were tested |
| Feasible (or fitting) *qpAdm* model | A *qpAdm* model satisfying the feasibility criteria above |
| Predicted positive/negative admixture model | An admixture model supported/not supported by 1 or several analytical tools such as *qpAdm*, PCA, *ADMIXTURE* |
| Positive rate | The fraction of all tested *qpAdm* models that are predicted positive (feasible, fitting) |
| FDR | The fraction of feasible admixture models of a certain complexity (e.g. 2-way) that are classified as false in the case of AGS and as nonoptimal in the case of SSL simulations. For topological criteria used for classifying admixture models on AGS histories into true and false ones, see *Results* and *Methods* sections. For approaches to calculating FDR in the SSL context, see the definition for "optimal and nonoptimal models" in Box 2. A general formula for FDR is as follows (equation 1): $$FDR = 1 - PPV = \frac{FPR}{\left(\frac{true}{false} - FNR \times \frac{true}{false} + FPR\right)}$$ • PPV is positive predictive value, • FPR is false-positive rate, also known as type I error probability ($\alpha$) = 1−true-negative rate, • FNR is false-negative rate, also known as type II error probability ($\beta$) = 1−TP rate = 1−power, $\frac{true}{false}$ is the ratio of the number of true or optimal models to false or nonoptimal models in the set of tested models, also known as prestudy odds (Ioannidis 2005) |
| False omission rate | The fraction of feasible *qpAdm* models that are classified as true considering the simulated graph topology and simulated admixture proportions but are not supported by another method (*qpAdm* model competition, PCA, or *ADMIXTURE*) or a combination of methods |
| High-throughput *qpAdm* protocol or admixture screen | A protocol, which for a target group, tests all possible combinations of *k* proxy sources from a set of *n* alternative sources. When proxy sources are sampled over a large genetic landscape (in an exploratory study) and not from a small list of probable ancestry sources for a target (in a focused study), such study designs lead to low prestudy odds, especially for large *k*, and hence to high FDR according to [equation (1)] above |
| "Rotating" *qpAdm* protocol | A protocol having the following feature: a large subset of reference populations or all of them are distributed between the "right" and "left" sets according to the principle "whatever is not in the left set is in the right set," testing all possible bisections of this sort for a given rotated set and a given range of model complexities. In the most extreme case, target groups are also included in this rotation. The goal of this approach, when compared with "nonrotating" protocols, is to increase the power of the method to reject nonoptimal proxy ancestry sources (Harney *et al.* 2021) |
| "nonrotating" *qpAdm* protocol | This protocol we also termed as "standard" or "basic": all models are tested with 1 or few fixed sets of "right" groups, which usually predate "left" groups or are contemporaneous with them. In practice, modern populations genetically divergent from the target are often included in such a "right" set if ancient reference groups are unavailable |
| "Distal" *qpAdm* model | A model where the target postdates or is contemporaneous with all proxy sources; in other words, they are temporally stratified |
| "Proximal" *qpAdm* model | A model where the target predates at least one proxy source |
| "Distal" *qpAdm* protocol | A rotating or nonrotating *qpAdm* protocol considering distal models only |

**Box 1. (Continued)**

| | |
|---|---|
| "Proximal" *qpAdm* protocol | A rotating or nonrotating *qpAdm* protocol without temporal stratification attempted, that is considering both proximal and distal models. Such a protocol should not be confused with a protocol deliberately considering proximal models only |
| Model competition | A *qpAdm* protocol starts from sets of alternative feasible *qpAdm* models of a certain complexity level for a given target, outcomes of a nonrotating protocol. Alternative proxy sources from these models form a rotated set. Rotation can be performed in 2 ways (groups from the rotated set are placed in the "left" set one by one, and the other groups from the rotated set are placed in the "right" set, or groups from the rotated set are placed in both "left" and "right" sets one by one), and a composite feasibility criterion is applied. The goal of this approach is to increase the power of the method to reject nonoptimal proxy ancestry sources |
| Proximal model competition *qpAdm* protocol | As introduced by Narasimhan *et al.* (2019), this is a nonrotating *qpAdm* protocol with temporal stratification of "right" and "left" sets, with a subsequent model competition step and with no (or very limited) temporal stratification of targets and proxy sources at both steps |
| Maximal absolute difference between EAFs and equal fractions | This metric, abbreviated as max\|EAF−EF\|, measures how far *EAFs* in an admixture model deviate from equal fractions ($\frac{1}{n}$ for an *n*-way model) |
| Maximal SE of EAF | This is a maximal SE reported by *qpAdm* for admixture fractions in a model |
| $AEH_T-av.\ AEH_S$ | Difference between number of admixture events in a target's history and average number of admixture events in sources' histories (in the context of an AGS history) |

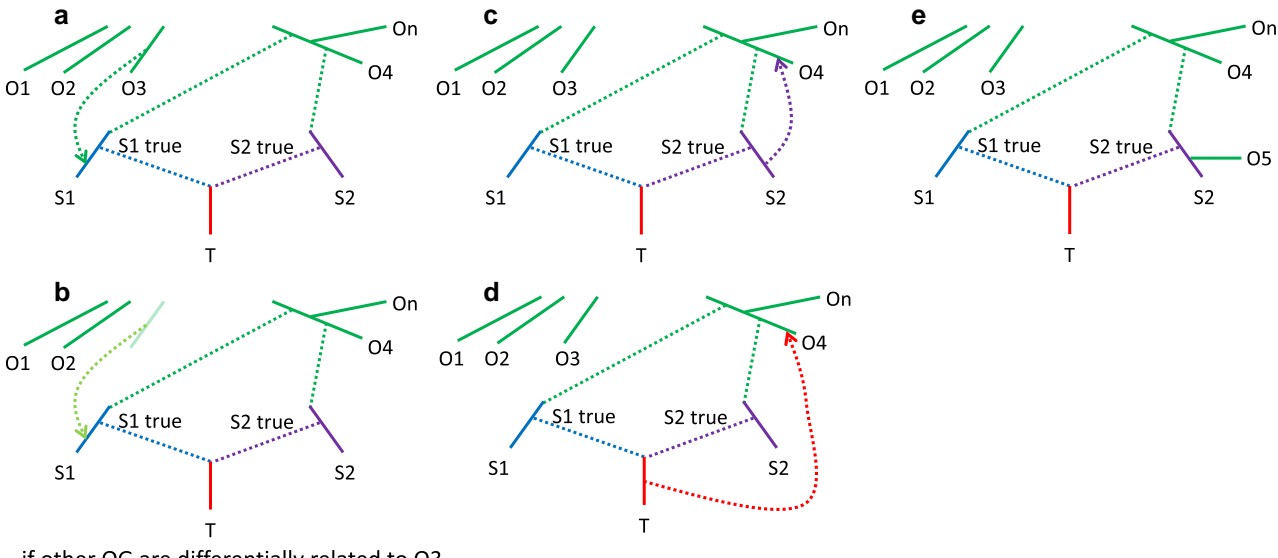

**Fig. 1.** Admixture graphs showing an exhaustive list of assumption violations of the standard *qpAdm* protocol that may lead to rejection of the true simple model and thus prompt the researcher to test overly complex models. a) A gene flow from an outgroup (OG) O* to a proxy source after the divergence of the latter from the true source. b) A gene flow from an unsampled source to a proxy source after the divergence of the latter from the true source. This case is problematic only if the OGs are differentially related to the unsampled source. c) A gene flow from a proxy source to an OG after the divergence of the former from the true source. d) A gene flow from a target to an OG. e) An OG is cladal with a proxy source.

2021). See Box 1 for definitions of various *qpAdm*-related terms used in this study.

Ten years later, we find *qpAdm*-based protocols routinely employed in large-scale screens of ancient human or animal populations for admixture (often between closely related sources) and used as formal tests for admixture (see Lazaridis *et al.* 2016, 2022; Skoglund *et al.* 2017; Harney *et al.* 2018; Mathieson *et al.* 2018; Antonio *et al.* 2019; Narasimhan *et al.* 2019; Fernandes *et al.* 2020; Marcus *et al.* 2020; Wang *et al.* 2020; Yang *et al.* 2020; Ning, Li, *et al.* 2020; Carlhoff *et al.* 2021; Librado *et al.* 2021; Papac *et al.* 2021; Sirak *et al.* 2021; Wang *et al.* 2021; Yaka *et al.* 2021; Zhang *et al.* 2021; Bergström *et al.* 2022; Gnecchi-Ruscone *et al.* 2022; Maróti *et al.* 2022; Oliveira *et al.* 2022; Patterson *et al.* 2022; Changmai, Jaisamut, *et al.* 2022; Changmai, Pinhasi, *et al.* 2022; Brielle *et al.* 2023; Lee *et al.* 2023; Nakatsuka *et al.* 2023; Taylor *et al.* 2023; Zeng *et al.* 2023; Speidel *et al.* 2025 for examples).

*qpAdm* fits admixture models to a matrix of $f_4$-statistics of the form $f_4$("left" group$_i$, "left" group$_j$; "right" group$_i$, "right" group$_j$), which in the absence of missing data at the group level can be reduced to a smaller matrix $f_4$(target group, "left" group$_j$; "right" group$_1$, "right" group$_j$), considering algebraic relationships between different $f_4$-statistics (Peter 2016).

A *qpAdm* protocol that has become the standard in archaeogenetics (Lazaridis *et al.* 2016) can be broken down into 2 parts: estimation of the number of gene flows connecting the "right" and "left" sets (this method was first published as a tool named "*qpWave*"; Reich *et al.* 2012) and inference of admixture proportions in a target group in the "left" set (Haak *et al.* 2015). *qpWave* tests for the number of distinct gene flows connecting the "right" and "left" population sets, does not infer directionality of gene flows, and does not identify recipients of gene flow in the "left" or "right" population sets. Notably, the standard *qpAdm* protocol

relies on the following assumptions in the admixture graph paradigm (Lazaridis *et al.* 2016; Harney *et al.* 2021): (1) there is at least one "right" population differentially related to the proxy sources; (2) proxy sources are strictly cladal with the true ancestral admixing sources (Fig. 1a and b), and (3) there are no gene flows to populations located in the "right" set from the proxy source or target lineages either after the split of the proxy source from the true admixing source population or between the target population and the admixture event that gave rise to it (Fig. 1c–e). In the context of our study, true sources are unsampled populations that participated in a simulated admixture event (labeled as "S1 true" and "S2 true" in Fig. 1, see also Box 1). A more concise definition of these assumptions exists: there should be no genetic drift shared exclusively by proxy sources and "right" populations (but not by the target) or shared exclusively by the target and "right" populations (but not by proxy sources).

If the above assumptions are satisfied, it is safe to say that *qpWave/qpAdm* rejections of simpler models, and a failure to reject more complex models, are due to a genuinely complex admixture history that connects the source and target populations rather than due to a false rejection of a simple model due to violations of any one of the assumptions described above. Most notably, violations of the 2nd or 3rd assumptions raise the probability of rejecting a simpler (true) model if enough data are available and prompt the researcher to test more complex (false) models (such as in Fig. 1 rejecting a 2-source *qpAdm* model and exploring 3-source models).

Harney *et al.* (2021) demonstrated on AGS (admixture-graph-shaped) simulated histories that if the *qpAdm* assumptions are satisfied, it is highly favorable for statistical power of the method (for distinguishing between alternative proxy sources that are unequally distant genetically from the true ancestry source) to move at least some alternative proxy sources between the "left" and "right" sets. In other words, having "on the right" populations who do not violate the topological assumptions of *qpAdm* but are closely related to proxy sources "on the left" increases the statistical power greatly (see also Ning, Fernandes, *et al.* 2020 and Williams *et al.* 2024 for demonstrations of this on simple and complex AGS simulated histories, respectively, and Speidel *et al.* 2025 for another exploration of a *qpAdm*-like algorithm on simple AGS simulations).

This new type of *qpAdm* protocol, termed the "rotating" protocol, has been adopted in archaeogenetics widely (see, e.g. Skoglund *et al.* 2017; Harney *et al.* 2019; Narasimhan *et al.* 2019; Olalde *et al.* 2019; Carlhoff *et al.* 2021; Fernandes *et al.* 2021; Librado *et al.* 2021; Bergström *et al.* 2022; Lazaridis *et al.* 2022; Oliveira *et al.* 2022; Taylor *et al.* 2023; Zeng *et al.* 2023; Speidel *et al.* 2025). The most extreme version of the "rotating" protocol simply divides a set of reference populations into all possible combinations of "right" and "proxy source" subsets of certain sizes and rotates these combinations through successive *qpAdm* analyses.

Additional constraints can be placed on the rotating combinations such as restricting a set of groups (usually highly divergent from the target) to remain in the "right" set in all models. When evaluating the totality of multiple *qpAdm* tests, the simplest feasible models (e.g. 1-way, i.e. unadmixed) are usually favored, and increasingly complex models are explored upon the rejection of simpler models. Model rejection is made according to a chosen *P*-value threshold such that *qpAdm* models are considered feasible or "fitting" the data when the *P*-value is above such a threshold (Skoglund *et al.* 2017; Harney *et al.* 2018; Narasimhan *et al.* 2019; Olalde *et al.* 2019; Yang *et al.* 2020; Carlhoff *et al.* 2021; Fernandes *et al.* 2021; Librado *et al.* 2021; Zhang *et al.* 2021; Bergström *et al.* 2022; Lazaridis *et al.* 2022; Oliveira *et al.* 2022; Nakatsuka *et al.*

2023; Taylor *et al.* 2023; Zeng *et al.* 2023; Kerdoncuff *et al.* 2024; Speidel *et al.* 2025). As an additional criterion of a fitting model, all estimated admixture proportions (see, e.g. Harney *et al.* 2018; Olalde *et al.* 2019; Yang *et al.* 2020; Zhang *et al.* 2021; Lazaridis *et al.* 2022; Oliveira *et al.* 2022; Carlhoff *et al.* 2023; Zeng *et al.* 2023; Kerdoncuff *et al.* 2024; Speidel *et al.* 2025), or proportions ±2 SEs (Narasimhan *et al.* 2019), may be required to lie between 0 and 1. We note that estimated admixture fractions (EAFs) not restricted to [0, 1] are reported by *qpAdm* by default, and this setting is used in a great majority of published studies (see the study by Harney *et al.* 2021 for a justification of this unusual property of the algorithm). It is important to remember that the statistical significance of the *qpAdm/qpWave* test is, strictly speaking, a function of model rejection, and thus the failure to reject a model may have underlying causes other than approximating the true situation well enough (such as lack of statistical power or a lack of suitable "right" groups that capture the divergent ancestry sources among the "left" set of populations).

A less exhaustive version of the rotating *qpAdm* protocol, termed "model competition" (e.g. Narasimhan *et al.* 2019; Fernandes *et al.* 2020; Carlhoff *et al.* 2021; Sirak *et al.* 2021; Zhang *et al.* 2021; Maróti *et al.* 2022; Brielle *et al.* 2023; Lee *et al.* 2023), is used as widely as the basic rotating protocol. It involves an initial (standard nonrotating) *qpAdm* analysis on a few source populations (see Box 1). Upon identifying a list of plausible sources for a target, the algorithm retests feasible models for this target rotating these plausible sources between the "left" and "right" sets with the expectation of improving the power to reject models including proxy sources that are genetically distant from the true sources.

The rotating *qpAdm* protocol and model competition are increasingly used as central methods for testing admixture hypotheses proposed after inspecting distributions of individuals in PC spaces, outcomes of *ADMIXTURE* analyses, and *f/D*-statistics indicative of a gene flow rather than a simple tree relationship. Yet, the first study reporting detailed testing of *qpAdm* on simulated data (Harney *et al.* 2021) was performed in extremely favorable conditions: the simulated AGS history included just 2 nonnested admixture events; the sources for the principal target group diverged about 1,200 generations ago (almost 35,000 years ago in the case of humans); the proxy sources were strictly cladal with the actual ancestral groups for the target; several groups differentially related to these ancestry sources were available; the simulated data were free of ascertainment bias since sites were sampled in a random way; 1 million sites were used for most analyses; and only 50/50% simulated admixture proportions were tested for some analyses. That study confirmed that the method behaves as expected under these ideal conditions and offered some important guidelines on the choice and number of "right" populations for optimal specificity of the method and on model comparison strategies and also showed that the results are robust to the presence of missing data, imbalanced group sizes, ancient DNA damage, a wide range of genomic block sizes for the jackknife algorithm, and to a particular type of SNP ascertainment bias: selecting sites heterozygous in 1 individual from a certain population (Patterson *et al.* 2012). Among challenging historical scenarios, only multifurcation followed by uniform and continuous gene flows in a 1D stepping stone model composed of 6 (Harney *et al.* 2021) or 9 (Speidel *et al.* 2025) groups was explored in the few studies aimed at testing *qpAdm* performance. Meanwhile, false discovery rate (FDR) of the method and violations of the topological assumptions of *qpAdm* (Fig. 1) remained virtually unexplored, and 2D stepping stone landscapes (SSLs, see Box 1) and FDR in situations when the ratio of true to false models in the set of models tested (also termed "prestudy odds"

by Ioannidis 2005, see Box 1) is low were completely untouched by these simulation studies (Harney *et al.* 2021; Speidel *et al.* 2025). Thus, the method was proven to work in extremely favorable conditions. But what happens in other cases where arguably most of the history of humans and other species fits: a history that is not a nearly perfect tree and can probably be approximated by complex admixture graphs or 2D SSL? This question motivated the current study exploring both simulated AGS and SSL histories and a companion paper focused on AGS histories (Williams *et al.* 2024).

As detailed above, various versions of *qpAdm* protocols form a core of many recent archaeogenetic studies. These protocols are usually aimed at finding the simplest feasible *qpAdm* models for target groups, where feasibility is defined by setting a threshold for *qpAdm/qpWave P*-values and by setting plausibility criteria for admixture proportions estimated by *qpAdm*. Finding a feasible 2-way or more complex admixture model for a target is often interpreted as solid evidence for gene flow, especially if PCA and *ADMIXTURE* methods confirm the same signal. Thus, *qpAdm* protocols are used as supposed formal tests for admixture, whereas the latter 2 methods are not formal tests.

Relying on general principles, we argue that any high-throughput *qpAdm* protocol (Box 1) on poorly understood complex demographic relationships approximated as admixture graphs and/or SSL is questionable as a formal test for admixture since a *P*-value threshold is used to *reject*, but not to *accept* models, and it is safer to interpret those models as a certain number of gene flows connecting "left" and "right" sets in any direction, not in the form of proxy sources and admixture proportions for a target. A model feasibility criterion including both *P*-values and EAF is a complex construct relying on the topological assumptions outlined in Fig. 1. We expect that taking "left" and "right" sets that are not well separated in time or contemporaneous (Supplementary Fig. 1), and where relationships among groups are poorly understood (which is almost always true for exploratory studies), enriches the system for "left-to-right" gene flows, which in turn leads to frequent rejection of true simple admixture models. Since the behavior of *qpAdm* EAF under diverse demographic scenarios is poorly understood, it is possible that a large fraction of nonrejected complex models emerges as feasible, resulting in false signals of admixture. Another approach probably leading to the same outcome (enriching the system for "left-to-right" gene flows and other violations of the topological assumptions) is selection of pairs of "right" populations maximizing deviations of statistics $f_4(\text{left}_i, \text{left}_j, \text{right}_i, \text{right}_j)$ from 0 since such deviations due to tree-like topologies or "left-to-right" gene flows are impossible to tell apart. This approach is sometimes used for increasing the power of *qpAdm* to discriminate between alternative proxy sources.

However, the extent to which these approaches are problematic remains to be explored, especially in the case of 2D SSL. To address this, we analyze both simulated AGS histories of random topology and fixed complexity and SSL histories with randomized nonuniform gene flow intensities on a grid of fixed size and test various types of high-throughput *qpAdm* protocols common in the literature: rotating and nonrotating, with or without temporal stratification of target groups and proxy ancestry sources and with or without a model competition step. We also reproduced other aspects of a typical archaeogenetic study on AGS simulated data: we combined various *qpAdm* protocols with PCA, $f_3$-statistics, and an unsupervised *ADMIXTURE* analysis to explore FDR of complex "admixture screening pipelines." In the last section of our study, we reanalyzed from the SSL perspective large sets of published

*qpAdm* models based on real data (Zeng *et al.* 2023; Speidel *et al.* 2024) and interpreted the resulting patterns using insights from the extensive exploration of simulated SSL data.

## Results

Since the space of simulation parameters, *qpAdm* protocols, and interpretative approaches we explore is complex, we first provide a section-by-section guide to our results (see Boxes 1 and 2 for a terminology glossary):

1) *AGS histories: approach to simulation and testing of qpAdm protocols*: In this section, we introduce our AGS simulations and 4 high-throughput *qpAdm* protocols (rotating and nonrotating, distal, and proximal).
2) *AGS histories: case studies illustrating false and true feasible qpAdm models*: We introduce our classification of fitting (feasible) models into true and false ones that is based on a set of topological criteria. We discuss selected false and true fitting *qpAdm* models along with PCA and *ADMIXTURE* results for the respective simulated demes.
3) *AGS histories: influence of temporal stratification and data amount/quality on the performance of qpAdm protocols*: We first make an overview of all *qpAdm* testing outcomes for 2-way (2 source) admixture models by placing them in the coordinates of *qpAdm P*-values and a metric derived from EAF. Next, we apply a composite *qpAdm* model feasibility criterion borrowed from the literature and show that given the prestudy odds in our experiments with 12 or 6 alternative proxy sources per target, FDR exceeds 50% when proxy sources have admixture history more complex than that of the target, and that situation is more common in proximal than in distal admixture models.
4) *Introducing SSL histories*: We describe a very different kind of simulations: 2D SSLs based on a hexagonal lattice (formed by triangles tiling the plane), with nonuniform gene flow intensities between neighboring demes.
5) *SSL histories: randomized sampling of landscapes*: We explain why sets of targets, proxy sources, and outgroups drawn randomly from a 2D SSL make a good conservative baseline for a typical archaeogenetic sample set. We introduce *qpAdm* protocols on SSL that reproduce high-throughput *qpAdm* protocols popular in the literature (with a typical range of model complexities and typical numbers of source combinations per target) and result in low prestudy odds for complex models.
   a. *qpAdm performance at the level of models: model optimality metrics in spaces of EAFs and P-values*: Since strict delineation of true and false models in the SSL context is impossible, we make a detour in our analyses and start from a bird's-eye view on millions of 2- to 4-way randomized *qpAdm* models we tested on various types of SSL. For that, we place them in the space of *P*-values vs EAF introduced in the section *AGS histories: influence of temporal stratification and data amount/quality on the performance of qpAdm protocols* and inspect how geometric properties of admixture models (also termed "model optimality metrics") map on this space. We show that a great majority of models with all EAF between 0 and 1 correspond to symmetric arrangements of sources around targets.
   b. *qpAdm performance at the level of models: feasible models in spaces of model optimality metrics*: We apply various

composite model feasibility criteria (based on both EAF and *P*-values, as is typically done in the *qpAdm* literature) and place feasible models in a space formed by key optimality metrics introduced above: maximal or average source–target (ST) distance and minimal ST–source (STS) angle. We juxtapose in this space sets of models to be tested and sets of feasible models, define a threshold separating optimal from nonoptimal models, and proceed to calculate prestudy odds, false-positive rate (FPR), false-negative rate (FNR), and FDR. We show that it is usually impossible to extract a needle (models with symmetric sources close to the target) from a haystack of models with asymmetrically arranged distant sources, given that the variance of *P*-values is high (resulting in high FNR and low but nonzero FPR) and that EAFs between 0 and 1 are reported for a small percentage of these abundant nonoptimal models (also contributing to FPR).

c. *qpAdm performance at the level of experiments*: We go beyond individual admixture models and look at "experiments," an approach common in the *qpAdm* literature: all possible source combinations are tested for a target, moving from simple to more complex models, and an experiment is stopped as soon as at least one model satisfying a composite feasibility criterion is found. We demonstrate that on randomly sampled SSL, results of experiments are often misleading: depending on the choice of feasibility criteria and on SSL properties, either accurate but simple (2-way) models are found or inaccurate complex (3- or 4-way) models are reported or experiments progress even further. We discuss factors influencing complexity of models at which experiments end, including average "left–right" spatial distance and an important difference between rotating and nonrotating protocols in this respect.

6) *SSL histories: systematic sampling of landscapes*: We move from randomized to systematic construction of admixture models on SSL, with proxy sources and outgroups drawn from circles of demes at various distances around the target.

a. *Systematic sampling: model optimality metrics in the spaces of EAFs and P-values*: In this subsection, we apply the analytic approaches introduced in sections *qpAdm performance at the level of models: model optimality metrics in spaces of EAFs and P-values* and *qpAdm performance at the level of models: feasible models in spaces of model optimality metrics* to the systematic models and explore the influence of average "left–right" distances on *qpAdm* results.

b. *Systematic sampling: exploring truly ideal 6-way models*: We move to even more complex, 6-way, models and show that separating highly nonoptimal and optimal complex models with *P*-value thresholds is achievable only with the nonrotating *qpAdm* protocol but not the rotating one.

c. *Systematic sampling: qpAdm performance at the level of experiments*: We show that high-throughput *qpAdm* protocols applied to SSL can produce counter-intuitive results: when sampling of the target's vicinity or the whole landscape is dense, simple (2-way) models are most frequent outcomes of *qpAdm* experiments, but when sampling of the target's vicinity is poor, experiments tend to stop at much more complex, and inaccurate, models. We discuss how these patterns are explained by average "left–right" spatial distance influencing model *P*-values.

7) *Geographical analysis of large sets of qpAdm models on real data*: In this section, we explore geometric properties of admixture models from 2 sets of high-throughput *qpAdm* experiments on real human archaeogenetic data (Zeng *et al.* 2023; Speidel *et al.* 2024). Interestingly, distributions of fitting *qpAdm* models in the spaces of optimality metrics are similar on simulated and real landscapes matched by sampling

---

**Box 2. Definitions relevant for *qpAdm* models on SSL.**

| | |
|---|---|
| Maximal (or average) source–target distance (max. ST distance, average ST distance) | The largest (or average) spatial distance on an SSL between source demes and the target deme within one admixture model. In this study, noncontemporaneous samples from the same deme were assigned a distance of 0, the nearest neighbors a distance of 1, and so on |
| Minimal source–target–source angle (min. STS angle) | The smallest angle formed by any pair of proxy sources and the target in an admixture model in the context of an SSL. If location of a source coincides with the target deme, minimal STS angle is undefined |
| Average "left–right" distance | Average spatial distance between demes in the "left" and "right" sets in a *qpAdm* model. Smaller distance increases the probability of encountering "proxy-to-right" and "right-to-proxy" gene flows bypassing the target and of "target-to-right" gene flows bypassing the proxy sources or, more formally, of observing genetic drift shared exclusively by the proxy sources and "right" populations (but not by the target) or shared exclusively by the target and "right" populations (but not by the proxy sources). These violations of the *qpAdm* assumptions (Fig. 1) make *P*-values smaller on average |
| Symmetrically arranged proxy sources | Demes arranged on an SSL with respect to the target at 180°, 120°, or 90° (for 2-, 3-, and 4-way models, respectively) |
| Ideal symmetric model | In the case of 2-way and more complex models on an SSL, these are models where all sources are the nearest neighbors of the target (distance = 1 in this study) and are arranged symmetrically. Although models with a source at a distance of 0 are even better, they were not included in this definition since for them min. STS angles were undefined. Due to the geometry of the underlying hexagonal lattice, we lacked ideal 4-way models. Although we use the term "ideal," these models are ideal only in the context of the respective model complexity class since full description of target's ancestry on our simulated SSL requires up to 7 sources: the direct ancestor and up to 6 neighboring demes |
| Distance to the ideal symmetric model (in the space of optimality metrics) | Euclidean distance in a 2D space formed by 2 model metrics: max. (or average) ST distance and min. STS angle. Min. STS angles are normalized: they are divided by a constant such that the maximal angle (180°, 120°, or 90° for 2-, 3-, and 4-way models, respectively) equals 9, which is the maximal possible ST distance on our landscapes. Distance in the space of |

Box 2. (Continued)

| | optimality metrics is measured between a given admixture model and the ideal symmetric model of the same complexity |
|---|---|
| Model optimality metrics | The following metrics are collectively termed "optimality metrics": max. (or average) ST distance, min. STS angle, and distance to the ideal symmetric model in the optimality metric space |
| Optimal and nonoptimal models | For calculating prestudy odds, FPR, FNR, and FDR in the SSL context, we must set a threshold for classifying models into optimal and nonoptimal ones. Such a threshold was based on distance to the ideal symmetric model in the space of max. ST distances and min. STS angles and was set at ≤5 (it equals the median value of this metric in the whole dataset divided by 2 and rounded). Obviously, other thresholds can be used for illustrating the same patterns in the data. Alternatively, we considered models including only the nearest neighbors and/or direct ancestors of a target as optimal and the other models as nonoptimal |
| Landscape sparsity | Landscape sampling density depends both on the range of gene flow intensities simulated and on the number of demes selected for an analysis. Landscapes with lower gene flow intensity and/or deme sampling density for *qpAdm* are sparser, given their equal size in demes and identical topology. We use a term "dense landscape" instead of "landscape with low sparsity" |

sparsity. This illustrates low prestudy odds and high FDR in published *qpAdm* admixture screens and suggests that our conclusions derived on simulated SSL are relevant for critical reassessment of the existing archaeogenetic literature.

## AGS histories: approach to simulation and testing of *qpAdm* protocols

Below we explore, relying on simulated AGS histories, performance (mainly FDR) of *qpAdm* protocols representing the spectrum of protocols used in the literature (for an overview of our AGS workflow, see Supplementary Fig. 2). The most extreme example is a protocol where all groups are rotated and all are treated alternatively as outgroups, targets, and proxy sources, i.e. there is no temporal stratification between the latter categories. We term this protocol as "proximal rotating." Although such an extreme situation is, to our knowledge, rare among published *qpAdm* protocols (see Carlhoff *et al.* 2021; Oliveira *et al.* 2022; Zeng *et al.* 2023), we use it to illustrate the potential effects of poor temporal stratification of targets and proxy sources in published rotating protocols (Supplementary Fig. 1 and Text 1). Models with targets predating proxy sources are encountered in high-throughput *qpAdm* screens but do not constitute a majority of models (Narasimhan *et al.* 2019; Librado *et al.* 2021; Bergström *et al.* 2022; Lazaridis *et al.* 2022; Taylor *et al.* 2023; Zeng *et al.* 2023; Speidel *et al.* 2025). We also explore FDR of the proximal nonrotating (Harney *et al.* 2018; van de Loosdrecht *et al.* 2018; Narasimhan *et al.* 2019; Prendergast *et al.* 2019; Sikora *et al.* 2019; Wang *et al.* 2020; Carlhoff *et al.* 2021; Wang *et al.* 2021; Zhang *et al.* 2021; Maróti *et al.* 2022; Changmai, Jaisamut, *et al.* 2022; Changmai, Pinhasi, *et al.* 2022; Brielle *et al.* 2023; Lee *et al.* 2023; Nakatsuka *et al.* 2023), distal rotating (Narasimhan *et al.* 2019; Librado *et al.* 2021; Bergström *et al.* 2022; Lazaridis *et al.* 2022; Taylor *et al.* 2023; Zeng *et al.* 2023; Speidel *et al.* 2025), and distal nonrotating protocols (Haak *et al.* 2015; Mathieson *et al.* 2015, 2018; Lazaridis *et al.* 2016; Antonio *et al.* 2019; Sikora *et al.* 2019; Marcus *et al.* 2020; Yang *et al.* 2020; Papac *et al.* 2021; Yaka *et al.* 2021; Patterson *et al.* 2022). In the distal protocols, only *qpAdm* models where the target group's sampling date is strictly contemporaneous with, or postdates, sampling of both proxy sources were considered.

We tested performance of these *qpAdm* protocols on relatively complex AGS simulated histories: 13 populations connected with each other by admixture graphs of random topology, including 10 pulse-like admixture events. Further details on the simulated population histories are presented in Supplementary Fig. 2 and in *Methods*, and illustrated by examples in Fig. 2 and Supplementary Fig. 3. To explore the influence of data amount and population divergence levels on *qpAdm* performance, we generated 3 independent sets of 10 simulation replicates for each graph topology (referred to as setups nos. 1, 2, and 3 below) and single simulation replicates for 2 further setups (no. 4 and 5; see Supplementary Fig. 2 for details and Supplementary Fig. 4 for the number of SNP loci polymorphic in the simulated datasets). The simulations with the 3,000-generation maximal history depth resulted in median $F_{ST}$ at the intercontinental levels of human genetic differentiation (~0.1) and those with the 800-generation depth matched intracontinental levels (median $F_{ST}$ from 0.022 to 0.051, see Supplementary Fig. 5).

A typical archaeogenetic dataset is composed of pseudohaploid data with high proportions of missing sites and with widely different group sizes, including singleton groups. To generate more realistic "noisy" data, we also performed randomized subsampling of SNP datasets for simulation setup nos. 1, 2, and 3 (300-, 1,000-, and 3,000-Mb-sized genomes, respectively), for 1 simulation replicate per each setup (Supplementary Fig. 2; see *Methods* for details). The resulting data were pseudohaploid, had missing data rates ranging from 5 to 95% across individuals, and had uneven group sizes ranging from 1 to 10 individuals. Ten independent subsampled datasets were generated for each simulated topology (for counts of polymorphic loci, see Supplementary Fig. 4).

The *qpAdm* protocols we applied to the simulated data were focused on the simplest admixture models, 1- and 2-way models. The median number of 2-source admixture events in a population from our AGS simulations is 1 (average = 1.17), suggesting that 2-way models are broadly adequate for this type of history, especially considering that by far not all internal branches in AGS histories were sampled (Supplementary Fig. 3). Histories that are more complex than 2-way mixtures were also not rare in the data, accounting for 38% of all models tested (Supplementary Table 1a). The model feasibility criteria followed the study by Narasimhan *et al.* (2019) (Box 1). Thus, we tested all possible 2-way admixture models for 40 complex population histories (34,320 proximal and distal rotating models per simulation setup and replicate).

The nonrotating *qpAdm* approach was implemented as follows: for each simulated graph, the 6 most ancient groups were selected as a fixed "right" set (ties were resolved in alphabetical order; these "right" sets remained unchanged for a given graph topology across independent simulations), and for the remaining 7 groups, all possible 1-way and 2-way admixture models were tested (4,200

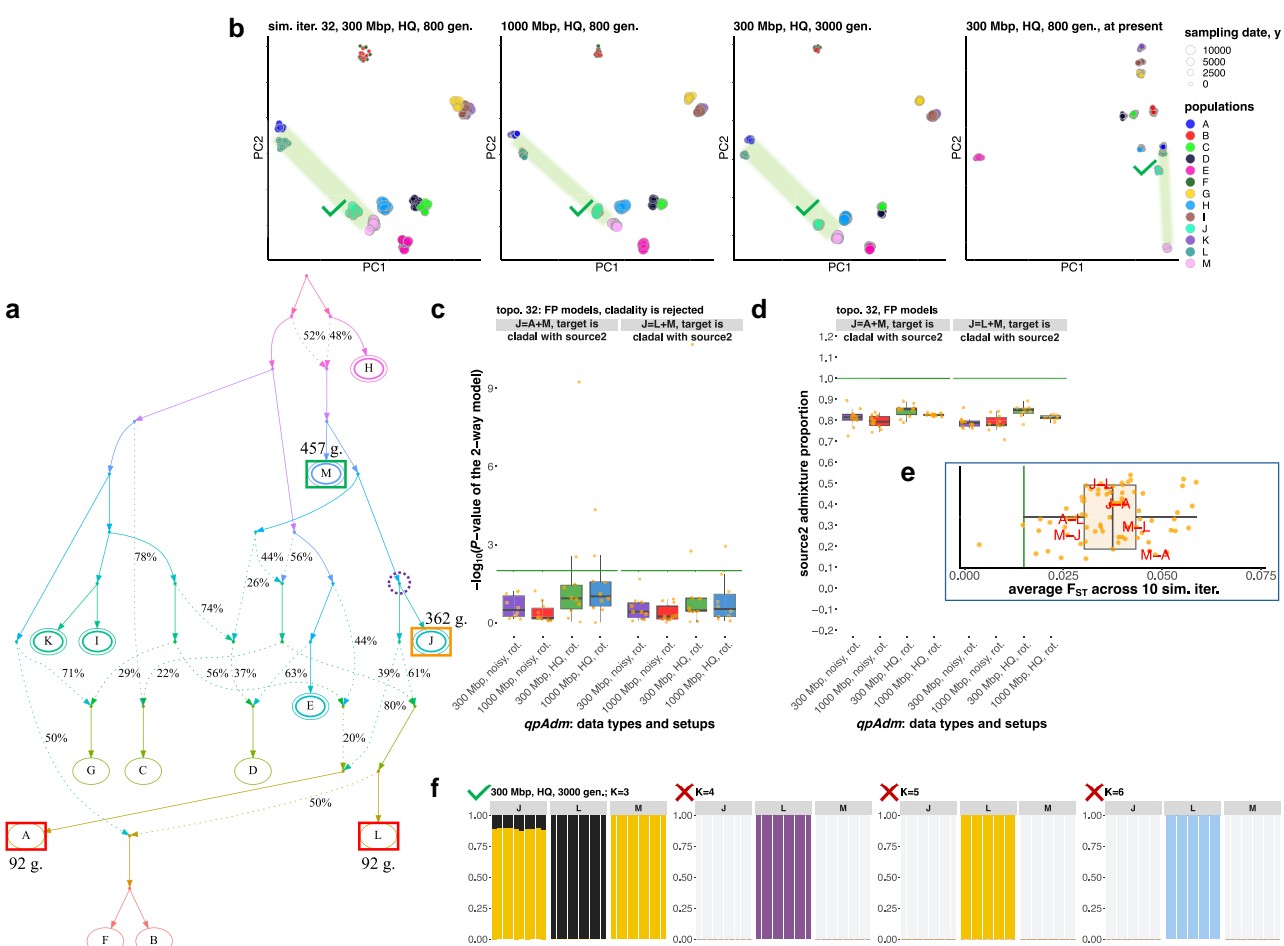

**Fig. 2.** A case study illustrating the most common class of FP *qpAdm* models supported by the proximal rotating protocol. Models of this type include at least one proxy ancestry source that is simulated as fully cladal with the target. The other proxy source may be simulated as a descendant of the target lineage (as shown here), may belong to an unrelated lineage (Supplementary Fig. 6a), or may be also cladal with the target. Both models shown here, "*J = A + M*" and "*J = L + M*," are also fully supported by 3D PCA and by an unsupervised *ADMIXTURE* analysis at 1 or more K values. a) Simulated AGS history (only topology is shown here; for divergence/admixture dates and effective population sizes, see the respective simulated history shown in Supplementary Fig. 3). Sampled populations are marked by letters. The target population from the *qpAdm* models illustrated here is enclosed in an orange rectangle; correct proxy source is in a green rectangle, and inappropriate proxy sources are in red rectangles. Sampling dates for these groups (in generations before present) are shown beside the rectangles (the dates are from the simulations up to 800 generations deep). True simulated ancestry source(s) for the target are enclosed in dashed violet circles. Six populations used as outgroups for the nonrotating *qpAdm* protocol are enclosed in double ovals. b) Two-dimensional PCA plots for 4 simulated high-quality datasets are indicated in plot titles. Simulated populations are colored according to the legend on the right, and larger points correspond to older sampling dates. The space between the proxy sources is shaded in green. If admixture model(s) are supported by 3D PCA (PC1 vs PC2 vs PC3) according to the criteria listed in *Methods*, a green tick mark is placed beside the target group in the plot, and a red cross mark is placed otherwise. c) Boxplots summarizing *P*-values of 2-way *qpAdm* models (indicated in plot titles) across simulation or subsampling replicates, grouped by simulation setups and *qpAdm* protocols as indicated on the x-axis. Green lines show the *P*-value threshold of 0.01 used in this study for model rejection. The *qpAdm* models shown here were not tested using the nonrotating protocol since the target falls into the set of 6 "right" populations chosen for that protocol. d) Boxplots summarize EAF across simulation or subsampling replicates, grouped by simulation setups and *qpAdm* protocols. The admixture proportions are shown either for the 1st or 2nd proxy source, as indicated on the y-axis; green lines show the simulated admixture proportion. e) All $F_{ST}$ values for population pairs from the simulated history. Average $F_{ST}$ values are shown across 10 simulation replicates for 1,000-Mb-sized genomes and high-quality data. Population pairs formed by components of the illustrated 2-way *qpAdm* model(s) are labeled in red. The green line shows median $F_{ST}$ (0.015) for the Bronze Age West Asian populations analyzed by Lazaridis *et al.* (2016) as an example of human genetic diversity at the subcontinental scale. f) Ancestry proportions estimated with unsupervised *ADMIXTURE* for the groups constituting the *qpAdm* model(s). For brevity, results are shown for 1 or 2 *qpAdm* models, for 1 simulated dataset indicated in the plot title, and for 4 selected K values. If admixture model(s) are supported by this analysis at a given K value, a green tick mark is placed beside the target group in the plot, and a red cross mark is placed otherwise.

proximal and distal models per simulation setup and replicate). Thus, models tested with the nonrotating protocol represent a subset of models tested with the rotating protocol.

## AGS histories: case studies illustrating false and true feasible *qpAdm* models

In the context of complex and random admixture graph topologies, it is hard to draw a strict boundary between true and false admixture models composed of a target and 2 proxy sources.

However, we classified the feasible *qpAdm* models into false and true ones relying on a set of rules. By far the most common class of false feasible *qpAdm* models (referred to as "false positive" or FP models), comprising 50.9% of all FP models generated by the proximal rotating protocol across all the simulation and subsampling replicates (for setup nos. 1 and 2), occurs when the target group is falsely rejected as forming a clade with one or both proxy sources while they are simulated as clades. In other words, these are false rejections of single-source models.

An example of such a situation is shown in Fig. 2. The clade relationship between the target (*J*) and source (*M*) is rejected by the rotating protocol due to "left-to-right" gene flows violating the topological assumptions of *qpAdm*, and more complex models ("*J = A + M*" and "*J = L + M*") are evaluated as true. When the true 1-way model "*J = M*" is tested, *M* represents a proxy for the only true source of ancestry in *J*, and outgroups *A*, *C*, *D*, and *L* are partially derived from the target lineage, resulting in rejection of the 1-way model with *P*-values from $\sim10^{-32}$ to $\sim10^{-50}$ (in the case of 1,000-Mb-sized genomes and high-quality data). Rejections of 2-way models "*J = A + M*" and "*J = L + M*" according to *P*-values are rare (Fig. 2c), and nearly all EAF $\pm$ 2 SE are between 0 and 1 even on low-quality pseudohaploid data (Fig. 2d), that is why these models are classified as fitting in nearly all cases (in 34 and 36 of 40 simulation/subsampling replicates, respectively; setup nos. 1 and 2 were considered). Groups *A* and *L* are 270 generations younger than *J*, and >50% of their ancestry was simulated as derived from the *J* lineage (Fig. 2a). These FP models illustrate not only cladality rejection but also incorrect gene flow direction suggested by the *qpAdm* results.

These results illustrate risks of applying rotating *qpAdm* protocols to sets including ancient groups "on the left" and much younger groups "on the right" (Carlhoff *et al.* 2021; Oliveira *et al.* 2022), since those may be descended partially from the "left" groups (Fig. 1c–e) and thus lead to *qpAdm* assumption violations and FP findings of gene flow. PCA on 4 datasets (Fig. 2b) and unsupervised *ADMIXTURE* results at some *K* values (Fig. 2f) support these FP models. Thus, a researcher coanalyzing ancient and modern groups due to lack of ancient samples from certain regions may encounter this situation in practice and, given support by all methods in the standard archaeogenetic toolkit (*qpAdm*, PCA, and *ADMIXTURE*), is likely to conclude that FP models are true. We illustrate another similar situation often leading to rejection of cladality and acceptance of 2-way or even more complex models (in 33–37 of 40 simulation/subsampling replicates, depending on model and *qpAdm* protocol; setup nos. 1 and 2 were considered) in Supplementary Fig. 6a.

Other topological classes of FP models can be concisely described as follows (and are more precisely defined in *Methods*): (1) gene flow goes from the target lineage (after the last admixture event in its history) to a proxy source lineage (Fig. 2; Supplementary Fig. 6a and b); (2) a proxy source included in the model is symmetrically related to all real sources of ancestry in the target (Supplementary Fig. 6c and d), or both proxy sources represent the same true source and are symmetrically related to all the other true sources (Supplementary Fig. 6e); (3) both proxy sources represent distinct real sources; however, a proxy source is a heavily admixed group sharing <40% of ancestry with the real source (Supplementary Fig. 6a and d); and (4) the target gets a gene flow from a deep-branching source not represented by any sampled population, and an inappropriate proxy source is included in the fitting model (Supplementary Fig. 6f and g). Violations of the topological assumptions of *qpAdm* encountered in these FP model classes and corresponding *qpAdm*, PCA, and *ADMIXTURE* results are illustrated in Supplementary Text 2. Definitions and examples of true-positive (TP) models are also presented in Supplementary Text 2.

## AGS histories: influence of temporal stratification and data amount/quality on the performance of *qpAdm* protocols

To make a qualitative overview of all the *qpAdm* results on AGS histories and to gain intuition about factors influencing frequency of false results, we placed models in the coordinates of admixture fractions estimated by *qpAdm* (EAF) and *P*-values (Supplementary Fig. 7). Both high- and low-quality data generated by the most realistic simulation setup were considered (3,000-Mb-sized genomes and maximal population divergence depth at 3,000 generations). Since EAFs reported by *qpAdm* may vary from <<−1 to >>1 (see a justification for this unusual property of the algorithm in the study by Harney *et al.* 2021) and to make this visualization approach scalable to models more complex than 2-way, we use 1 number instead of several EAF per an *n*-way model: maximal absolute difference between EAF in that model and equal fractions $\frac{1}{n}$ (max|EAF−EF|) (Box 1). Each 2-way model was tested many times: on different simulation setups, data qualities, replicates, and *qpAdm* protocols. We explore the max|EAF−EF| vs *P*-value spaces for 3 model classes separately: models that were consistently nonfitting in all our analyses, models that were fitting at least once and classified as false, and models that were fitting at least once and classified as true (Supplementary Fig. 7). For comparison, we also visualized models populating the space of EAF SE reported by *qpAdm* and *P*-values (Supplementary Fig. 8). Remarkably, the models consistently rejected (not only on the simulations used for making Supplementary Fig. 7 but also throughout this study) demonstrated EAF outside the (0, 1) interval in nearly all cases, especially on high-quality data, while models that were fitting at least once and classified as either false or true are characterized by EAF largely in the (0, 1) interval, but their *P*-values vary widely (Supplementary Fig. 7). Of the consistently rejected models, 40% are models whose target populations have 0 admixture events in their history (Supplementary Table 1a).

This qualitative overview suggests that EAF are much less variable than *P*-values, and we quantify this using standard deviation (SD) of $\log_{10}$(*P*-value) or EAF across 10 (or 20, if a model was evaluated under both rotating and nonrotating protocols) simulation/subsampling replicates (Supplementary Fig. 9). *P*-values of false, true, and consistently negative models vary by many orders of magnitude between replicates, and SD grows with increasing amount of data, up to ca. 30–60 orders of magnitude in the case of simulations that reproduce typical archaeogenetic data most faithfully: noisy data derived from 3,000-Mb-sized genomes (Supplementary Fig. 9). In contrast, median EAF SDs are consistently between 0.01 and 0.1 for models that were fitting at least once in this study (with very small differences between false and true models) and between 0.1 and 1 for consistently rejected models (Supplementary Fig. 9). This analysis confirms that there are 2 classes of 2-way *qpAdm* models on AGS data that encompass a great majority of all models: those with relatively stable EAF far from (0, 1) and those with stable EAF within (0, 1). The latter models are universally fitting very rarely (Supplementary Fig. 10) due to the high variability of *qpAdm* *P*-values.

It is important to understand if some of these 3 model classes are enriched for certain features of target's and proxy sources' admixture histories and to understand how this is shaped by the choice of *qpAdm* protocol and by temporal stratification of targets and proxy sources. According to our definitions, proximal models, and especially proximal models tested with the rotating protocol, are enriched for targets sampled deep in the past (see Supplementary Fig. 7 for a mapping on the max|EAF−EF| vs *P*-value space and Supplementary Table 1 for a quantitative summary and statistical tests). As admixture events were simulated at random points in time and placed randomly on the graphs, proximal models tested with the rotating protocol are actually enriched for unadmixed targets (Supplementary Fig. 7 and Table 1a). In contrast, proximal nonrotating and distal rotating/nonrotating models demonstrate

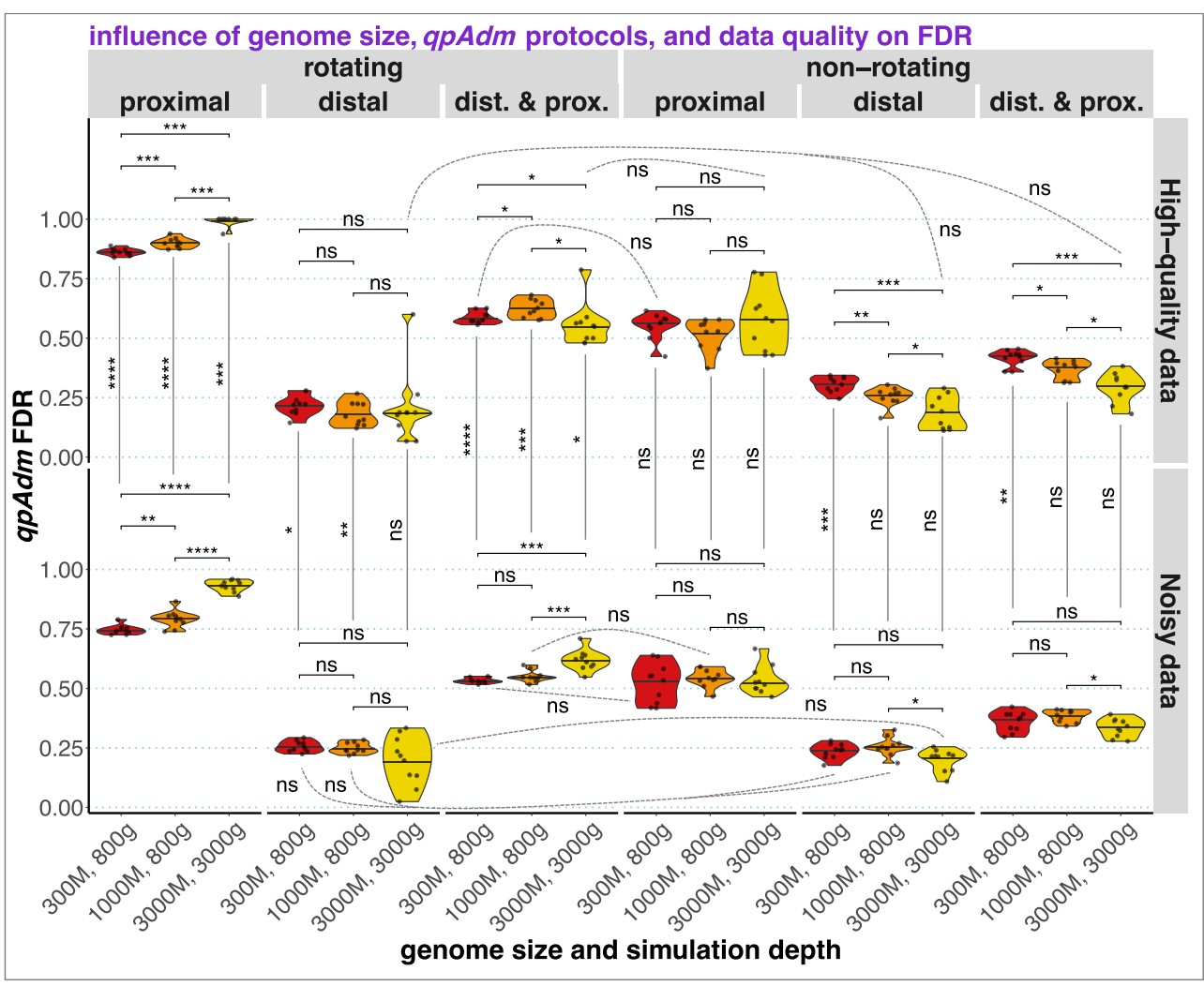

**Fig. 3.** Distributions of FDR values over 10 simulation replicates (for high-quality data) or over 10 subsampling replicates derived from a single simulation replicate (for low-quality data). The distributions are summarized as violin plots with medians and stratified by *qpAdm* protocols (rotating and nonrotating); by model subsets (proximal, distal, or both model types included); by data quality; and by 3 combinations of simulated genome sizes (300, 1,000, or 3,000 Mb) and maximal simulation depths (800 or 3,000 generations). The following statistical comparisons were performed with the 2-sided Wilcoxon test: across genome sizes, with protocol and data quality fixed (36 nonpaired tests); across data qualities, with protocol and genome size fixed (18 nonpaired tests); and across protocols, with genome size and data quality fixed (90 paired tests). Paired tests were used in the latter case since the corresponding sets of models are not independent: nonrotating models form a subset of rotating models, and distal and proximal models can be considered together. *P*-values (adjusted for multiple testing using the Holm method) are coded in the following way: ns > 0.05; * ≤ 0.05; ** ≤ 0.01; *** ≤ 0.001; **** ≤ 0.0001. Most comparisons across protocols are significant at the 0.05 level (omitted for clarity), and only nonsignificant comparisons of this type are shown with dashed lines.

unimodal, not monotonically decaying, distributions of target history complexity (Supplementary Table 1a).

For the same fundamental reasons, proximal models that were consistently nonfitting or occasionally fitting and false are characterized by fewer admixture events in a target's history when compared with average number of admixture events in sources' histories (median difference is −1 in most cases), but this is not the case for proximal models that were occasionally fitting and true (median difference is around 0, see Supplementary Table 1b). This distinction between true proximal models and the other 2 model classes is highly significant for both the rotating and nonrotating protocols (Supplementary Table 1b). Statistically significant trends in the same direction are observed for distal models as well, but in this case median differences between number of admixture events in a target's history and average number of admixture events in sources' histories (AEH$_T$−av. AEH$_S$) are positive (Supplementary Table 1b). The chance of encountering

violations of the *qpAdm* assumptions (Fig. 1) grows when admixture history of proxy sources is more complex than that of a target group, explaining these results. Considering all models on the 3,000-Mb-sized simulations, AEH$_T$−av. AEH$_S$ is just weakly correlated to max|EAF−EF| or *P*-values: Spearman's correlation coefficients for various *qpAdm* protocols and data qualities range from −0.49 to 0.3 in 5 practically relevant sections of the max|EAF−EF| vs *P*-value space (Supplementary Fig. 7 and Table 1c). Thus, although false/consistently nonfitting and true models differ significantly according to the AEH$_T$−av. AEH$_S$ metric, using *P*-values or EAF for predicting which model is true and which is false is not a fruitful approach, especially considering negative correlation between AEH$_T$−av. AEH$_S$ and *P*-values seen for some sections of this space and for some *qpAdm* protocols (Supplementary Table 1c).

While FP *qpAdm* models are expected for complex genetic histories, the practical usage of high-throughput *qpAdm* protocols

relies on an assumption that FPs are rare. As discussed above, due to a fundamental property of AGS histories where complexity (number of admixture events) grows with time, FDR of high-throughput *qpAdm* protocols is expected to depend on enrichment of the sets of tested models for those where proxy source history is more complex than that of the target. First, we calculated FDR on the complete set of simulated AGS histories, per each simulation/ subsampling replicate, and stratified the results by amount and quality of data, by model types, and by *qpAdm* protocols (Fig. 3).

Indeed, considering distal models only (tested with both rotating and nonrotating protocols) results in the lowest median FDR across replicates, from 16.4 to 31.2% (Fig. 3). FDRs based on proximal models tested with the rotating protocol varied from 72.5 to 100% and grew significantly with increasing amount of data. A large fraction of false proximal models emerges as fitting due to false rejections of 1-way models because of violations of the topological assumptions (Fig. 1), for instance, due to "left-to-right" gene flows, and these prompt the investigator to test more complex 2-way models, which are often fitting (Supplementary Text 2). And model rejection is more efficient when more data is available, explaining the effect on FDR observed here.

FDRs based on proximal models tested with the nonrotating protocol were lower (medians from 52.1 to 57.7%) and did not depend on the amount of data, suggesting that false model rejections due to assumption violations play a less important role here, which is expected for "right" and "left" population sets which are stratified chronologically. Similarly, the amount of data had no statistically significant influence on FDR when distal models were considered, except for the nonrotating protocol on high-quality data (increasing the amount of data reduced FDR, Fig. 3). We also note that random subsampling of SNPs and individuals and a less robust algorithm for calculating $f_4$-statistics on the low-quality data (with different statistics calculated on different SNP sets) did not lead to dramatic increases/decreases in FDR (Fig. 3). The observation that the decrease in FDR with increasing amount of data is small and often nonsignificant is possibly explained in the following way. Positive models are defined based on both *P*-values and EAF (Box 1): while EAF are relatively stable (Supplementary Fig. 9), *P*-values of individual models decrease exponentially with the amount of data (Supplementary Fig. 6). However, model rejection is not always "beneficial" for protocol's performance assessed via FDR since both models with inappropriate proxy sources (false models) and true models with violations of the topological assumptions (Fig. 1) are rejected more often when more data is added.

In summary, even in optimal conditions (simulation depth of 87,000 years on human timescale, 3,000-Mb-sized genomes, and high-quality diploid data) we did not achieve negligible FDR when distal models were tested on a collection of random AGS histories, and high data amount/quality makes FDR based on proximal models grow. In nearly all conditions tested there is very high variability in FDR across simulated AGS topologies, with many topologies demonstrating FDR of either 0% or 100% (Supplementary Fig. 11), and hence it is hard to predict if real genetic histories are prone to generating FP signals of admixture when *qpAdm* protocols are applied. For an illustration of 3 topologies most problematic for distal *qpAdm* protocols, see Supplementary Fig. 12. These differences in *qpAdm* performance across topologies are probably driven at least in part by the fact that some random AGS histories have much "denser" violations of the *qpAdm* topological assumptions than others, but this frequency of assumption violations was not quantified in our study since it requires another layer of manual topological analysis.

The fraction of 2-way admixture models that are inappropriate according to our topological criteria (the fraction of negatives) in a random sample of 400 proximal and distal rotating models from all the simulated AGS histories is 82.3% (prestudy odds = 0.215), which allows us to approximately estimate not only FDR (48–78.7%), but the FPR of the proximal rotating protocol = number of FPs/number of negatives per simulation replicate (858 models × 40 graphs × fraction of negatives) = 0.04–2.6% across simulation parameter combinations and replicates summarized in Fig. 3. The FNR of the proximal rotating protocol = number of false negatives/number of positives per simulation replicate (858 models × 40 graphs × fraction of negatives) = 89.9–99.8%. Here are the same statistics for the distal nonrotating protocol: the fraction of negatives in a random sample of 400 relevant models, 74.3% (prestudy odds = 0.346); total number of distal nonrotating 2-way models across all graph topologies, 1,804; FDR across simulation parameter combinations and replicates from Fig. 3, 10.9–34.4%; FPR, 0.2–3.2%; and FNR, 75.6–96.5%. FDR depends on FPR and FNR through equation (1), which also includes prestudy odds (*R*). Given the relatively high *R* for 2-way models based on our AGS histories and low FPR, the principal contributor to the high FDR values compromising historical interpretation of such screens for admixture is high FNR via the term $R - FNR \times R$ in the denominator of equation (1). And, high FNR results from false rejection of true 2-way models due to violations of the topological assumptions (Fig. 1) and due to high variance in *P*-values across simulation/subsampling replicates (Supplementary Figs. 8 and 9). Importantly, variance in *P*-values is especially high in the case of the most realistic simulations: noisy data derived from 3,000-Mb-sized genomes (Supplementary Fig. 9).

As we show in Supplementary Text 3 (Supplementary Fig. 13 and Tables 2 and 3), it is impossible to decrease FDR of the proximal rotating protocol dramatically by combining it with other methods (PCA and unsupervised *ADMIXTURE*) or by adjusting *P*-value thresholds. However, it is possible to decrease FDR of distal *qpAdm* protocols to ca. 0% if positive *qpAdm* results are further validated by PCA and the unsupervised *ADMIXTURE* (Alexander *et al.* 2009) methods (Supplementary Table 4 and Text 4). In that supplemental section we also explore the performance of proximal nonrotating *qpAdm* protocols combined with model competition (Narasimhan *et al.* 2019) (Supplementary Table 5 and Text 4).

## Introducing SSL histories

Since effects of prolonged gene flows in the context of a simple AGS history were studied by Harney *et al.* (2021), here we explore a more extreme case investigated in that study briefly, namely SSLs (Box 1). We simulated SSLs that are more realistic than uniform gene flow scenarios and/or 1D strings of demes (Novembre and Stephens 2008; Duforet-Frebourg and Slatkin 2016; Harney *et al.* 2021; Tournebize and Chikhi 2025; Speidel *et al.* 2025), namely 2D landscapes with gene flow intensities that are nonuniform both in space and time (see similar stepping stone grids in, for example, the studies by Petkova *et al.* 2016 and Al-Asadi *et al.* 2019). Approximately circular landscapes composed of 64 panmictic demes of constant effective size (1,000 diploid individuals) were simulated as arising via multifurcation ("star radiation") and then evolving for ca. 2,500 generations with nonuniform bidirectional gene flows (Fig. 4).

Four types of SSL were simulated, characterized by varying ranges of gene flow intensities, with 10 simulation/gene flow intensity replicates per range: (ca. $10^{-5}$ to $10^{-2}$), (ca. $10^{-5}$ to $10^{-4}$), (ca. $10^{-4}$ to $10^{-3}$), and (ca. $10^{-3}$ to $10^{-2}$) (Supplementary Fig. 14b

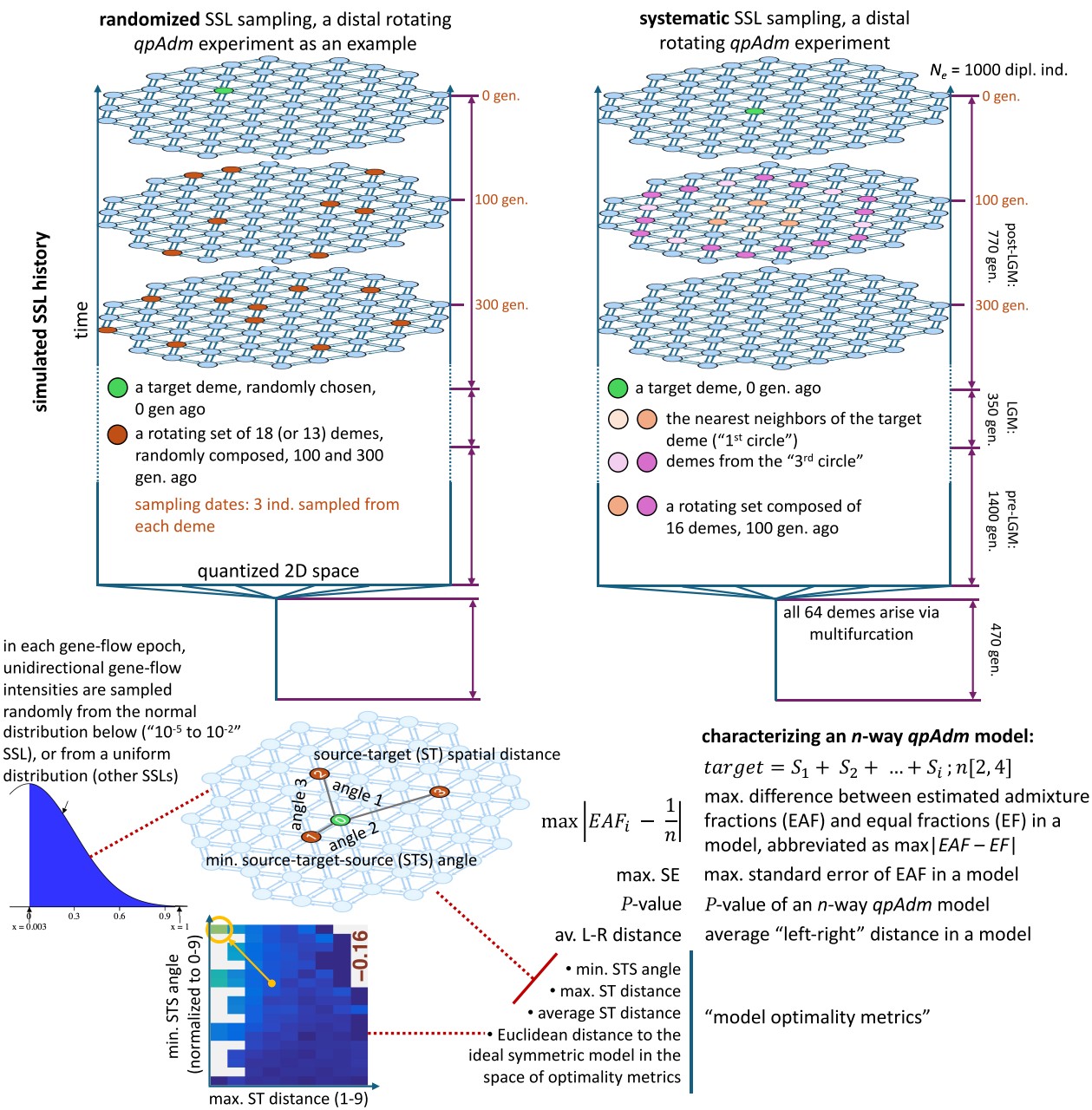

**Fig. 4.** Diagrams summarizing principles of SSL simulations, SSL-based *qpAdm* setups, and approaches we used for interpreting *qpAdm* results in this context. Key features of the simulations are as follows: (1) origin of all the demes via a multifurcation; (2) the 2D space is quantized into 64 demes of small and stable effective population size; (3) gene flow intensities between neighboring demes are generated by random sampling from normal or uniform distributions (in certain intervals), at the beginning of "gene flow epochs"; (4) at least 3 gene flow epochs were simulated ("pre-LGM," "LGM," and "post-LGM"); and (5) all the demes are sampled at 3 time points ("slices" of the landscape) in the 2nd half of the "post-LGM" epoch. *qpAdm* experiments (rotating and nonrotating, distal, and proximal) were constructed by subsampling this initial set of deme samples randomly or systematically, as illustrated by the examples on the left and the right, respectively. Each *qpAdm* model (2- to 6-way) was then characterized by 9 numbers: 3 of those were derived from *qpAdm* outputs, and 6 were properties of the model in the context of the landscape. The former metrics are as follows: max|EAF−EF|, max. SE, and *P*-value. Five of the latter numbers are termed "model optimality metrics": min. STS angle, max. ST distance, average ST distance, Euclidean distance to the ideal symmetric model (marked by the orange circle) in the space of model optimality metrics (illustrated in the lower left corner of the figure) based on min. STS angle and max. or average ST distance. Yet another metric is defined on the landscape but is not included in the optimality metrics, that is average distance from demes in the "left" set to demes in the "right" set (average "left–right" distance).

and c and Table 6). Each intensity range corresponds to gene flow between 2 neighboring demes in 1 direction per generation. An intensity value means that a given fraction of effective population size in a generation comes from migrants from a specified neighboring deme. In all the cases, gene flow intensities were sampled at the start of each "gene flow epoch" from uniform distributions, except for the ca. $10^{-5}$ to $10^{-2}$ range: in that case, intensities were

sampled from a normal distribution centered at 0 (Supplementary Table 6).

In the case of *f*-statistics, simulating $n \times m$ demes and sampling $m$ of them evenly is equivalent to simulating $m$ demes, but with proportionally lower gene flow intensities. Since the size of the landscapes (64 demes) was the same across the gene flow intensity brackets, these landscape types, judging by the $F_{ST}$

distributions (Supplementary Fig. 14a), can be thought of as representing regions of different size: from very sparse global sampling ($10^{-5}$ to $10^{-4}$, median $F_{ST} = 0.41$, which is close to the highest $F_{ST}$ between African and non-African humans) to dense sampling in a subcontinental region ($10^{-3}$ to $10^{-2}$, median $F_{ST} = 0.02$, corresponding, e.g. to the Bronze Age West Asian populations analyzed by Lazaridis *et al.* 2016). Thus, by varying gene flow intensity (Supplementary Fig. 14b and c), we explored different levels of "landscape sparsity" (Box 2): a landscape property that, as we show below, is probably the single most important predictor of *qpAdm* performance on SSL in the absence of rapid long-distance migrations (direct gene flows between nonneighboring demes). The "$10^{-5}$ to $10^{-2}$" type of landscapes, enriched not only for low-intensity gene flows due to sampling from a normal distribution but also including high-intensity flows (Supplementary Fig. 14b and c), is probably closer to real-world *qpAdm* protocols (Supplementary Text 1) where worldwide "right" groups are routinely combined with groups from a small region of interest where intensive admixture is hypothesized. Median $F_{ST}$ for this landscape type is 0.06 (Supplementary Fig. 14a), which is typical for the commonly studied human groups from the Stone and Bronze Age Europe ($F_{ST}$ between 0.05 and 0.09, Speidel *et al.* 2025).

To keep the simulations computationally tractable, the genome size was restricted to 500 Mb with mutation and recombination rates that are realistic for humans (see *Methods*). See Supplementary Figs. 14a and 15 for $F_{ST}$ distributions and counts of polymorphic sites in these simulations, respectively. In the following sections, we explore effects on performance of *qpAdm* protocols of the following variables: (1) landscape sparsity; (2) randomized vs systematic sampling of demes for *qpAdm* experiments (see examples in Fig. 4); (3) the 4 basic types of high-throughput *qpAdm* admixture screens described above (distal and proximal, rotating, and nonrotating, summarized in Supplementary Fig. 16); and (4) and a wide range of composite *qpAdm* model feasibility criteria (see *Methods*).

## SSL histories: randomized sampling of landscapes

Our motivation for focusing on randomized sampling of landscapes for subsequent *qpAdm* analysis is informed by the archaeological practice. In most regions and periods, a substantial fraction of ancient human population remains archaeologically (and thus archaeogenetically) invisible due to various reasons: (1) burial rites that leave no suitable remains such as cremation (Cerezo-Román *et al.* 2017), tree burials (Lindig 1964), or burial on or above ground (Pavlinskaya 2009); (2) poor DNA preservation conditions such as acidic soil or hot climate (Baxter 2004); (3) biases in archaeological research that is often driven by salvage excavations in industrial/agricultural landscapes, but leaving other landscapes out, or by political agendas; and (4) lack of exhaustive and systematic sampling of even the existing anthropological collections. Thus, even a completely random sampling of a genetic landscape, used for many analyses in this study, may be viewed as a reasonable conservative baseline for studying method performance.

As stated above, we explored 4 *qpAdm* protocols (Supplementary Fig. 16). For example, in the case of the distal rotating protocol, for a target deme randomly picked from the "present" sampling point, we sampled 13 or 18 demes randomly from the 2 older sampling points combined (100 and 300 generations before present), and they formed a rotating set of proxy sources and "right" groups (Fig. 4). All possible 1- to 4-way *qpAdm* models on this subset of 13 or 18 demes were tested, and this collection of models for one target we termed "a *qpAdm* experiment" (Box 1). The counts of alternative proxy sources per target (10 or 15 for the nonrotating and 13

or 18 for rotating protocols) and the model complexities explored match those in typical high-throughput *qpAdm* protocols from the literature (e.g. Narasimhan *et al.* 2019; Bergström *et al.* 2022; Lazaridis *et al.* 2022; Zeng *et al.* 2023; Speidel *et al.* 2025). In general, the protocols in this study were designed to reproduce prestudy odds typical for these published protocols, and we explore this issue more directly in the section *Geographical analysis of large sets of qpAdm models on real data*. Below we describe our analysis of *qpAdm* performance (based on 12,800 experiments and 27,511,200 individual *qpAdm* models) and propose guidelines for interpreting *qpAdm* results on randomly sampled SSL, both at the level of models and experiments (the FDR results on the AGS histories above are based on models, not experiments, but experiments were considered in Supplementary Text 2).

We characterized performance of high-throughput *qpAdm* protocols on SSL in 2 ways. A threshold separating optimal and nonoptimal models allows FDR and other terms from equation (1) to be calculated, but a drawback is that the estimates are threshold dependent. To avoid this artificial classification of models, we first used an indirect approach and ranked models by various "optimality metrics" based on (maximal or average) spatial distances between the target and sources (maximal or average ST distance) and on angles formed by pairs of sources and the target (minimal STS angle; Fig. 4; Box 2). Spatial distance is a noisy approximation for genetic distance, but since (1) gene flows between nonneighboring demes were not allowed, (2) gene flow intensities on the landscapes (in the last epoch including our sampling points) varied in relatively narrow ranges for all the SSL types except for "$10^{-5}$ to $10^{-2}$" (Supplementary Fig. 14c and Table 6), and (3) our main analyses were randomized and replicated (gene flow intensities were resampled randomly for each simulation replicate and sets of demes for *qpAdm* experiments were drawn randomly), relationships between genetic ($F_{ST}$) and spatial distances are strong, approximately linear (Pearson's *r* from 0.65 to 0.76), and remain identical across *qpAdm* protocols on the same SSL type and very similar across the SSL types, apart from "$10^{-5}$ to $10^{-4}$" (Supplementary Fig. 14d and e). Thus, differences in *qpAdm* performance which we are most interested in, namely those across *qpAdm* protocols, gene flow intensity ranges, and model complexities, depend on factors other than the shape of the genetic–spatial distance relationship. Moreover, genetic distance measures such as $F_{ST}$ and haplotype-sharing statistics on SSL depend on gene flows in both directions, but *qpAdm* results are traditionally interpreted in terms of source-to-target gene flows only.

## *qpAdm* performance at the level of models: model optimality metrics in spaces of EAFs and *P*-values

Our exploration of *qpAdm* FDR on AGS histories above and nearly all *qpAdm* results in the literature rely on fixed *P*-value thresholds, usually set at 0.01 or 0.05 (Skoglund *et al.* 2017; Harney *et al.* 2018; Narasimhan *et al.* 2019; Olalde *et al.* 2019; Yang *et al.* 2020; Carlhoff *et al.* 2021; Fernandes *et al.* 2021; Librado *et al.* 2021; Zhang *et al.* 2021; Bergström *et al.* 2022; Lazaridis *et al.* 2022; Oliveira *et al.* 2022; Nakatsuka *et al.* 2023; Taylor *et al.* 2023; Zeng *et al.* 2023; Kerdoncuff *et al.* 2024; Speidel *et al.* 2025). Implicit usage of higher "fuzzy" *P*-value thresholds is also common in the literature when only models with the highest *P*-values are interpreted from historical perspective (Narasimhan *et al.* 2019; Fernandes *et al.* 2020; Bergström *et al.* 2022; Lazaridis *et al.* 2022; Brielle *et al.* 2023; Carlhoff *et al.* 2023; Speidel *et al.* 2025). Another part of the composite feasibility criteria, namely conditions on EAFs, are also used very often (see *Introduction* for a brief overview of the literature). With these considerations in mind, we first inspect results

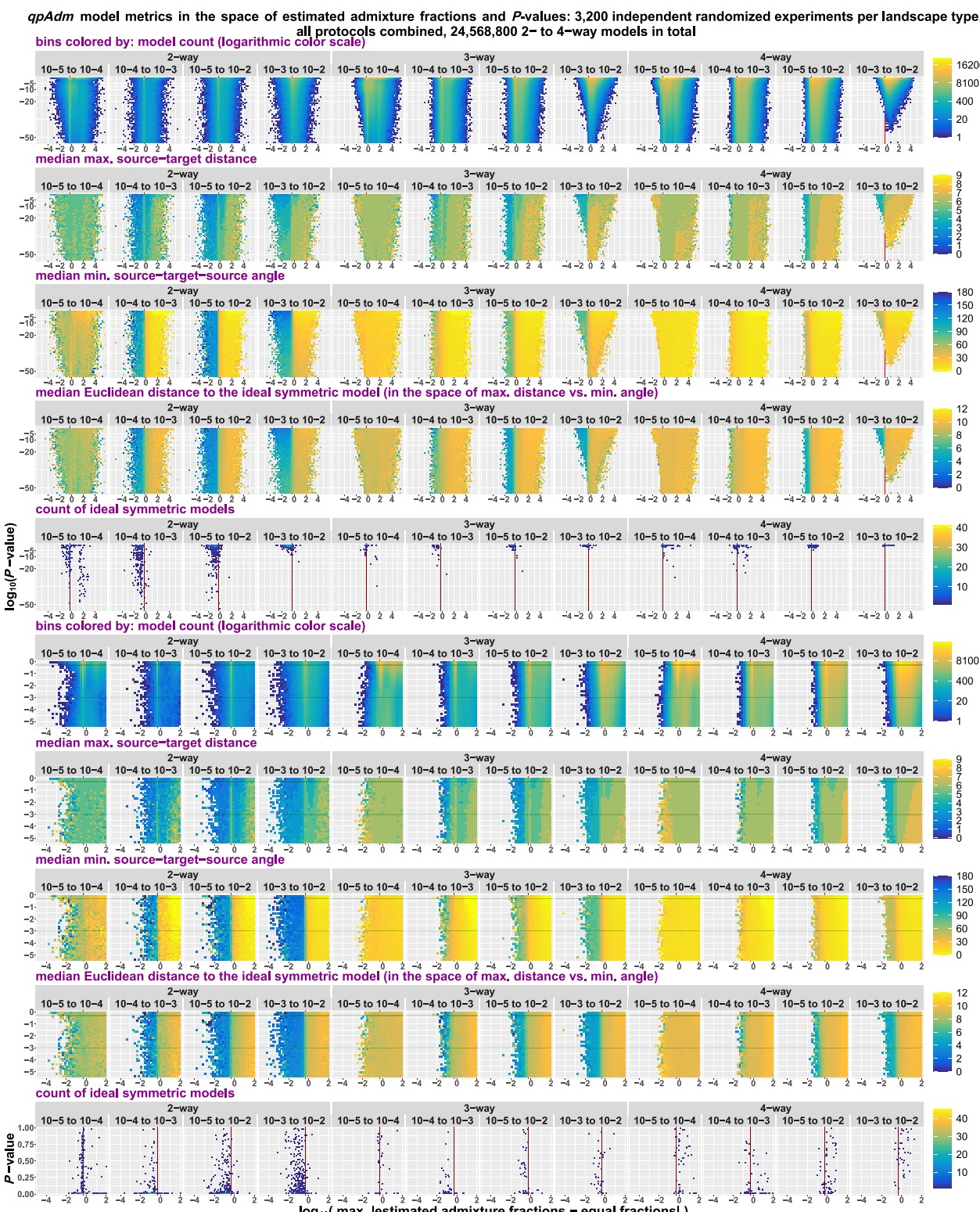

**Fig. 5.** Randomized 2 to 4-way *qpAdm* models in the space of max|EAF−EF| and *P*-values (both axes are logarithmic). Results for all *qpAdm* protocols are combined but stratified by model complexity and landscape type. Two sections of the space are shown: *P*-values from $10^{-55}$ to 1 (the upper 5 rows of plots) and from $3.16 \times 10^{-6}$ to 1 (the 4 rows below them). The space is divided into rectangular bins ($50 \times 50$ for the larger section and $35 \times 35$ for the smaller section), and they are colored by density of individual *qpAdm* models populating this space (logarithmic color scale) or by median values of model optimality metrics in those bins: max. ST distance, min. STS angle, and Euclidean distance to the ideal symmetric model. Density of ideal symmetric models (in the case of 4-way models, the most optimal nonideal models available on our SSL) in this space is also shown (in $75 \times 75$ bins). To assess if distributions of ideal symmetric models over *P*-values are uniform, we use a nonlogarithmic *P*-value scale (the bottom row of plots). The vertical lines (or the tick marks) mark max|EAF−EF| = 0.5, and the horizontal lines mark the lowest and highest *P*-value thresholds used in this study (0.001 and 0.5).

(all *qpAdm* protocols pooled) for the 4 landscape types in the coordinates of max|EAF−EF| and *P*-values (Fig. 5; Supplementary Fig. 17). On these 2D spaces, we mapped median values of the "optimality metrics" characterizing *qpAdm* models: maximal (Fig. 5) or average (Supplementary Fig. 17) ST distance in a model and minimal STS angle (Fig. 5; Supplementary Fig. 17).

It is immediately obvious that global genetic landscapes sampled very sparsely (approximated by SSL with gene flow intensities per generation from ca. $10^{-5}$ to $10^{-4}$) are unsuitable for *qpAdm* analyses since the bins with low *P*-values and EAF between 0 and 1 are not enriched in optimal models (those with small maximal ST distances and/or symmetrically arranged sources) (Fig. 5) and absolute Spearman's rank correlation coefficient ($\rho$) for max|EAF−EF| or *P*-value vs any optimality metric does not exceed 0.1 (if max|EAF−EF| ≤0.5 and *P*-value ≥$10^{-5}$; see Supplementary Table 7). Although this high level of sparsity is probably rare in practice, it may approach the current sampling situation for Paleolithic humans (Posth *et al.* 2023; Zeng *et al.* 2023; Allentoft *et al.* 2024) and for many species other than humans. In contrast, for the other landscape sparsity levels we simulated there are strong correlations between max|EAF−EF| and the model optimality metrics, for some model complexities at least (Fig. 5; Supplementary Table 7; see also correlation coefficients stratified by *qpAdm* protocols in Supplementary Table 8). For instance, in the case of 2-way models there is a sharp boundary between models with EAF between 0 and 1 and those outside of this interval (the latter corresponding to max|EAF−EF| >0.5): the area to the left of this boundary is enriched in symmetric models, while the area to the right of it is enriched in asymmetric models, and this is true for all 3 denser SSLs (Fig. 5). This corresponds to significant Spearman's $\rho$ between −0.65 and −0.81 for max|EAF−EF| vs minimal STS angle for 2-way models on the "$10^{-4}$ to $10^{-3}$", "$10^{-5}$ to $10^{-2}$", and "$10^{-3}$ to $10^{-2}$" landscape types (*P*-values ≥$10^{-5}$ or ≥$10^{-55}$; Supplementary Table 7).

This result shows that the method usually fails to report EAF in the interval (0, 1) for models with asymmetrically arranged proxy sources. There is a similar, albeit more complex, relationship of max|EAF−EF| with maximal (Fig. 5) or average (Supplementary Fig. 17) ST distances. These relationships of max|EAF−EF| with the angle- and distance-based optimality metrics gradually disappear with growing model complexity and/or landscape sparsity (Fig. 5; Supplementary Fig. 17) as illustrated by Spearman's $\rho$ (Supplementary Table 7). Thus, a limit to model complexity that makes sense to explore with high-throughput *qpAdm* on SSL is rather low in most of our simulations: just 2 or 3 sources (see more on this topic below in the paragraphs focused on FDR).

In the dimension of *P*-values, there also exists a relationship with the model optimality metrics, but it is much weaker (Fig. 5; Supplementary Fig. 17). For example, in the practically relevant range of *P*-values ($10^{-5}$ to 1) and for max|EAF−EF| ≤0.5, *P*-value is not correlated with maximal ST distance and with minimal STS angle (Supplementary Table 7) on the "$10^{-4}$ to $10^{-3}$", "$10^{-5}$ to $10^{-2}$", and "$10^{-3}$ to $10^{-2}$" landscapes (Fig. 5). There is weak correlation with average ST distance in the same part of the space (significant Spearman's $\rho$ up to −0.27, Supplementary Table 7). This relationship also decays with growing model complexity and landscape sparsity (Supplementary Table 7).

We observe that our randomized *qpAdm* protocols favor models including target's nearest neighbors that are, moreover, arranged symmetrically around the target, so-called "ideal symmetric models" (Box 2), and our analyses in the next section further support that statement (Supplementary Fig. 18a). Considering this observation, it is convenient to merge the angle- and distance-based

optimality metrics into a single metric, that is Euclidean distance in the space of maximal (or average) ST distances and min. STS angles between a given admixture model and the ideal symmetric model of the same complexity (Fig. 4; Box 2). Considering *P*-values in a wide range ($10^{-55}$ to 1) and max|EAF−EF| ≤0.5, there is no correlation between *P*-values and these 2 composite optimality metrics in the case of 3- and 4-way models (the most negative $\rho$ is −0.04; Supplementary Table 7), and weak correlation in the case of 2-way models, but on the "$10^{-3}$ to $10^{-2}$" landscapes only (significant $\rho$ = −0.39 and −0.48; Fig. 5; Supplementary Fig. 17 and Table 7). While conditions on EAF do enrich the set of retained randomized models for symmetric ones, *P*-value thresholds used in practice (0.001, 0.01, 0.05, and even higher implicit thresholds) discard fractions of randomized ideal symmetric models that are much greater than these thresholds (Supplementary Table 9): e.g. the 0.05 threshold discards 44.7% of 3-way ideal models on the "$10^{-5}$ to $10^{-2}$" landscapes. The only condition where the distributions approach uniformity is the "$10^{-3}$ to $10^{-2}$" landscapes and 3- and 4-way models (Fig. 5; Supplementary Fig. 17 and Table 9).

Thus, the distributions of ideal randomized models on the *P*-value axis become uniform with decreasing landscape sparsity and increasing model complexity, which is an expected behavior since none of the randomized models we tested are truly ideal considering the topology of the landscape with an immediate ancestor and up to 6 neighboring demes for each target deme (Fig. 4, see however an exploration of 6-way models in the systematic SSL sampling context below). Another reason for the nonuniformity of the *P*-value distributions are complex networks of gene flows that permeate the landscape and lead to low *P*-values of *qpAdm* models due to violations of the assumptions on the "left"–"right" gene flows (Fig. 1) that were derived originally on AGS histories (Haak *et al.* 2015; Harney *et al.* 2021), but not on SSL. In summary, in any *qpAdm* screening relying on stringent *P*-value thresholds such as 0.01 there probably exists a large class of 2- to 4-way models that have nonnegative admixture proportion estimates and are relatively close to optimal ones (Fig. 5) but are rejected according to *P*-values. We explore performance of *qpAdm* protocols based on composite feasibility criteria in the following sections.

We also analyze spaces of EAF SEs reported by *qpAdm* vs *P*-values (Supplementary Figs. 19 and 20 and Tables 7 and 8) and, relying on similar approaches, explore randomized 1-way models (Supplementary Fig. 21 and Table 10) in Supplementary Text 5.

## *qpAdm* performance at the level of models: feasible models in spaces of model optimality metrics

In the previous section, we undertook a very general overview of *qpAdm* results on SSL and reached some preliminary conclusions on performance of high-throughput *qpAdm* protocols used in practice for finding optimal models for target groups or individuals. Here, we apply the thresholds on *P*-values and EAF explicitly and explore a wide range of composite feasibility criteria, looking at distributions of models passing these criteria in 2D spaces formed by the optimality metrics (Fig. 4). As explained in Box 1, composite feasibility criteria may include not only conditions on EAF and *P*-value thresholds but also a condition that all "trailing" simpler models are rejected according to *P*-values ("trailing" models should not be confused with all simpler models tested for a given target in an experiment, see Box 1). As another condition (rarely used in practice, to our knowledge), we tested a *P*-value threshold for rejection of "trailing" models that differs from the "main" *P*-value threshold. In total, 36 composite feasibility criteria were tested, with

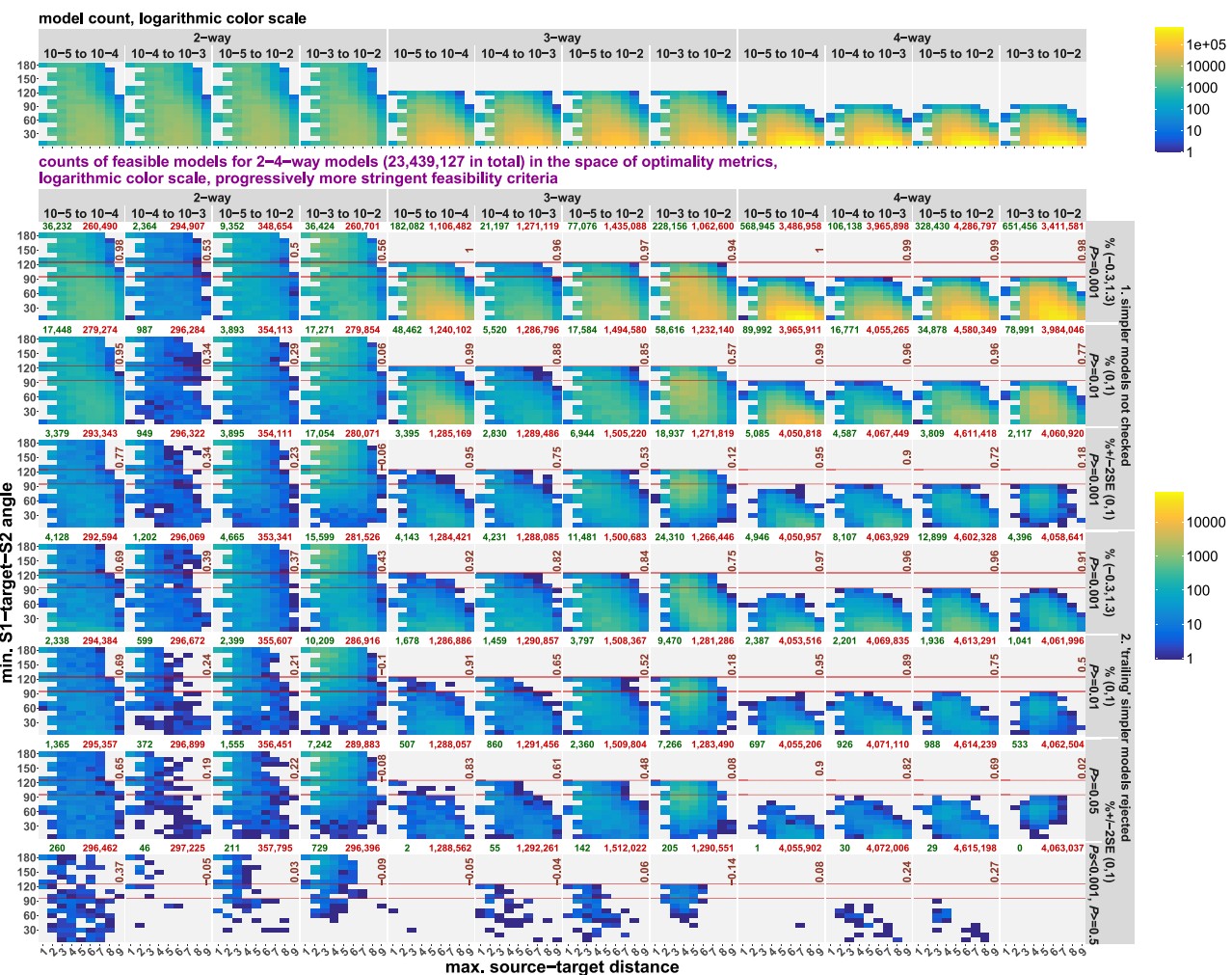

**Fig. 6.** Randomized 2- to 4-way *qpAdm* models in the space of model optimality metrics: max. ST distance and min. STS angle (models with the latter metric undefined were excluded from this analysis). Results for all *qpAdm* protocols are combined but stratified by model complexity and landscape type. The upper row of plots shows density of all randomized models tested, and the other rows show density of fitting models satisfying selected composite feasibility criteria listed in the captions on the right (7 of 36 criteria tested in this study, from the least stringent on top to the most stringent at the bottom). All the color scales are logarithmic. The space was divided into 9 bins on the x-axis (bin width = 1), and 18 bins on the y-axis (intervals [0°, 10°], (10°, 20°), and so forth). Spearman's correlation coefficients for counts of all models vs fitting models in these bins are shown in each panel in brown (bins not populated by any models were not considered). The horizontal lines mark the "ideal" min. STS angles for 3- and 4-way models (120° and 90°). The numbers above each plot on the right and on the left stand for counts of rejected and fitting models in those analyses, respectively.

P-value thresholds ranging from 0.001 to 0.5, which covers the whole range of thresholds used in the literature explicitly or implicitly, and with 3 types of conditions on EAF (see Supplementary Table 11 and *Methods* for a complete list). Results for 7 selected criteria are presented in the figures cited below. Since this analysis was performed on combinations of source demes drawn randomly from SSL, the spaces of optimality metrics are highly unevenly populated by models (Fig. 6; see the spaces based on average ST distance in Supplementary Fig. 22).

First, we explore the positive rate of *qpAdm* (Box 1), dividing the spaces of optimality metrics into 2D bins: the positive rate is simply a fraction of all models tested (within a bin) passing a feasibility criterion (Supplementary Figs. 18a, b, and d and 22a, c, and e). Under stringent criteria (such as EAF ± 2 SE in the interval (0, 1), P-value ≥ 0.05, "trailing" models rejected), positive rates approach 0 in most parts of the optimality metric space except for its upper left corner where ideal symmetric models lie (Fig. 4). This is true for most levels of landscape sparsity and for most model complexities we tested (Supplementary Figs. 18a and 22a). These results are based on all the *qpAdm* protocols combined, and we show results for

rotating and nonrotating protocols separately in Supplementary Figs. 18b and d and 22c and e. For assessing practical significance of *qpAdm*'s "preference" for nearly ideal symmetric models it is important to look at the system not only through the lens of positive rates (Supplementary Fig. 18), but also at density of fitting models in the space of optimality metrics (Fig. 6; Supplementary Figs. 18c and e, and 22b, d, and f). We use Spearman's ρ based on model counts in bins to compare 2D distributions (in the optimality metric space) of fitting models and of all models prior to *qpAdm* testing (Supplementary Table 11). Those highly nonuniform distributions of fitting models (Fig. 6) are "cocreated" by outcomes of *qpAdm* protocols (their FPR and FNR) and by randomized construction of proxy source combinations on a 2D grid, and we consider this approach as a conservative approximation for high-throughput *qpAdm* protocols on real-world archaeogenetic sampling (see a demonstration of this in the section *Geographical analysis of large sets of qpAdm models on real data*). In other words, random landscape sampling creates "sets of all possible admixture models for a target" where relatively symmetric models composed of target's neighbors constitute a small minority, and where highly

asymmetric models with distant sources are by far the most abundant (Fig. 6). See also alternative visualizations (Supplementary Fig. 23) combining the approaches from Figs. 5 and 6 and illustrating the same points.

In the cases of 4- and 3-way models, only very specific landscape properties and feasibility criteria led to depletion of the sets of fitting models for nonoptimal models (those with small min. STS angles and at least one distant source), which is evidenced by Spearman's $\rho$ (for total model counts vs fitting model counts in 2D bins) not significantly different from 0 under these conditions, but high and significant under nearly all the other conditions tested (Supplementary Table 11). These conditions are as follows: the densest landscape tested ("$10^{-3}$ to $10^{-2}$") and EAF $\pm$ 2 SE $\in$ (0, 1), with no significant influence of the $P$-value threshold on Spearman's $\rho$. These results are in line with our observations in the preceding section (Fig. 5), namely that EAF (and their SE correlated with them) and landscape sparsity, in contrast to model $P$-values (which are highly variable due to gene flows in all directions on the landscapes and due to average distances between "left" and "right" deme sets fluctuating across models, as discussed in the section *qpAdm performance at the level of experiments*), are correlated to both angle- and distance-based model optimality metrics strongly. We do not highlight other acceptable combinations of landscape sparsity and feasibility criteria such as "$10^{-3}$ to $10^{-2}$" and EAF $\in$ (0, 1) (Supplementary Table 11) since they are applicable to a narrower range of model complexities (do not perform well for 4-way models, for example) or yield very low overall positive rates.

Introducing very stringent feasibility criteria [such as $P$-value $\geq$0.5, "trailing" models rejected with $P < 0.001$, EAF $\pm$ 2 SE $\in$ (0, 1)] is probably impractical since it drives the overall *qpAdm* positivity rate to extremely low values, especially in the case of complex models: on the order of $10^{-4}$ to $10^{-6}$ for 3-way models and $10^{-6}$ to $10^{-7}$ and even lower for 4-way models (Fig. 6; Supplementary Table 11). As we discuss below in the section *Systematic sampling: exploring truly ideal 6-way models*, despite $P$-values growing with increasing model complexity, fraction of models with EAF $\in$ (0, 1) falls sharply due to increasing uncertainty in EAF estimates and that is responsible for the low positive rates based on composite feasibility criteria. For practical purposes, to keep overall *qpAdm* positive rate reasonably high (ca. $10^{-2}$ to $10^{-4}$ depending on model complexity, see Supplementary Table 11), we recommend using the least stringent feasibility criteria satisfying the conditions described above: restricting *qpAdm* sampling (in both "right" and "left" sets) to subcontinental regions such as West Eurasia in the case of humans (which corresponds approximately to the "$10^{-3}$ to $10^{-2}$" landscapes); stringent conditions on EAF such as EAF $\pm$ 2 SE $\in$ (0, 1); a low $P$-value threshold for "fitting" models (such as 0.001); and no $P$-value thresholds for rejecting simpler "trailing" models. We note that, following this approach, it is almost impossible to find a single best model complexity level for a target and a single best model at that complexity level, but that is difficult to achieve in principle on SSL as we discuss in the next section.

Inspecting the spaces formed by average ST distance and min. STS angle, we observe the same trends (Supplementary Fig. 22b, d and f). Considering the radius and shape of the SSL studied here (Fig. 4), the triangular region in the lower right corner of the optimality metric space (Supplementary Fig. 22b, d and f) is populated by models where a target is located close to (or on) one edge of the landscape and at least some proxy sources are located close to the opposite edge of the landscape (average ST distance >4.5, which is the radius of the landscape). Complete elimination of those highly asymmetric models from sets of fitting models is more achievable in the case of 2-way models, but is achievable for 3- and especially 4-way models only on the densest landscape "$10^{-3}$ to $10^{-2}$" and/or when the most stringent conditions on EAF are applied (Supplementary Fig. 22b and Table 11).

Admixture models on SSL lie on an "optimality continuum," but for illustrating our conclusions above in a more quantitative way it makes sense to introduce a threshold in the space of optimality metrics (such as that in Fig. 6) for classifying models into optimal and nonoptimal ones, which enables us to compute prestudy odds, FPR, FNR, and FDR. The distance threshold we chose (Box 2) corresponds to a difference in maximal ST distance (vs the ideal model) up to 5 and/or to differences in min. STS angle up to 100°, 66.7°, and 50° (for 2-, 3-, and 4-way models, respectively). Even considering this relaxed threshold, the SSL-based simulations designed to resemble typical high-throughput *qpAdm* protocols from the literature result in low prestudy odds: ca. 0.18, 0.04, and 0.015 for 2-, 3-, and 4-way models, respectively (Fig. 7). In this figure, we also show FPR, FNR, and FDR distributions across simulation replicates grouped by SSL sparsity and model complexity. See more fine-grained results stratified additionally by the number of alternative sources for a target and by *qpAdm* protocols in Supplementary Figs. 24–26 (there results are shown for all the composite feasibility criteria from Fig. 6 except for the most stringent one).

In Fig. 7 we compare results for 2 composite feasibility criteria of different stringency: EAF $\in$ (0, 1), $P$-value $\geq$0.01, simpler models not checked, and EAF $\pm$ 2 SE $\in$ (0, 1), $P$-value $\geq$0.05, "trailing" simpler models rejected. As expected, increasing the stringency changes FPR from 0.2–5.4% to 0.01–0.5% (these are FPR values from Fig. 7, averaged across simulation replicates), and these differences are statistically significant. There is the opposite trend in FNR: it grows from 62.9–98.7% to 86.9–99.9% with increasing feasibility criteria stringency (Fig. 7). Due to low prestudy odds and very high FNR, the improvement in FDR with increasing feasibility criteria stringency is much less striking than the improvement in FPR, although most differences are statistically significant (Fig. 7): for example, for models of varying complexity on the "$10^{-3}$ to $10^{-2}$" landscapes FDR falls from 31.5–81.8% to 18.4–40.9%. These results also illustrate in another way the trends discussed in this section above, namely the significant differences in *qpAdm* performance between model complexities and some SSL sparsity levels (Fig. 7). Since prestudy odds fall rapidly with increasing model complexity, to keep FDR for 3-way and more complex models below 50%, both FPR and FNR must remain relatively low for these model classes [equation (1)], but this is achievable only in the case of densely sampled landscapes and stringent conditions on EAF [EAF $\pm$ 2 SE $\in$ (0, 1)]. The power of *qpAdm* model screens (1−FNR) grows with SSL density, especially between the "$10^{-5}$ to $10^{-2}$" and "$10^{-3}$ to $10^{-2}$" landscape types (Fig. 7), and FPR is improved by orders of magnitude with the introduction of more stringent conditions on EAF. Compare, for example, the following criteria: EAF $\in$ (−0.3, 1.3), $P$-value $\geq$0.001, simpler models not checked and EAF $\pm$ 2 SE $\in$ (0, 1), $P$-value $\geq$0.001, simpler models not checked (Supplementary Fig. 24). Although different *qpAdm* protocols yield significantly different FPR and FNR in some cases (for some SSL types and model complexities), changes in these variables mostly cancel out, and there are no statistically significant differences in FDR between *qpAdm* protocols and different sizes of proxy source or rotating sets (10 or 15, 13 or 18 demes) (Supplementary Fig. 24a–c).

Importantly, high $P$-value thresholds often used in the literature explicitly or implicitly ($\geq$0.05) are counterproductive in the SSL context since they do not improve FDR significantly

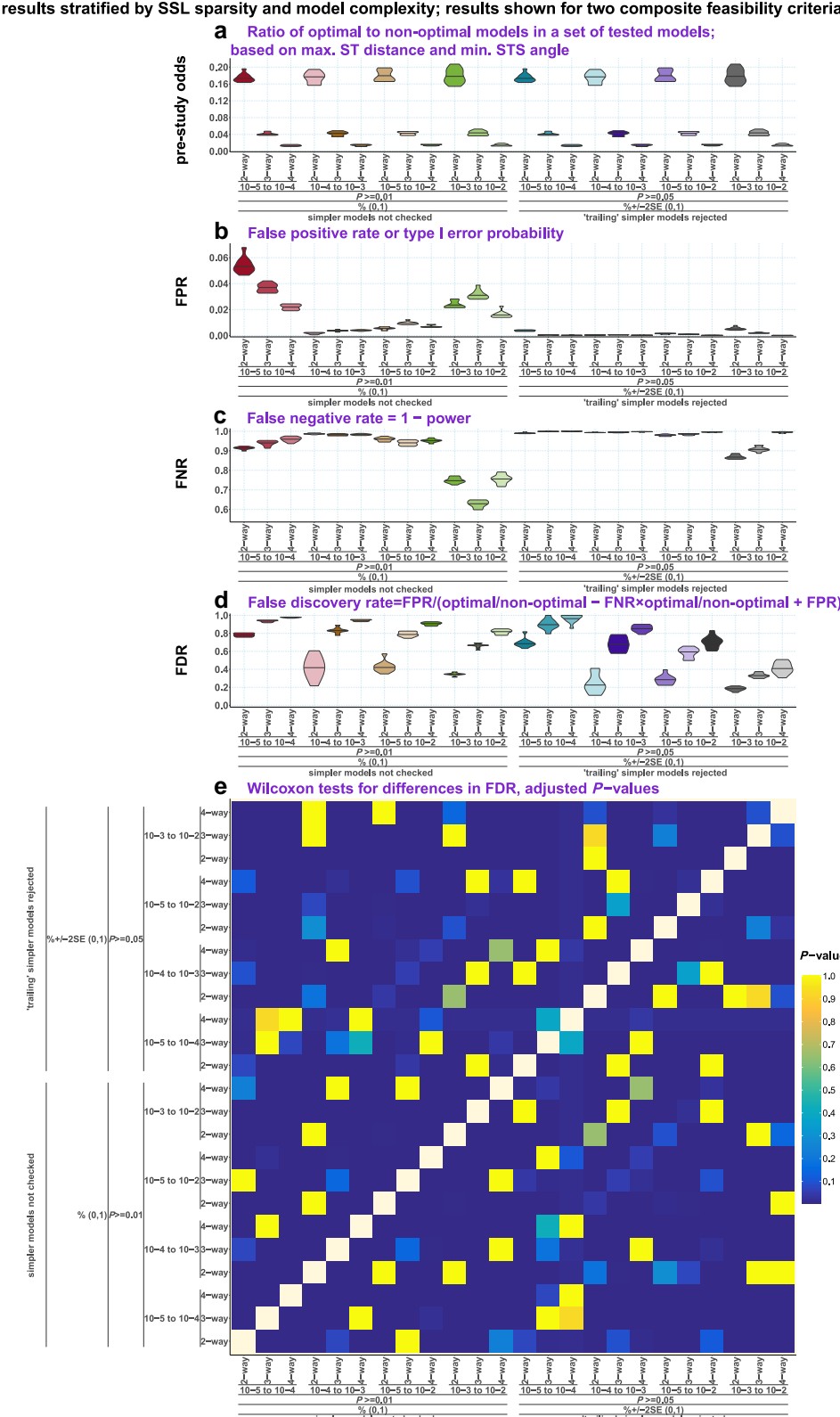

**Fig. 7.** Prestudy odds a), FPR b), FNR c), and FDR d) are shown for 2 composite model feasibility criteria. The violin plots with medians visualize distributions of these metrics across simulation replicates. The results are grouped by SSL sparsity and model complexity (all *qpAdm* protocols are considered together). Results of pairwise comparisons of the FDR distributions are shown in the matrix form in e) (P-values of nonpaired Wilcoxon tests were adjusted for multiple testing using the Holm method).

(Supplementary Fig. 24) but result in very low overall positivity rates (Supplementary Table 11). We note that due to the widespread usage of composite model feasibility criteria in the literature and in this study and since ideal models on SSL may be much more complex than those tested in practice (see *Systematic sampling: exploring truly ideal 6-way models*), P-value

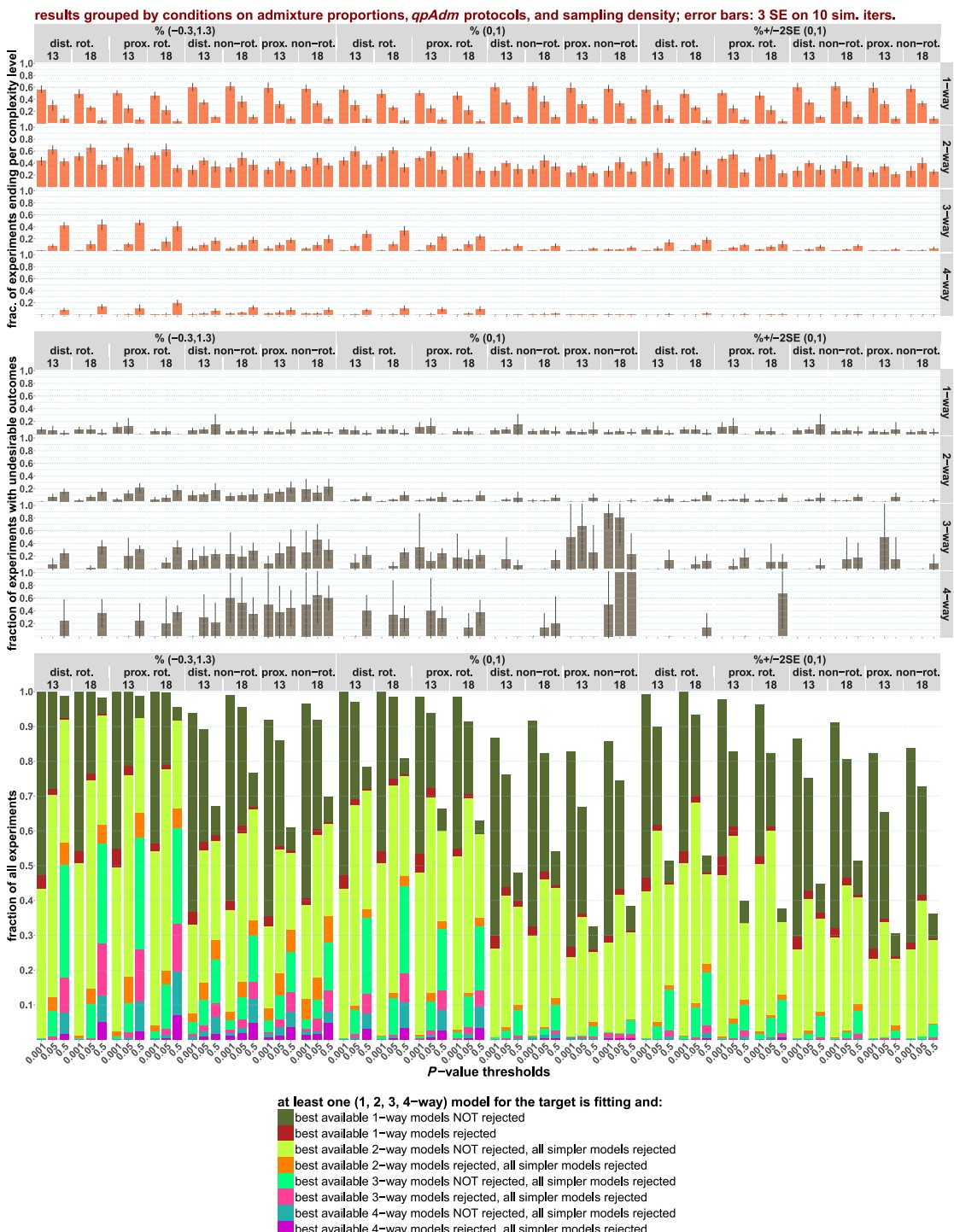

**Fig. 8.** Randomized *qpAdm* results at the level of experiments visualized for "dense" landscapes ($10^{-3}$ to $10^{-2}$). The results are stratified by *qpAdm* protocols and landscape sampling density (13 or 18 demes in rotating sets and 10 or 15 demes in nonrotating "right" and proxy source sets, respectively) and also by model complexity level at which experiments generate positive results, by conditions on EAF, and by P-value thresholds (results for 3 thresholds only are shown: 0.001, 0.05, and 0.5). We show as bar plots fractions of experiments producing positive results (at least one fitting model per target) at each complexity level and fractions of those experiments with positive outcomes classified as potentially misleading (see *Methods* for a definition based on Euclidean distances in the space of optimality metrics: max. ST distance and min. STS angle). In the case of "bad" experiments, the most optimal model (according to ST distances and STS angles) available in the chosen deme set was rejected, but a much less optimal model emerged as fitting. The error bars show 3 SE intervals calculated on 10 simulation replicates. Fractions of experiments falling into these different classes (producing positive results at complexity levels from 1 to 4 and classified as either misleading or not misleading) are visualized by the stacked bar plots. Error bars are not shown for visual clarity in the latter plot.

thresholds are not equivalent to FPR and, moreover, predicted positive findings interpreted in typical *qpAdm* analyses are not rejections of the null hypothesis, but nonrejections (see *Introduction*).

## *qpAdm* performance at the level of experiments

In this section, *qpAdm* results are interpreted from the perspective of experiments, which progress from simple to complex models (from 1-way to 2-, 3-, and 4-way models in our study) and are stopped (produce a positive result) as soon as at least one fitting model is found for a chosen target. Experiments are defined not only by a target but also by a set of "right" demes or a rotating set. Approaches of this sort are often used in high-throughput screens for admixture that rely on *qpAdm* (see, e.g. Narasimhan *et al*. 2019; Bergström *et al*. 2022; Lazaridis *et al*. 2022; Zeng *et al*. 2023; Speidel *et al*. 2025). When interpreting *qpAdm* results at the level of experiments, we paid attention to fractions of experiments producing at least one fitting model at various model complexity levels and to fractions of those "positive" outcomes that would be misleading in practice. In short, we define such misleading outcomes in the following way (see *Methods* for a precise definition): models rejected in those experiments are much closer to the ideal symmetric models, as compared to models fitting in those experiments. For declaring a positive result, it was required that all simpler models for the target (all possible source combinations) are rejected according to a chosen composite feasibility criterion.

For a complete overview of our results at the level of experiments, see Supplementary Fig. 27 (relying on maximal ST distance) and Supplementary Fig. 28 (relying on average ST distance). A key observation is that the most optimal *qpAdm* setup [the densest, "$10^{-3}$ to $10^{-2}$", landscape, EAF $\pm$ 2 SE $\in$ (0, 1), *P*-value $\geq$0.001, "trailing" models not checked] found in the previous section indeed results in the lowest fractions of positive experiments with "bad" outcomes, when compared with other conditions tested (compare Fig. 8 and Supplementary Fig. 27). Here, we focus on the results for all the *qpAdm* protocols under these optimal conditions, and they share a common pattern: fitting models were found for 92% of targets; 55% of experiments ended at 1-way models and 36.5% at 2-way models (on average across all simulation replicates and protocols; Fig. 8). Nearly, all experiments of both types were not classified as producing misleading results (Fig. 8). Nevertheless, the outcome can hardly be viewed as satisfactory since 1-way and 2-way models are too simple for these landscapes with high-intensity gene flows.

In general, increasing SSL density from "$10^{-4}$–$10^{-3}$" to "$10^{-5}$–$10^{-2}$" and to "$10^{-3}$–$10^{-2}$" leads to complex models emerging less often as outcomes of experiments (Supplementary Figs. 27 and 28). For example, experiments on the "$10^{-5}$ to $10^{-2}$" landscapes yielded 3- and 4-way models significantly more often than experiments on the "$10^{-3}$ to $10^{-2}$" landscapes discussed above. The same effect has increased stringency of conditions on EAF. Raising the *P*-value threshold, on the contrary, increases the fraction of complex models as experiment outcomes, but at the cost of significantly higher rates of positive experiments with potentially misleading outcomes (Fig. 8; Supplementary Figs. 27 and 28). Introducing the stringent condition on EAF [EAF $\pm$ 2 SE $\in$ (0, 1)] allowed low rates of "bad" positive outcomes (up to 10–20%) to be achieved even for high *P*-value thresholds (Supplementary Fig. 27c).

Another major trend is that a substantial fraction of nonrotating experiments did not end even at the stage of 4-way models (from ca. 90 to 30%, depending mostly on the *P*-value threshold), in contrast to rotating experiments, and the distributions of experiments by model complexity at which they end are even more skewed toward simpler models in this case (this effect is especially pronounced in the case of the $10^{-5}$ to $10^{-2}$ and sparser landscapes; see Supplementary Figs. 27 and 28). A fundamental difference between these 2 protocol types underlies this observation: a key point is that small average distance between "left" and "right" deme sets increases the probability of encountering violations of the *qpAdm* assumptions (e.g. "left-to-right" gene flows) that lower *P*-values and thus increase model rejection frequency (Fig. 1; Box 2). We explored this issue directly (Supplementary Fig. 29) in the systematic SSL sampling context (see *Systematic sampling: model optimality metrics in the spaces of estimated admixture fractions and P-values*): for 2 to 10 target demes in the center of the "$10^{-5}$ to $10^{-2}$" landscapes, we generated distal nonrotating and rotating experiments characterized by varying average "left–right" distances (proxy, rotating, and "right" demes were sampled from circles at varying distance around the targets as shown in Supplementary Fig. 16).

In a nonrotating experiment, distances from the "right" demes to the target and to each alternative source remain fixed, but in a rotating experiment, many bisections of the rotating set are tested; consequently, a wider range of average "left–right" distances is sampled (Supplementary Fig. 29d), and such an experiment may end at a model with a large average "left–right" distance since it has a relatively low rejection probability. As we consider experiment outcomes on many randomly generated "left" and "right" sets, some nonrotating experiments end quickly because of large average "left–right" distance in most optimal source configurations available in those experiments, others end at much more complex models because of small average "left–right" distance (in most optimal models), and rotating experiments end somewhere in between since "left–right" distance is optimized within each experiment. For an illustration of this argument, see contrasting *P*-value distributions for nonrotating and rotating systematic 2- and 3-way models matched by the number of target's nearest neighbors included (Supplementary Fig. 29d). At the level of systematic experiments, we observe that there are "left–right" distances which cause nearly all experiments (of 20 performed in each case) to progress beyond 4-way models, and the experiment ending rate grows with average "left–right" distance (Supplementary Fig. 29a). For example, even some "inverted" proxy and "right" deme arrangements (the latter closer to the target than the former) demonstrate relatively high experiment ending rates, such as all proxies at a distance of 4 and "right" demes at distances of 1 and 2 (Supplementary Fig. 29a). Indeed, *P*-value distributions are skewed toward values <<0.01 for small average "left–right" distances and become more uniform as "left–right" distance grows (Supplementary Fig. 29c and d).

A general conclusion from this section is that it is very difficult to find setups for randomized *qpAdm* experiments in the SSL context that give both complex and accurate results: at least 3-way models that are close to the ideal symmetric ones. To illustrate this trend, we show outcomes of 2 experiments (Supplementary Fig. 30) relying on one of the most optimal composite feasibility criteria [EAF $\pm$ 2 SE $\in$ (0, 1), *P*-value $\geq$0.001, "trailing" models not checked], where only one fitting 2-way model was found in each (the "$10^{-5}$ to $10^{-2}$" landscapes, distal rotating protocol, rotating sets composed of 18 demes sampled 100 and/or 300 generations ago). Attempting to get more complex models (3-way) resulted in nonoptimal models being accepted and much more optimal models being rejected (Supplementary Fig. 30). Considering patchy sampling simulated here (Supplementary Fig. 30), this

situation may be wrongly interpreted as a signal of a long-distance migration, especially in a high-throughput setting when each experiment is not carefully assessed.

## SSL histories: systematic sampling of landscapes

We aimed to explore how the availability of close ancestry sources (the nearest neighbors of a target deme, also termed "1st circle") influences outcomes of *qpAdm* experiments on the most realistic simulated landscapes ("$10^{-5}$ to $10^{-2}$", see $F_{ST}$ distributions in Supplementary Fig. 14a). In contrast to the randomized deme sets tested above, we used "right," proxy, and rotating sets composed of demes equidistant from the target and distributed uniformly on the landscape, forming circles around the target (Fig. 4; Supplementary Fig. 16). Two kinds of *qpAdm* protocols were tested: distal rotating and distal nonrotating (Supplementary Fig. 16).

## Systematic sampling: model optimality metrics in the spaces of EAFs and *P*-values

We first explored rotating sets composed of 2 deme circles around the target: at distances of 3 and 1 (for a description of the corresponding nonrotating setup see Supplementary Fig. 16). We again inspect median values of the model optimality metrics in the spaces of max|EAF−EF| and *P*-values (Supplementary Figs. 31 and 32) and show the corresponding Spearman's $\rho$ for the optimality metrics vs max|EAF−EF| or *P*-values in Supplementary Table 7 and correlation coefficients stratified by *qpAdm* protocols in Supplementary Table 8. In these tables, the results relying on randomized and systematic deme sampling are juxtaposed. In Supplementary Figs. 31 and 32 (based on maximal and average ST distance, respectively), we also juxtapose results relying on randomized and systematic sampling of the "$10^{-5}$ to $10^{-2}$" landscapes. These 2 sampling approaches demonstrate similar results (Supplementary Fig. 32), however there are generally stronger correlations between max|EAF−EF| or *P*-values and the optimality metrics in the case of systematic sampling (Supplementary Table 7). These results suggest that a particular arrangement of "right" groups (evenly distributed on a circle around the target) improves performance of the method (that is, correlation between *P*-values and model optimality metrics); however, the improvement is not striking. This arrangement of "right" demes also leads to rejection of a larger fraction of ideal 2- or 3-way models as compared to that under randomized deme sampling (Supplementary Table 9). We note that for nearly all ideal 2- and 3-way models under systematic sampling, EAF $\in$ (0, 1) (Supplementary Table 9).

Next, we broadened the sets of proxy sources tested to encompass all possible 1- to 4-deme sets at a particular time (this is feasible in the nonrotating context only) and inspected the influence of target–"right" distance on *qpAdm* performance: minimal target–"right" distance varied from 1 to 4 for 2 suitable target demes at the center of the landscapes, while keeping the size of the "right" set constant at 16 demes. We present the results in the usual way: in the spaces of max|EAF−EF| and *P*-values (Supplementary Figs. 33 and 34) and as Spearman's correlation coefficients between the model optimality metrics and max|EAF−EF| or *P*-values in various, more or less practically relevant, parts of these spaces (Supplementary Table 12). These spaces are in general similar to those based on randomized *qpAdm* tests explored above (Fig. 5; Supplementary Fig. 17), with simpler models (2-way) demonstrating better performance than 3- or 4-way models. The same difference in *qpAdm* performance is seen when models composed exclusively of target's nearest neighbors are considered true, and all the other models are considered false: prestudy odds fall with increasing model complexity, and FDR

grows mainly for this reason (Supplementary Table 13). The best performance of the method (strongest correlation between the model optimality metrics and max|EAF−EF| or *P*-values) is observed when "right" demes are in the 3rd or 4th circles around the target (see, e.g. the correlation between *P*-values and average ST distances in Supplementary Table 12). The distributions of ideal symmetric 2- to 4-way models also become less uniform on the *P*-value axis with decreasing target–"right" distance (Supplementary Figs. 33 and 34) since "right" demes move closer to "left" demes adjacent to the target, which increases the probability of "left-to-right" gene flows bypassing the target and causes *P*-values to drop on average (see the discussion in the preceding section *qpAdm performance at the level of experiments*).

## Systematic sampling: exploring truly ideal 6-way models

Due to computational limitations, in all our randomized and systematic *qpAdm* experiments described above, the most complex models tested were 4-way. However, on the SSL simulated in this study (Fig. 4), truly ideal models for most demes are 6-way. Here, we do not consider cases when a direct ancestor is available since this makes modeling population history trivial in our systematic setups based on the "$10^{-5}$ to $10^{-2}$" landscapes: instead of 7-way models, 1-way models fit the data in nearly all cases. Here we explore all possible 1- to 4-way and 6-way models for our standard systematic setup: distal rotating and distal nonrotating protocols based on demes from the 1st and 3rd circles around the target (all demes with complete 3rd circles were included as targets). More precisely, demes from the 1st and 3rd circles comprised 16-deme rotating or proxy sets (Supplementary Fig. 16). Results based on this setup are also presented in Supplementary Figs. 31 and 32 and Tables 7–9, but here we show results in ideal conditions only, where all 6 demes from the 1st circle were available in *qpAdm* experiments (not to be confused with individual models). We first look at the results in the max|EAF−EF| vs *P*-value spaces (Fig. 9a; Supplementary Fig. 35a and b).

We noted above that *P*-value is not an optimal criterion for ranking randomized or systematic *qpAdm* models on the "$10^{-5}$ to $10^{-2}$" landscapes (and most other landscape types tested) and that max|EAF−EF| correlates better with the model optimality metrics (Fig. 5; Supplementary Tables 7 and 9). For example, large fractions of "ideal" symmetric 3-way models are rejected according to *P*-value thresholds such as 0.01 or 0.05 (Supplementary Table 9); however, this happens precisely because any 3-way model is not truly ideal on our hexagonal grid, and therefore, *P*-value distributions are not uniform for this model complexity (Supplementary Figs. 33 and 34). In contrast, 6-way models composed of target's nearest neighbors demonstrate uniform distributions of *P*-values (Fig. 9a), as expected for ideal models (Harney *et al.* 2021). Nonoptimal sources in this setup come from the 3rd circle, the same circle all the "right" groups (200 generations deeper in time) in the nonrotating setup and most of the rotating set come from (Supplementary Fig. 16). Thus, removing all demes from the 3rd circle from a *qpAdm* model eliminates detectable "left-to-right" gene flows and results in a high *P*-value.

To look at feasibility criteria systematically, we show *qpAdm* positive rates for models stratified by complexity and optimality, which is defined here as the number of demes from the 1st circle included in a model (Fig. 9b; Supplementary Fig. 35c). For simplicity, just 3 criteria are tested: *P*-value ≥0.01 and EAF $\in$ (0, 1), *P*-value ≥0.01 only, and EAF $\in$ (0, 1) only. The systematic distal nonrotating protocol with the second feasibility criterion (the *P*-value threshold) demonstrates attractive properties in the case of 3-, 4-,

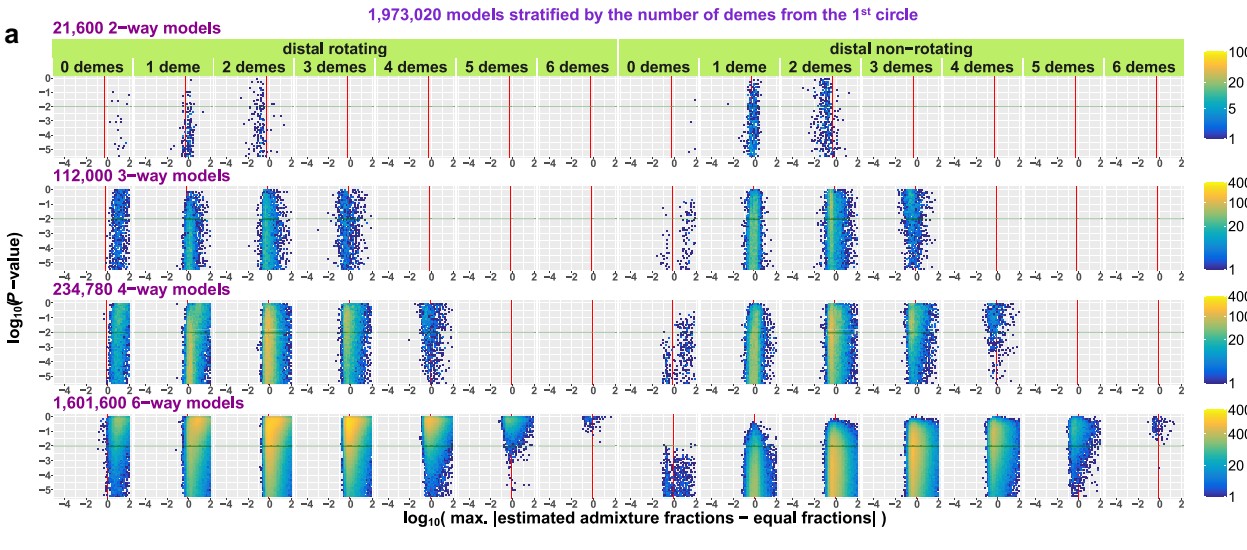

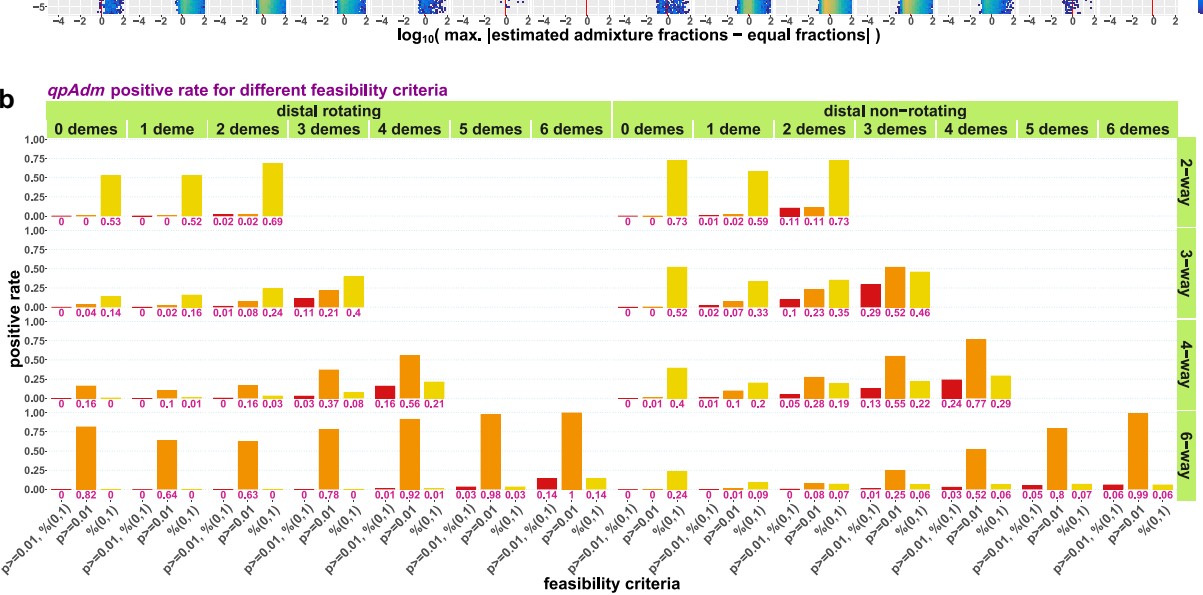

**Fig. 9.** a) Systematic 2- to 4- and 6-way *qpAdm* models on the "$10^{-5}$ to $10^{-2}$" landscapes in the space of max|EAF−EF| and *P*-values. Results are stratified by model complexity, *qpAdm* protocol, and number of target's nearest neighbors in a model, which is the only optimality metric in this analysis. A part of the whole space is shown (*P*-values from $3.16 \times 10^{-6}$ to 1), and for a wider section of the space and for 1-way models, see Supplementary Fig. 35a. The space is divided into rectangular bins ($50 \times 50$), and they are colored by density of individual *qpAdm* models populating this space (see the logarithmic color scales on the right). The vertical lines (or the tick marks) mark max|EAF−EF| = 1−EF; for 3-way and more complex models, this boundary is similar but not identical to the boundary between EAF within (0, 1) and outside (0, 1). The horizontal lines mark the *P*-value threshold used in all our systematic experiments (0.01). b) We show as bar plots fractions of tested models that satisfy 3 kinds of feasibility criteria marked on the x-axis. The fractions are also indicated below the bars.

and 6-way models: there is strong correlation between the fraction of models passing the *P*-value threshold and model optimality, and the percentage of truly ideal models rejected equals the *P*-value threshold, as expected (Fig. 9b). In contrast, such a "clean" result was not achieved with the distal rotating protocol: the fraction of models passing the *P*-value threshold grows with increasing model complexity, but at the same time its correlation with model optimality disappears (Fig. 9b). The fraction of models satisfying the EAF condition demonstrates the opposite pattern: it is correlated with model optimality when the rotating protocol is used and uncorrelated if the nonrotating protocol is used (Fig. 9b). But 86% or 94% of ideal 6-way models were rejected according to the condition EAF ∈ (0, 1) in the rotating and nonrotating setups, respectively (Fig. 9b), demonstrating that AF estimates become highly uncertain for complex models. Thus, in this particular systematic setup (Supplementary Fig. 16), the nonrotating protocol with model

feasibility criteria based solely on *P*-values is performing substantially better than the rotating one with model feasibility criteria based on EAF and/or *P*-values, but it is hard to say if this observation can be generalized to other regions of the huge parameter space we try to explore.

It is possible to look at the same system through the lens of FDR. For simplicity, admixture models composed exclusively of target's nearest neighbors were considered true and all the other models were considered false. Although (if feasibility conditions on both EAF and *P*-values are used) FPR is low (0–3.7%) and independent of model complexity (Supplementary Table 13), FDR for 2- and 3-way models are drastically different: 0 and 40%, and FDR reaches 98.4% in the case of 6-way models [the distal rotating protocol; model feasibility criteria: *P*-value ≥0.1 and EAF ∈ (0, 1); see Supplementary Table 13]. These results are due to prestudy odds falling rapidly with increasing model complexity: from

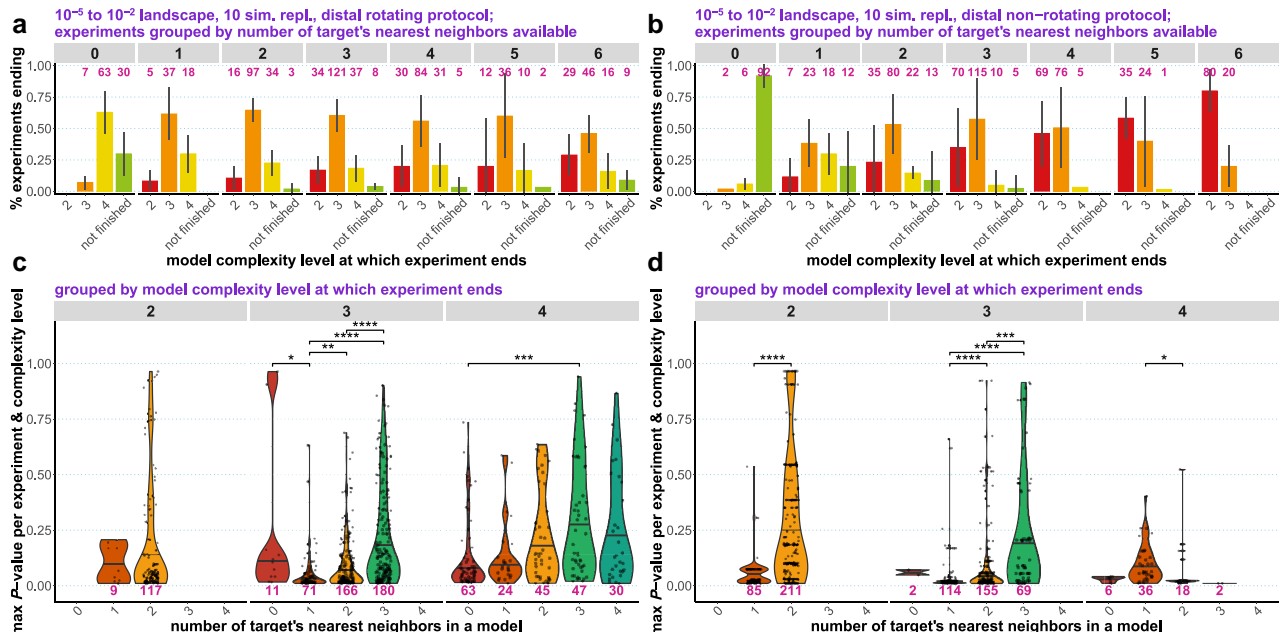

**Fig. 10.** *qpAdm* performance in the case of systematic and symmetric landscape sampling, interpreted at the level of experiments. Results for the distal rotating (a and c) and distal nonrotating (b and d) protocols are shown. The protocols relied on the following composite feasibility criterion: *P*-value threshold at 0.01, EAF between 0 and 1. In a) and b), distributions of experiments over model complexity levels at which their ends are shown, and experiments are grouped by the number of target's nearest neighbors (demes from the 1st circle) available. The other proxy sources and "right" groups were taken from the 3rd circle around the target. The error bars show standard deviation calculated on simulation replicates. In c) and d), each experiment is represented by only one fitting model with the highest *P*-value. We show distributions of *P*-values for these models (the scale on the left) as violin plots with medians, stratified by 2 variables: model complexity level at which an experiment ends (the scale on top), and number of target's nearest neighbors included in a model that was chosen to represent an experiment (the scale at the bottom). For each model complexity level, all pairs of the distributions were compared using the 2-sided nonpaired Wilcoxon test, and only significant *P*-values (adjusted for multiple comparisons with the Holm method) are shown. The asterisks stand for the following significance levels: * ≤ 0.05; ** ≤ 0.01; *** ≤ 0.001; **** ≤ 0.0001. The numbers above the bar plots and below the violin plots in panels a–d) stand for the number of experiments in the respective categories in all simulation replicates combined.

0.143 for 2-way, 0.037 for 3-way, and $1.2 \times 10^{-4}$ for 6-way models in this setup (Supplementary Table 13). FDR is very strongly (negatively) correlated with prestudy odds: Pearson's *r* ranges from −0.86 to −0.99 across various model feasibility criteria we explored (Supplementary Table 13). More relaxed definitions of true models can be used, but the problem remains the same in principle. We also note that, when viewed from the FDR perspective, the composite feasibility criteria are superior to *P*-value thresholds and the distal rotating protocol is superior to the nonrotating one (Supplementary Table 13). These conclusions differ from those based on Fig. 9 since relative sizes of the model optimality classes were not considered in the latter interpretation, but they do influence FDR of high-throughput *qpAdm* protocols where all possible combinations of chosen sources are tested.

## Systematic sampling: *qpAdm* performance at the level of experiments

In this section, we discuss problems with interpretation of systematic *qpAdm* experiments' outcomes. For both the rotating and nonrotating protocols based on demes from the 1st and 3rd circles around the target (Supplementary Fig. 16), we observe the following trend: model complexity at which an experiment ends (the first feasible model is found) decreases with the number of close ancestry sources (demes from the 1st circle) available in that experiment (Fig. 10a and b). This trend is especially striking in the case of the distal nonrotating protocol: when 0 close sources were available, although we relied on a relatively relaxed composite feasibility criterion (*P*-value threshold at 0.01, EAF between 0 and 1), nearly all experiments did not end at the stage of 4-way

models, suggesting that 5-way or even more complex models are needed (Fig. 10b). If 1–4 close sources were available, experiments most often ended at 3-way models, and if all 6 nearest neighbors of a target were available, a great majority of experiments ended at 2-way models.

We also expanded the sets of available proxy sources in the distal nonrotating protocol to all possible combinations of demes at 100 generations before "present" and varied target–"right" distance (see *Systematic sampling: model optimality metrics in the spaces of estimated admixture fractions and P-values*). In line with the results above, most of those experiments with the widest selection of sources available ended at 2-way models, and a majority of those fitting [EAF ∈ (0, 1); *P*-value ≥0.01] 2-way models included just one nearest neighbor of a target (Supplementary Fig. 36). The fraction of experiments ending at 3-way models grew with decreasing target–"right" distance, and no experiment ended at 4-way models (Supplementary Fig. 36).

Naively, we would expect to see an opposite trend: *qpAdm* stops at more complex models when more true ancestry sources are available in a panel of populations tested. But in our experiments *qpAdm* results can be considered misleading in all cases. When no or few close ancestry sources are available, the method tends to pick distant sources in the context of complex models. When (nearly) all closest ancestry sources are available, the method oversimplifies the history of the target by picking just 2 or 3 sources.

This seemingly puzzling pattern (Fig. 10; Supplementary Figs. 36 and 37) can be explained by the relationship between average "left–right" spatial distance and *qpAdm* *P*-values (Supplementary

Fig. 29; see also *qpAdm performance at the level of experiments*). The rotating sets in the experiments discussed here were split in different ratios between the 1st and 3rd circles (Supplementary Fig. 16). Consider, for instance, 3- and 4-way rotating models including no proxy sources from the 1st circle, that is, with all proxy sources coming from the 3rd circle (Supplementary Fig. 29d). Even though both source and "right" demes are sampled from the same circle in this case, there exists a fraction of these rotating set bisections with average "left–right" distances >4 and *P*-values >0.01 (Supplementary Fig. 29d), which corresponds to ca. 75% of experiments ending at 3- and 4-way models in the leftmost panel of Fig. 10a. In contrast, nonrotating 3- and 4-way models of the same type (with both proxy and "right" demes coming from the 3rd circle) demonstrate average "left–right" distances in a narrow range (3.5–3.75) and *P*-values mostly <<0.01 (Supplementary Fig. 29d), and hence ca. 90% of experiments progress beyond 4-way models (see the leftmost panel in Fig. 10b). And, the opposite situation is observed at the other end of the spectrum for models composed of target's nearest neighbors only: 3- and 4-way nonrotating models demonstrate larger "left–right" distances on average and higher *P*-values as compared to rotating models (Supplementary Fig. 29d), corresponding to the significantly different experiment ending rates in the rightmost panels in Fig. 10a and b. When the whole landscape is sampled in an experiment (Supplementary Fig. 36), average "left–right" distance varies widely in models from such an experiment, but only "left–right" distances in most

optimal models composed of sources close to the target determine how fast experiments end. Thus, decreasing target–"right" distance also decreases "left–right" distance for optimal models, drives their *P*-values down (Supplementary Fig. 29c), and causes experiments to end at more complex models. The patterns highlighted here are peculiar to our "circular" rotating and nonrotating setups, but they are instructive and help to understand the patterns observed on random landscape samples (see *qpAdm performance at the level of experiments*).

## Geographical analysis of large sets of *qpAdm* models on real data

In this section, we use the intuition and approaches from our analyses on SSL to interpret results of high-throughput *qpAdm* protocols applied to human archaeogenetic data from 2 preprints: a study of the Paleolithic to Bronze Age Siberia (Zeng *et al.* 2023) and a study of Medieval Europe relying on mostly Iron Age and Roman-period source groups (Speidel *et al.* 2024). Considering that a much more ancient period was studied in the former paper and that the archaeogenetic sampling in Siberia is much poorer as compared to Europe, the studies are markedly different in the sparsity of their geographic sampling (all the proxy sources/outgroups and "fixed outgroups" used by Speidel *et al.* 2024 come from Europe and Anatolia, except for "Ekven IA" from Chukotka; these are small regions when compared with North, Central, and East Eurasia in the study by Zeng *et al.* 2023).

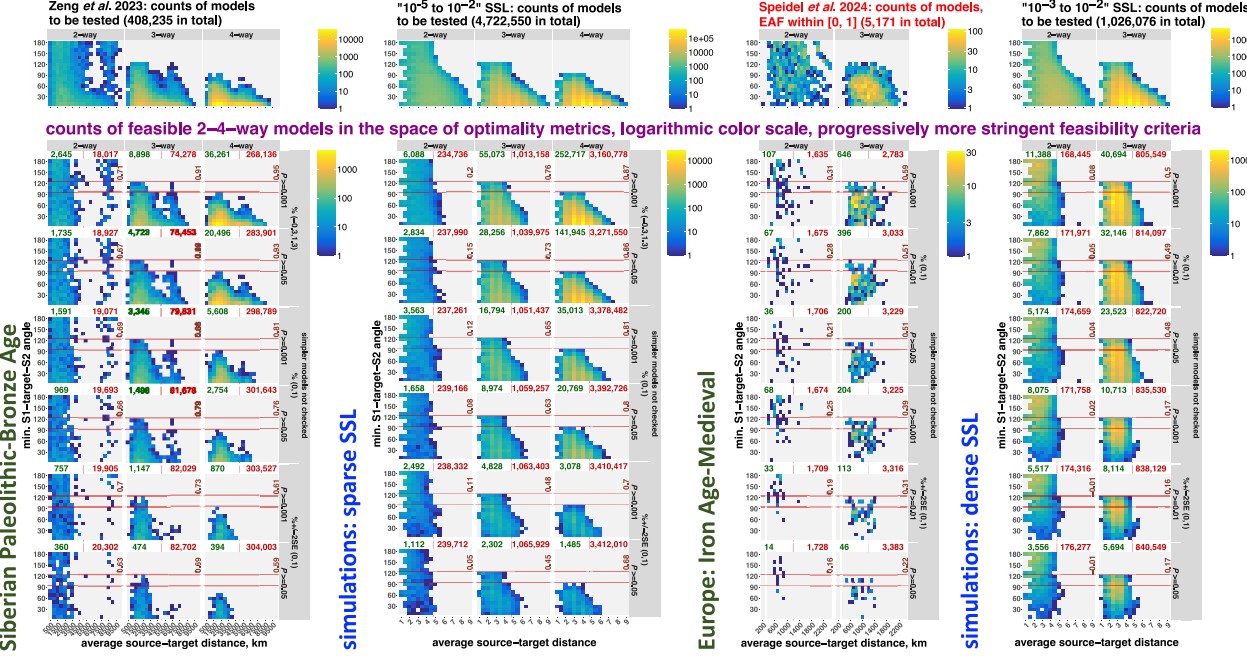

**Fig. 11.** Comparing *qpAdm* models analyzed by the high-throughput protocols from the studies by Zeng *et al.* (2023) and Speidel *et al.* (2024) and results of the randomized *qpAdm* protocol on the "$10^{-5}$ to $10^{-2}$" and "$10^{-3}$ to $10^{-2}$" landscapes, respectively, by placing them in the spaces of optimality metrics (average ST distance vs min. STS angle). Results are stratified by model complexity. The upper row of plots shows density of all published models in this space (all models tested in the case of Zeng *et al.* and models with EAF ∈ [0, 1] in the case of Speidel *et al.*) and the other rows show density of fitting models satisfying selected composite feasibility criteria listed in the captions on the right (6 of 36 criteria tested in this study, from the least stringent on top to the most stringent at the bottom). All the color scales are logarithmic. The *x*-axis is divided into 20 or 22 bins in the case of the real data and 18 bins in the case of the simulated data; the *y*-axis is divided into 18 bins (intervals [0°, 10°], (10°, 20°], and so forth). Spearman's correlation coefficients for counts of all published models vs fitting models in these bins are shown in each panel in brown (bins not populated by any models were not considered). The horizontal lines mark the "ideal" min. STS angles for 3- and 4-way models (120° and 90°). The numbers above each plot on the right and on the left stand for counts of rejected and fitting models in these 2 analyses, respectively, and the vertical lines mark the average ST distance (4.5) that equals the radius of the landscape on simulated data and the midpoints (5,000 or 1,200 km) between minimal and maximal average ST distances on the real landscapes. The distances and angles on the real data (great circle distances in km and angles between 2 bearings) are based on centroids of groups calculated using an *R* implementation of the *Mean Center* tool from *ArcGIS Pro*. We considered ST distances ≤50 km to be negligible, and for such models, min. STS angles were not defined (following the approach we applied to zero-length ST distances on the simulated data), excluding them from this analysis.

We interpret each model (ST configuration) tested with *qpAdm* on real data in a geographical sense by calculating great circle distances between centroids of source and target groups and min. STS angles on a sphere. Thus, we calculated all the model optimality metrics defined in the SSL context (Box 2). The analysis is based on ca. 422,000 two- to 4-way models collected across 725 distal and proximal rotating experiments from the study by Zeng *et al.* (2023) (some experiments in the cited study progressed far beyond 4-way models, but to match our randomized protocol on SSL, these complexity classes were not considered) and on ca. 5,200 two- to 3-way models collected across 28 distal and proximal rotating experiments from the study by Speidel *et al.* (2024) (see Supplementary Table 3 in that study). All the model metrics (EAF, *P*-values, and geographical coordinates of individuals) were taken from the respective studies. We note that neither in the preprint (Speidel *et al.* 2024), nor in the final study version (Speidel *et al.* 2025) unfiltered *qpAdm* results are reported since models with EAF $\notin$ [0, 1] were omitted ("we remove models with infeasible admixture proportions," to quote the study).

Placing the real *qpAdm* models from Zeng *et al.* in the spaces of max|EAF−EF| and *P*-values and juxtaposing them with the most realistic sparse SSL simulated in this study ("$10^{-5}$ to $10^{-2}$"; Supplementary Fig. 14), we observe that distributions of models in these spaces are strikingly similar (Supplementary Fig. 38; we cannot use the results from Speidel *et al.* for this analysis since models with EAF $\notin$ [0, 1] were not presented). Model optimality metrics such as min. STS angles and maximal/average ST distances are correlated with max|EAF−EF| and (weakly) with *P*-values on both the real and simulated data (Supplementary Fig. 38 and Table 14). It is not unexpected that on the simulated SSL data the correlations are stronger (Supplementary Table 14) since, as mentioned above, on the real data, we used just great circle distances and not realistic travel distances on the landscape of Eurasia. Moreover, some groupings on the real data cover large areas of irregular shape (Zeng *et al.* 2023), adding to uncertainties about their centroids.

At the next step, we explored how both real admixture models from Zeng *et al.* and simulated models populate the spaces of optimality metrics (average ST distances vs min. STS angles, Fig. 11). Since in both cases all possible models constructed from dozens of alternative sources were tested for each target, the distributions of models tested have the hallmarks of random samples: both are skewed toward highly asymmetric models with distant sources (those with min. STS angles close to 0 and average ST distances approaching the landscape radius). And on both the real and simulated data, *qpAdm*, equipped with stringent composite feasibility criteria, does reject efficiently such models with average ST distances above a certain threshold (3,500 km for the real landscape from Zeng *et al.*). However, in the remaining densely populated area of the optimality metric space, the distributions of models passing the *qpAdm* "filter" and the starting distributions are similar as evidenced by Spearman's correlation coefficients (Fig. 11). In other words, there is just very minor enrichment for optimal models achieved on both real data and simulated "$10^{-5}$ to $10^{-2}$" SSL, in contrast to the enrichment level achieved for denser "$10^{-3}$ to $10^{-2}$" landscapes. To be precise, on the "$10^{-5}$ to $10^{-2}$" SSL, the pre*qpAdm* and post-*qpAdm* distributions of models in the optimality metric space are uncorrelated just for the simplest, 2-way, models (Figs. 6 and 11; Supplementary Table 11), but in the case of the real data these distributions are correlated even for 2-way models (Fig. 11).

Although we do not have access to the starting model distributions from Speidel *et al.*, we can juxtapose the distributions of

fitting models in the optimality metric spaces to those on our simulated data. We do observe enrichment of the set of fitting 3-way models for relatively symmetric source configurations and relatively small ST distances, which matches our results on the densest simulated landscapes, "$10^{-3}$ to $10^{-2}$" (Fig. 11). The *Twigstats* algorithm by Speidel *et al.* (2024, 2025) is different from the original *qpAdm* in one way only: it is based on more accurate (Speidel *et al.* 2025) $f_2$-statistics calculated from genome-wide genealogical trees for haplotypes rather than from allele frequencies, and these $f_2$-statistics are then passed to the *ADMIXTOOLS* 2 package (Maier *et al.* 2023) for calculation of $f_4$-statistics and *qpAdm* (the same package was used by Zeng *et al.* and in our study). Our juxtaposition with the simulated results on SSL of varying sparsity (Fig. 11) suggests that denser landscape sampling is probably enough to explain the superior performance of *Twigstats* as compared to that of the classic *qpAdm* from Zeng *et al.* Moreover, it is unclear if the interpretations of fitting *qpAdm* models in terms of historical human migrations (Speidel *et al.* 2025) are justified: while some asymmetric source configurations and some distant proxy sources are retained by the composite feasibility criterion used in the original study (*P*-value >0.01, EAF $\in$ [0, 1]), with increasing criteria stringency distant sources and asymmetric source configurations (min. STS angle <60°) are progressively pruned (Fig. 11). In other words, *qpAdm* models are retained based on their geometric properties, but historically attested long-distance migrations are not expected to be aligned with them.

## Discussion
### AGS histories: general conclusions

Temporal stratification tested in this study and practiced in the literature is of 2 sorts: (1) most or all populations in the "right" set are sampled deeper in the past than those in the "left" set (nonrotating protocols) and (2) a target group postdates (or is as old as) all its proxy sources (distal protocols). Since complexity of AGS histories (number of admixture events) grows over time, temporal stratification of targets and proxy sources is a very efficient way of reducing FDR of *qpAdm* protocols (Fig. 3; Supplementary Table 1b) since it avoids situations where admixture history of proxy sources is more complex than that of the target, which increases the probability of encountering violations of the topological assumptions of *qpAdm* (Fig. 1), and temporal stratification is indeed common in the *qpAdm* literature. The distal rotating and nonrotating protocols invariably demonstrated FDR significantly lower than those of the rotating and nonrotating protocols without temporal stratification (Fig. 3). Both distal protocols demonstrated FDRs around 20% and not different significantly from each other, at least in the case of 3,000-Mb-sized genomes (Fig. 3). In contrast, application of the proximal rotating protocol (Box 1) that can be summarized as "whatever is not on the right is on the left" without any attempt at temporal stratification of the "right" and "left" sets and of proxy sources and targets carries a risk of an FDR above 50%. Adding further levels of support (considering only models supported by PCA and/or an *ADMIXTURE* analysis) does not help to decrease FDR drastically in this case (Supplementary Table 2 and Text 3). Although the proximal model competition protocol (Narasimhan *et al.* 2019; Carlhoff *et al.* 2021; Zhang *et al.* 2021; Maróti *et al.* 2022; Brielle *et al.* 2023; Lee *et al.* 2023) was not tested on multiple simulation or subsampling replicates (Supplementary Table 5), we note that it demonstrated FDR values higher than those of the distal nonrotating protocol (Supplementary Table 4). It is hard to extrapolate this aspect of our results to

real analyses and predict FDR on real data since they depend on prestudy odds, and they in turn depend on complexity and other topological properties of simulated histories, but only one graph complexity level was tested, that is 13 groups and 10 admixture events.

A potential way of improving FDR of high-throughput *qpAdm* protocols applied to AGS histories is combining *qpAdm* with an unsupervised *ADMIXTURE* analysis (Supplementary Text 4); however, this method has known problems even on high-quality data and AGS histories (Lawson *et al.* 2018), and our recommendation is tentative. $f_3$-statistic is a simple method for proving that a population is admixed, and it demonstrated FDR values much lower (6%, see Supplementary Text 3) than those of standalone *qpAdm* protocols, but $f_3$-statistics are applicable only to recent admixture events and/or populations of large effective size since postadmixture drift on the target lineage obscures the signal (Patterson *et al.* 2012; Peter 2022). Moreover, calculating $f_3$-statistics for a target composed of a single pseudohaploid individual is impossible since a heterozygosity estimate is required (Maier *et al.* 2023), and such singleton groups are common in archaeogenetic studies. Researchers should also be aware that $f_3$-statistics are defined on unrooted trees, and that may lead to rare but strong false signals of admixture (Supplementary Fig. 6f).

## If population history conforms to SSL, *qpAdm* results are probably misleading in practice

Although it is very often assumed that the genetic history of humans and many other species is approximated by complex admixture graphs (see a short review in the study by Maier *et al.* 2023), this assumption remains unproven. Strictly speaking, it remains unknown if admixture history of humans, wolves (Bergström *et al.* 2022), and horses (Librado *et al.* 2021, 2024; Taylor *et al.* 2023), to cite some recent examples from the archaeogenetic literature, is approximated reasonably well by admixture graphs composed of isolated populations and episodes of admixture, or rather by 2D stepping stone models, or a combination thereof.

We discuss here several aspects of *qpAdm* protocols on SSL demographies that, in our view, lie at the core of the "high FDR problem" affecting typical high-throughput *qpAdm* protocols: (1) the ratio of true or optimal to false or nonoptimal models (mostly highly asymmetric models with at least one distant proxy source) is low both for relatively small sets of 10–18 alternative sources we constructed per simulated target and for the *qpAdm* screens on real data we revisited and falls rapidly with increasing model complexity (Figs. 6, 7, and 11; Supplementary Table 13); (2) a small fraction of suboptimal *qpAdm* models demonstrates EAF within (0, 1) (Supplementary Figs. 39 and 40 and Text 6) and/or *P*-values above the typical thresholds between 0.001 and 0.1, contributing to FPR <1% (Figs. 6, 7, 9, and 11); and (3) conversely, a large fraction of optimal *qpAdm* models demonstrates EAF outside (0, 1) (this fraction grows with increasing model complexity and reflects growing uncertainties in EAF estimates) and *P*-values below the typical thresholds (this fraction falls with increasing model complexity), contributing to FNR between 50 and 100% (Figs. 7 and 9). Given the low prestudy odds common in exploratory studies (Ioannidis 2005), even very small FPR values are enough to inflate FDR to ca. 50–100%, with high FNR playing a role too [Fig. 7; Supplementary Fig. 24 and Table 13; equation (1)]. Since many equally valid thresholds for classifying admixture models into optimal and nonoptimal ones can be set in the SSL context, we also visualize and compare geometric properties of models in prestudy sets and in sets of fitting models (Figs. 6 and 11; Supplementary Fig. 23). We note that these 3 reasons behind high FDR (low

prestudy odds, very high FNR, and nonzero FPR) are shared by all the high-throughput *qpAdm* protocols we tested: on AGS, on randomly and systematically sampled SSL, despite different simulation approaches and definitions of optimal and nonoptimal models. An interplay between prestudy odds and the properties of the simulated landscapes (mainly sampling sparsity) and composite feasibility criteria affecting FPR and FNR (1−power) creates the patterns that we explored on simulated and real data with an emphasis on practical applications, both at the level of individual *qpAdm* models (Figs. 5, 7 and 9) and at the level of "experiments" (Figs. 8 and 10). Below, we discuss some of the most important patterns observed in this complex system.

First, due to gene flows in all directions violating the topological assumptions of *qpAdm* designed in the AGS framework (Fig. 1), correlation between *P*-values and distance- and angle-based model optimality metrics on SSL is very poor, but EAF correlate with both types of optimality metrics better (Fig. 5; Supplementary Table 8). In addition, our analysis on AGS histories showed that EAF are stable across simulation/subsampling replicates, while *qpAdm* *P*-values vary by many orders of magnitude (Supplementary Fig. 9). Thus, considering both AGS- and SSL-based results we conclude that *P*-value-based model sorting (Narasimhan *et al.* 2019; Fernandes *et al.* 2020; Bergström *et al.* 2022; Lazaridis *et al.* 2022; Brielle *et al.* 2023; Carlhoff *et al.* 2023; Speidel *et al.* 2025), where slight *P*-value differences may affect if a model receives interpretation or not, is problematic.

Another tricky aspect of the composite model feasibility criteria common in the literature is the fact that EAF lie largely within (0, 1) and correlate with simulated gene flow intensities only for sources located symmetrically around the target, but for asymmetric sources they vary widely both outside and, complicating the matters even further, inside (0, 1) (Supplementary Figs. 39 and 40 and Text 6). This property of admixture fraction estimation is not recognized in practice and thus may often confound interpretation of *qpAdm* results on SSL (if the interpretation of rejected and fitting source configurations is not explicitly spatial). For instance, a very distant proxy source may be chosen instead of a much closer one (Supplementary Fig. 30) because the latter configuration of sources happens to be slightly less symmetric and yields a several orders of magnitude lower *P*-value due to complex patterns of "left–right" gene flows on SSL. And highlighting a distant source and rejecting a close source may lead to false findings of long-distance migrations.

For many species, geographical regions and historical periods, archaeogenetic sampling is very sparse and may be random with respect to population density of ancient individuals in those regions/periods. Moreover, a general quantitative algorithm for selection of optimal "right" group sets for *qpAdm* analyses has never been published, and "right" groups are usually chosen based on qualitative criteria (Harney *et al.* 2021) or borrowed from study to study. Motivated by these considerations and by the occurrence in the literature of *qpAdm* protocols exploring thousands of alternative models for dozens of target groups (e.g. Narasimhan *et al.* 2019; Lazaridis *et al.* 2022; Zeng *et al.* 2023; Speidel *et al.* 2025; the latter 2 studies were revisited here), we aimed to find expectations for *qpAdm* performance under random sampling of "right," target, and source demes from SSL, and our central analysis relies on thousands of such random sets of "right" demes or random rotating sets of 2 sizes (Figs. 5, 6 and 8). Highly asymmetric models with distant sources are by far the most common in random sets of demes drawn from simulated landscapes (Figs. 6; Supplementary Fig. 23) and real archaeogenetic data (Fig. 11): that is, prestudy odds are low (Fig. 7). Efficient rejection of these nonoptimal models (with

average ST distance greater or approximately equal to the landscape radius) is possible only for very specific combinations of landscape sparsity (influencing mostly power = 1−FNR) and model feasibility criteria (influencing both FPR and FNR). Furthermore, this optimal combination of sparsity ("10$^{-3}$ to 10$^{-2}$" landscapes) and feasibility criteria we found is very uncommon in practice: restricting both "right" and "left" deme sets to a subcontinental region, using a low *P*-value threshold such as ≥0.001, and requiring that EAF ± 2 SE ∈ (0, 1) are very rare approaches, to our knowledge. For example, in the study by Speidel *et al.* 2025, a study we revisited, the former condition was satisfied, but the latter 2 were not. The "10$^{-5}$ to 10$^{-2}$" landscapes resemble real *qpAdm* setups much more in distributions of $F_{ST}$ (Supplementary Fig. 14) and in distributions of models in the max|EAF−EF| vs *P*-value coordinates (Supplementary Fig. 38), and they demonstrated significantly poorer *qpAdm* performance in this respect (Fig. 7; Supplementary Fig. 24 and Table 11).

Moving from interpreting individual models to the level of experiments which are usually stopped as soon as a simplest fitting model is found (see, e.g. Narasimhan *et al.* 2019; Lazaridis *et al.* 2022; Zeng *et al.* 2023), we see that *qpAdm* protocols generally find either accurate, but simplistic models (1- to 3-way models with sources close to the target and approximately opposite each other), or complex models that are often inaccurate (Fig. 8; Supplementary Figs. 27 and 30), as illustrated in the FDR-centered discussion above (Fig. 7; Supplementary Fig. 24). The range of model complexities at which experiments end depends on details of *qpAdm* protocols (rotating vs nonrotating), SSL sparsity, *P*-value thresholds and conditions on EAF. Systematic sampling of proxy source and "right" demes in circles around the target deme (Supplementary Fig. 16) demonstrates similar problems at the level of experiments: when the vicinity of the target is poorly sampled *qpAdm* tends to report very complex inaccurate models, but when the vicinity of the target or even the whole landscape is well-sampled, the *qpAdm* experiment approach tends to find overly simple 2- or 3-way models (Fig. 10, Supplementary Fig. 36). We found a common explanation both for this counter-intuitive behavior of the particular systematic setup we tested and for the significant differences in performance of the randomized rotating and nonrotating experiments (Figs. 8–10; Supplementary Figs. 24, 27–29). That is the influence of average spatial distance between "left" and "right" deme sets on the probability of encountering genetic drift shared by proxy and "right" demes to the exclusion of the target or shared by the target and "right" demes to the exclusion of proxy sources (Fig. 1; Box 2) and, consequently, on *qpAdm* *P*-values. The rotating approach, in contrast to the nonrotating one, allows average "left–right" distance to be maximized in a single experiment (Supplementary Fig. 29c and d), generating higher *P*-values on average (Supplementary Fig. 29c and d) and causing randomized rotating experiments to stop at simpler models (Fig. 8; Supplementary Figs. 27 and 28). We note that performance of the randomized rotating and nonrotating protocols is very similar when inspected through the lens of max|EAF−EF| or *P*-value correlations with model optimality metrics, and there are no statistically significant differences between these protocols in FDR (Supplementary Figs. 24–26 and Table 8), but the differences we discuss at the level of experiments are statistically significant (Fig. 8; Supplementary Figs. 27 and 28). These contrasting patterns are seen because for an experiment to end at more complex models it is enough for spatial proximity of "left" and "right" sets to decrease *P*-values in just few optimal models available in that experiment (Supplementary Fig. 29d), with other models having very diverse "left" and "right" arrangements and being rejected for various reasons.

As mentioned above, the accuracy of fitting 2-way *qpAdm* models on SSL is much higher than that of more complex models such as 3- and 4-way (Figs. 5–7 and 11), but the latter could have been more informative about complex gene flow topologies of SSL (see 2 case studies in Supplementary Fig. 30). And, stopping the analysis at simple models may (Figs. 8 and 10; Supplementary Figs. 27 and 36) favor thinking about history in terms of simple graphs instead of SSL. Moreover, under the conditions (SSL sparsity level and model feasibility criteria) producing most accurate 3- and 4-way fitting models (Figs. 6 and 7), *qpAdm* experiments overwhelmingly end at 1- and 2-way models (Fig. 8; Supplementary Figs. 27 and 28). Thus, in the context of SSL it is probably counterproductive to search for a single best model complexity level and a single best *qpAdm* model for a target. This is especially true since both prestudy odds and SSL sparsity are inherently hard to estimate, and choice of optimal feasibility criteria and optimal model complexity range on SSL depends on these factors in a profound way (Figs. 6, 7 and 9; Supplementary Fig. 24 and Table 11). For example, assuming that population history conforms largely to SSL, "distal *qpAdm* modeling" as used in the literature (Haak *et al.* 2015; Mathieson *et al.* 2015, 2018; Lazaridis *et al.* 2016, 2022; Antonio *et al.* 2019; Narasimhan *et al.* 2019; Sikora *et al.* 2019; Marcus *et al.* 2020; Yang *et al.* 2020; Librado *et al.* 2021; Papac *et al.* 2021; Yaka *et al.* 2021; Bergström *et al.* 2022; Patterson *et al.* 2022; Taylor *et al.* 2023; Zeng *et al.* 2023) likely falls in the category of sparsely sampled continent-wide landscapes (Supplementary Fig. 14) since worldwide "right" groups are combined with potential sources distant in time and/or space from the target, and all our analyses concurred that at a certain level of sparsity exploring any models above 2-way generates very noisy results (Figs. 6, 7 and 9, Supplementary Fig. 24 and Table 11).

Our geographical analysis of 2 large sets of *qpAdm* models tested on real human archaeogenetic data (Zeng *et al.* 2023; Speidel *et al.* 2024) revealed patterns strikingly similar to those observed on SSL (Fig. 11, Supplementary Fig. 38), namely poor enrichment of sets of feasible *qpAdm* models for models optimal in the SSL context (that is, high FDR) in the case of sparse landscape sampling in Paleolithic to Bronze Age North, Central, and East Eurasia (Zeng *et al.* 2023), and much better enrichment (low FDR) in the case of dense landscape sampling in Iron Age to Medieval Europe (Speidel *et al.* 2024) with appropriate composite feasibility criteria. This lack of enrichment for optimal models in the former case has 2 potential explanations. First, it is possible that the genetic history of humans in the Paleolithic to Bronze Age Siberia (Zeng *et al.* 2023) was dominated by long-distance migrations and not by an isolation-by-distance landscape with barriers to gene flow. However, our exploration of simulated SSL suggests another alternative: we are dealing with a sparsely sampled landscape which is not very suitable for application of high-throughput *qpAdm* protocols with their low prestudy odds (Supplementary Table 11).

Considering all the problems highlighted above, and the inherent uncertainty about types of genetic history we are dealing with, we recommend that it is safest to use the *qpAdm* method in controlled conditions when relationships among populations are understood well enough to exclude violations of the topological assumptions (Fig. 1; these are mostly "proxy-to-right" and "right-to-proxy" gene flows bypassing the target or "target-to-right" gene flows bypassing the sources), when radiocarbon or context dates of ancient populations are reliable and allow accurate temporal stratification, or when sets of potential proxy sources are well-constrained based on archaeological or historical scholarship (which means that prestudy odds are high): see, for instance, the earliest publications where the

*qpAdm* method was employed (Haak *et al.* 2015; Mathieson *et al.* 2015) and some recent studies (e.g. Marcus *et al.* 2020; Papac *et al.* 2021; Yaka *et al.* 2021; Patterson *et al.* 2022; Changmai, Jaisamut, *et al.* 2022, Changmai, Pinhasi, *et al.* 2022). Obviously, the amount of new information that the *qpAdm* method provides in these conditions is unique for each study and may be low. In contrast, high-throughput *qpAdm* screens aiming to find optimal models (e.g. Narasimhan *et al.* 2019; Bergström *et al.* 2022; Lazaridis *et al.* 2022; Zeng *et al.* 2023; Speidel *et al.* 2025) are not recommended, especially in sparsely sampled regions/periods, such as the Paleolithic in the case of human population history (e.g. Posth *et al.* 2023; Villalba-Mouco *et al.* 2023; Zeng *et al.* 2023; Allentoft *et al.* 2024), Paleolithic to Neolithic Siberia (Sikora *et al.* 2019; Zeng *et al.* 2023; Gill *et al.* 2024), ancient South (Narasimhan *et al.* 2019; Kerdoncuff *et al.* 2024) and Southeast Asia (Carlhoff *et al.* 2021, 2023; Oliveira *et al.* 2022; *qpAdm* played a central role in some of those studies).

In other words, high-throughput *qpAdm* protocols on SSL generate results that very often do not look suspicious but are either overly simple or inaccurate (favoring very distant sources over available spatially and genetically close sources). For instance, finding a 2-way fitting model and rejecting all alternative models of the same complexity can be interpreted in several ways: (1) as a true or (much less likely for distal protocols) a false model in the context of a relatively simple AGS history; (2) as a suboptimal model (with large ST distances and/or small STS angles) in the context of a relatively poorly sampled SSL and/or unsuitable feasibility criteria; and (3) as an optimal but overly simple model in the context of a densely sampled SSL with well-chosen feasibility criteria. We contend that this variety of potential interpretations makes the *qpAdm* method (strictly speaking, the protocols aiming to find and interpret one or few "best-fitting" models for a target in large sets of alternative models) hardly suitable for one of its chief use cases in the literature, namely as a method supplying evidence for long-distance migrations in exploratory studies.

## Methods
### Limitations of the AGS-based analyses in this study

We explored performance of various *qpAdm* protocols on a collection of simulated AGS histories of random topology and uniform complexity, where admixture history of target groups may vary from the simplest (no admixture) to complex (up to 5 admixture events in group's history). We have not explored *qpAdm* performance over a range of simulated admixture graph complexities, over a range of model feasibility criteria (except for those in Supplementary Table 3), for models more complex than 2-way, and have estimated FDR instead of FPR due to an important technical limitation: the process of model classification into true and false ones is difficult to automate fully since it requires careful interpretation of the simulated topology and simulated admixture proportions (this is illustrated by the case studies in Supplementary Fig. 6; note that we attempted to estimate FPR and FNR for *qpAdm* results on AGS histories indirectly; see the end of *AGS histories: influence of temporal stratification and data amount/quality on the performance of qpAdm protocols*).

Due to this limitation, we did not attempt to classify universal rejections of 2-way models by *qpAdm* or other methods into false rejections due to violations of the topological assumptions of *qpAdm* (Fig. 1) and true rejections when the true admixture history of the target does not fit a 2-way model. Since archaeogenetic sampling is often very sparse, we also did not interpret fitting,

but overly simple, models including correct sources as false in the AGS context (see the section *True-positive models* in Supplementary Text 2), but we highlight this problem in the SSL context where the simulated history is even more complex. We started from a collection of random simulated AGS histories of uniform complexity, tested all possible models of chosen complexity classes (only 1- and 2-way), considered 2-way models supported by a *qpAdm* protocol, classified them manually into false and true positives according to a set of topological rules, and subjected them to further screening by the PCA and/or *ADMIXTURE* methods (Supplementary Text 4).

Another important caveat is that complexity of genetic history in a region and period of interest often remains unknown, and it is difficult to judge if a particular admixture graph complexity is adequate for simulating the real history (and if admixture graphs are appropriate models at all, see below).

### Limitations of the SSL-based analyses in this study

While complex admixture models and a wide range of feasibility criteria were not explored on AGS histories above, the following issues were, conversely, not explored on SSL histories: effects of differing amounts and quality of data (all the simulations were based on high-quality diploid data from 500-Mb-sized genomes with mutation and recombination rates typical for humans), combining *qpAdm* with other methods (PCA, *ADMIXTURE*, and $f_3$--statistics), and model competition *qpAdm* protocols. Some of these differences in research coverage stem from the fact that it is impossible to strictly delineate false and true admixture models on SSL since at the timescales we explore (up to 300 generations) all demes are connected by gene flows, directly or indirectly. Instead, it is possible to rank proxy sources by genetic or spatial proximity to the target and symmetry with respect to the target. Since hard-to-automate classification of admixture models into false and true ones can be avoided, SSL histories are more tractable analytically than complex AGS histories.

Since SSL simulations are computationally demanding, our analysis was restricted to relatively small landscapes composed of 64 demes, small samples per deme (3 individuals), and just 3 sampling points in time. For the same reason, sampling was not randomized over >>3 points in time, and different time intervals between sampling points were not tested. More complex landscapes with rapid long-distance migrations, extinctions, repopulations, and changes in effective population size were also not explored for reasons of computational complexity. Thus, our SSLs are still simple when compared with the real population history of humans and other species as we understand them. Ultimately, true isolation-by-distance landscapes should be explored instead of quantized deme-based landscapes, and the former can also be simulated by the *slendr* package (Petr *et al.* 2023) we used.

Another notable limitation discussed in the section *SSL histories: randomized sampling of landscapes* is our reliance on spatial ST distances for measuring model optimality. $F_{ST}$ (Supplementary Fig. 14a) and haplotype-sharing statistics depend on both source-to-target and target-to-source gene flows, and for that reason, they are also not ideal as metrics for ranking proxy sources and were not used as such. Ideally, density clouds of ancestors (Grundler *et al.* 2024) should be inferred for each target and then used for ranking proxy sources or models or for prediction of EAF. However, this analysis is also computationally challenging in our setup aiming to explore a wide range of *qpAdm* models, protocols, and landscape properties, and most prominent features of

*qpAdm* performance on SSL such as effects of prestudy odds, landscape sparsity, and average "left–right" distance can be explored even through the lens of our noisy spatial approximation for true ancestry density (see Supplementary Fig. 14d and e and its discussion above). Moreover, our spatial analytical framework based on distances and angles can be directly applied to rich sets of *qpAdm* models from the literature (Fig. 11; Supplementary Fig. 38) without inference of clouds of ancestry density, which is even more complicated for real ancient DNA data.

## AGS histories: simulations

For simulating genetic data, we used *msprime v.1.1.1*, which allows accurate simulation of recombination and of multichromosome diploid genomes relying on the Wright–Fisher model (Nelson *et al.* 2020; Baumdicker *et al.* 2022). We simulated 3, 10, or 30 diploid chromosomes (each 100 Mb long) by specifying a flat recombination rate ($2 \times 10^{-8}$ per nt per generation) along the chromosome and a much higher rate at the chromosome boundaries ($\log_e 2$ or $\sim$0.693 per nt per generation, see https://tskit.dev/msprime/docs/stable/ancestry.html#multiple-chromosomes). A flat mutation rate, $1.25 \times 10^{-8}$ per nt per generation (Scally and Durbin 2012), and the binary mutation model were used. To maintain the correct correlation between chromosomes, the discrete time Wright–Fisher model was used for 25 generations into the past, and deeper in the past the standard coalescent simulation algorithm was used (as recommended by Nelson *et al.* 2020).

AGS histories of random topology including 13 populations and 10 pulse-like admixture events were generated using the *random_admixturegraph* and *random_sim* functions from the *ADMIXTOOLS 2* package (https://uqrmaie1.github.io/admixtools/reference/random_sim.html), which produced scripts for running the *msprime v.1.1.1* simulator. Demographic events were separated by date intervals ranging randomly between 20 and 120 generations, with an upper bound on the graph depth at 800 generations (or ca. 23,000 years in the case of humans). In other sets of simulations, all the dates were scaled up 3.75 times, with an upper bound on the graph depth at 3,000 generations (or 87,000 years in the case of humans). To be more precise, demographic events were not placed in time entirely randomly but were tied to one or few other events of the same "topological depth" within the graph, as illustrated by 10 examples of simulated topologies in Supplementary Fig. 3. The same principle was applied to sampling dates, which were tied to other demographic events such as divergence and admixture of other populations. These restrictions were used to ensure topological consistency of random graphs.

Ten diploid individuals with no missing data were sampled from each population at "leaves" of the graph. Effective population sizes were constant along each edge and were picked randomly from the range of 1,000–10,000 diploid individuals. Admixture proportions for all admixture events varied randomly between 10 and 50%. This setup generates groups sampled at widely different dates in the past or, in other words, located at various genetic distances from the root. Alternatively, all terminal branches were extended to the "present" of the simulation and sampled at "present," keeping their respective effective population sizes and topological relationships unchanged. Thus, another set of simulations was generated for the same topologies, where groups were more drifted with respect to each other (see $F_{ST}$ distributions in Supplementary Fig. 5).

In summary, 5 sets of independent simulations (simulation setups) differing by the amount of data generated and by population divergence metrics were performed for a set of 40 random admixture graph topologies (see an overview in Supplementary Fig. 2):

1) Three 100-Mb-sized chromosomes; groups sampled at different points in time; maximal simulated history depth at 800 generations (10 simulation replicates; for median number of polymorphic sites, see Supplementary Fig. 4).
2) Ten 100-Mb-sized chromosomes; groups sampled at different points in time; maximal simulated history depth at 800 generations (10 simulation replicates).
3) Thirty 100-Mb-sized chromosomes; groups sampled at different points in time; maximal simulated history depth at 3,000 generations (10 simulation replicates).
4) Three 100-Mb-sized chromosomes; groups sampled at different points in time; maximal simulated history depth at 3,000 generations (1 simulation replicate).
5) Three 100-Mb-sized chromosomes; all terminal branches extended to the "present" of the simulation and sampled at that point; maximal simulated history depth at 800 generations (1 simulation replicate).

To create more realistic datasets, we performed randomized subsampling of polymorphic sites and individuals (replicate no. 1 of the 1st, 2nd, and 3rd simulation setups were used for this, see the list above). First, we randomly sampled alleles at heterozygous sites, creating pseudohaploid data. Then, we introduced missing data by randomly selecting a missing rate between 5 and 95%, followed by randomly selecting sites according to the missing rate. This site subsampling was repeated for each individual independently. Last, we randomly sampled $n$ (from 1 to 10) individuals from each population independently. The subsampling procedure described above was conditioned on the number of sites polymorphic in the set of 13 simulated populations and was repeated until a subsampling replicate with more than 20,000 (for 300-Mb-sized genomes) or 66,000 such sites (for 1,000- and 3,000-Mb-sized genomes) was obtained (Supplementary Fig. 4). We generated 10 independent subsampled replicates for each topology and simulation setup (1,200 replicates in total). Polymorphism data in the *EIGENSTRAT* format were generated from the tree sequences using the *TreeSequence.genotype_matrix* function (https://tskit.dev/tskit/docs/stable/python-api.html#tskit.TreeSequence.genotype_matrix) and used for all subsequent analyses (*f*-statistics and *qpAdm*, PCA, and *ADMIXTURE*).

## AGS histories: calculating *f*-statistics

For all the work on *f*-statistics and *qpAdm*, the *ADMIXTOOLS 2* software package (Maier *et al.* 2023) was used. For diploid SNP sets without missing data, we first calculated all possible $f_2$-statistics for 4-Mb-sized genome blocks (with the "*maxmiss = 0*," "*adjust_pseudohaploid = TRUE*," and "*minac2 = FALSE*" settings) and then used them for calculating $f_3$- and $f_4$-statistics as linear combinations of $f_2$-statistics and for testing *qpAdm* models using the *qpadm* function in *ADMIXTOOLS 2* (https://uqrmaie1.github.io/admixtools/) under default settings. Harney *et al.* (2021) showed that *qpAdm* is robust to changes in genome block size except for extreme values: 0.0001 and 1 Morgans (0.05 Morgans is a recommended default value, which corresponds to $\sim$5 Mb in the case of humans). Inferred admixture proportions in our analysis were not constrained between 0 and 1 (see the study by Harney *et al.* 2021 for a justification of this unusual property of the default *qpAdm* algorithm). For pseudohaploid SNP sets with missing data and uneven group sizes, the *qpadm* function was applied to genotype files directly, with the "*allsnps = TRUE*" setting. In other

words, $f_4$-statistics utilized by *qpAdm* and $f_3$-statistics were calculated off the genotype files without intermediate $f_2$-statistics, and removal of missing data was done for each population quadruplet or triplet separately. This setup is often used in the literature in the presence of missing data (e.g. Harney *et al.* 2018, 2019; Narasimhan *et al.* 2019; Lazaridis *et al.* 2022).

## AGS histories: *qpAdm* protocols

*QpWave* tests were performed on sets of 13 groups divided randomly into 2 "left" and 11 "right" groups, testing all possible bisections of this form. *QpAdm* was applied to the same sets of 13 groups divided randomly into 3 "left" and 10 "right" groups, testing all possible bisections of this form for all possible target groups in "left" sets (858 such models per simulated history). This proximal rotating protocol was applied to all simulation setups. Subsequent work was focused only on feasible *qpAdm* models defined as follows: (1) *P*-values calculated by *qpWave* for 1-way models "target = proxy source$_1$," "target = proxy source$_2$," and "proxy source$_1$ = proxy source$_2$" are all below 0.01; (2) in the case of the 2-way model "target = proxy source$_1$ + proxy source$_2$," EAF ± 2 SEs are between 0 and 1; and (3) the *P*-value calculated by *qpAdm* for the 2-way model ≥0.01. For exploring performance of the distal rotating protocol, feasible 2-way *qpAdm* models were simply filtered according to sampling dates of target groups and proxy sources. If target group's sampling date was equal to or smaller than sampling dates of both proxy sources, such a model was considered distal.

In the nonrotating protocol, for each simulated admixture graph, 6 oldest sampled groups were selected as a fixed "right" set (ties in sampling dates were resolved in alphabetical order; these "right" sets remained unchanged for a given topology across all independent simulations), and for the remaining 7 groups, all possible 1-way and 2-way admixture models were tested (105 models), applying the same composite feasibility criterion that was used above for the rotating protocol. This is the proximal nonrotating protocol, and alternatively, we focused on distal admixture models only (distal nonrotating protocol).

In the proximal model competition protocol, subsequent analysis was focused on targets for whom 2 or more alternative *qpAdm* models emerged as feasible at the first step. For each target, alternative proxy sources were pooled and rotated between the "left" and "right" sets, testing only the models that emerged as feasible at the first step and applying the composite feasibility criterion (*P*-value ≥0.01, EAF ± 2 SE are between 0 and 1). Rotation of alternative proxy sources was performed in 2 alternative ways: "whatever is not on the left is on the right" or placement of alternative sources in the "right" set one by one. In the latter case, several "right" sets were tested for each model, and the model was considered supported by the model competition protocol only if it was not rejected under any of these "right" sets (the latter protocol follows the study by Maróti *et al.* 2022). If only one model was feasible for a target, such a model was evaluated as passing the model competition procedure. A distal model competition protocol was not tested in this study.

For testing statistical significance of differences in FDR between *qpAdm* protocols, the following approach was used. FDR was calculated either on low-quality data for 10 random site/individual subsampling replicates derived from simulation replicate no. 1 (simulation setup nos. 1–3) or on high-quality data for 10 independent simulation replicates (simulation setup nos. 1–3). Comparisons of 6 *qpAdm* protocols (rotating and nonrotating; considering exclusively proximal, exclusively distal models, or both) were performed on these 6 sets of replicates using the 2-sided

paired Wilcoxon test (implemented as the *pairwise_wilcox_test* function in the *rstatix* R package v. 0.7.2). Comparisons of the same *qpAdm* protocols on simulations with different amounts or quality of data were performed using the 2-sided nonpaired Wilcoxon test since simulation replicates were independent unlike alternative *qpAdm* protocols applied to the same data (Fig. 3). In all applications of the Wilcoxon test for various purposes (Figs. 3 and 10; Supplementary Fig. 37 and Table 1b), *P*-values adjusted for multiple testing with the Holm method are presented.

For comparing numerical results of *ADMIXTOOLS* 2 v. 2.0.6 (released in 2024) with those of classic *ADMIXTOOLS* v. 7.0 (*qpAdm* v. 1201, released in 2020), we used the most realistic AGS simulations we generated, that is low-quality pseudohaploid data for 3,000-Mb-sized genomes: 10 AGS topologies, with 1 simulation and 1 data subsampling replicate per topology. All possible 2-way models were tested with the rotating protocol. *ADMIXTOOLS* 2 was run with the "*fudge_twice = TRUE*" argument, which makes the results more similar numerically to those of classic *ADMIXTOOLS*. Since the data were of low quality (pseudohaploid, with high missing rates, and with some populations composed on 1 individual), we used the "*allsnps = TRUE*" setting in *ADMIXTOOLS* 2 and the identical "*useallsnps: YES*" setting in classic *ADMIXTOOLS* (all the other settings were default ones). If "*useallsnps*" is set to "*YES*," the latest *ADMIXTOOLS* v. 8.0.1 (*qpAdm* v. 2050, released in 2024) automatically calls *qpfstats* (https://github.com/DReichLab/AdmixTools/blob/master/README. qpfstats), an algorithm for handling low-quality data that is substantially different from that of the "*allsnps = TRUE*" setting in *ADMIXTOOLS* 2 (https://uqrmaie1.github.io/admixtools/ reference/qpfstats.html). To perform a fair comparison, we used *ADMIXTOOLS* v. 7.0 instead of the latest version v. 8.0.1.

The results produced by these 2 packages are not identical, but highly correlated across 8,580 unique models we tested on low-quality data (Supplementary Table 15). Pearson's correlation coefficient is 0.99 for *P*-values (this analysis prioritizes high *P*-values), 0.91 for $\log_{10}(|EAF_1|)$, 0.73 for $\log_{10}(P\text{-values})$, and 0.79 for $\log_{10}(SE)$. Spearman's coefficients are slightly higher for the logarithmic values (all coefficients are statistically significant with 2-sided *P*-values = 0). The weaker correlation seen for logarithmic *P*-values is in line with the high variability in logarithmic *P*-values calculated by *ADMIXTOOLS* 2 on simulation replicates (Supplementary Fig. 9).

## AGS histories: classifying 2-way admixture models into FP and TP

Since the simulated admixture graph topologies were complex and random, target groups modeled with *qpAdm* had complex admixture history in some cases, being a part of gene flow networks. In this context it is hard to draw a strict boundary between true and false admixture models composed of a target and only 2 proxy sources. Two-way admixture models were considered false only if at least one of the following criteria was satisfied (considering only graph topologies and admixture proportions):

1) The target and at least one of the proxy sources are simulated as strictly cladal (Fig. 2; Supplementary Fig. 6a). In this case, the target may either be unadmixed, or it may have experienced gene flows earlier in its history that do not break its cladality with one of the proxy sources.

2) A proxy source lineage is a recipient of gene flow from the target lineage (after the last admixture event in a target's history), possibly mediated by other lineages (Fig. 2;

Supplementary Fig. 6a and b). In other words, the incorrect proxy source is a descendant of the target lineage, i.e. the expected gene flow direction is reversed.

3) A proxy source does not represent any true source. In other words, it is symmetrically related to all true sources of ancestry in the target (Supplementary Fig. 6c and d). Alternatively, both proxy sources represent the same true source and are symmetrically related to all the other true sources (Supplementary Fig. 6e).

4) A proxy source shares genetic drift with the corresponding true source that is not shared by the second proxy source (and the same is true for the other proxy source and another true source, i.e. condition no. 3 above is not satisfied); however, <40% of its ancestry is derived from the true source (Supplementary Fig. 6a and d).

5) The target gets a gene flow from a deep-branching source not represented by any sampled population, and an inappropriate proxy source is included in the fitting model (Supplementary Fig. 6f and g).

We illustrate these topological rules with 8 case studies of FP and feasible *qpAdm* models in Fig. 2 and Supplementary Fig. 6a–g. Two-way models for targets whose population history is best approximated with 3-way and more complex models were considered as TP if they included source proxies (that do *not* satisfy the criteria above) for at least 2 of 3 or more true ancestry sources (see 3 case studies in Supplementary Fig. 6k–m).

## AGS histories: PCA

PCA was performed for 1 simulation replicate per simulation setup. Only high-quality data was used, and all individuals sampled from a simulation were coanalyzed. Prior to the analysis, sites with allelic states in statistical association were pruned with *PLINK v.2.00a3LM* (Chang *et al.* 2015) using the following settings: window size, 2000 SNPs; window step, 100 SNPs; and $r^2$ threshold = 0.5 (argument "–indep-pairwise 2000 100 0.5"). This kind of linkage disequilibrium (LD) pruning is common in the population genetic literature since complex LD structure may confound PCA, especially higher PCs (Privé *et al.* 2020). PCA was also performed using *PLINK v.2.00a3LM* under default settings, calculating 10 PCs. Interactive 3D plots visualizing PC1, PC2, and PC3 were made using the *plotly* R package. A 2-way admixture model was considered supported by PCA if: the target group (the center of the cluster of target individuals, to be precise) lay between the clusters of proxy source individuals on a straight line in the 3D PC space or it was located at a distance of no more than 3 target cluster diameters from that straight line connecting the proxy source clusters.

The 2nd pattern was more common among both TP and FP 2-way admixture models: 1.5 and 1.3 times, respectively (across all nonsubsampled simulated datasets). This situation is expected since many targets represent 3-way and more complex mixtures and due to postadmixture genetic drift (Peter 2022).

## AGS histories: *ADMIXTURE* analysis

*ADMIXTURE* analysis was performed for 1 simulation replicate per simulation setup. Only high-quality data were used, and all individuals sampled from a simulation were coanalyzed. Prior to the analysis, sites with allelic states in statistical association were pruned with *PLINK v.2.00a3LM* (Chang *et al.* 2015) using the following settings: window size, 2000 SNPs; window step, 100 SNPs; and $r^2$ threshold = 0.5 (argument "–indep-pairwise 2000 100 0.5"). This kind of LD pruning relying on $r^2$ is recommended since the underlying model assumes sites in linkage equilibrium (Alexander *et al.* 2009). *ADMIXTURE v.1.3* (Alexander *et al.* 2009) was used in the unsupervised mode under the default settings. The algorithm was run on each SNP dataset only once, with the number of hypothetical ancestral populations (K) ranging from 3 to 10. This range was selected since the total number of populations in each simulated history was 13. A 2-way admixture model was considered supported by *ADMIXTURE* analysis if:

1) for at least one K, at least 5 of 10 target individuals were modeled as a mixture of at least 2 ancestry components, with a minor ancestry component exceeding 2%;

2) typically, ancestry component A in the target group was shared with at least 5 individuals in proxy source 1, but not in proxy source 2, and ancestry component B was shared with at least 5 individuals in proxy source 2, but not in proxy source 1 (see examples in Fig. 2 and Supplementary Fig. 6); in some cases, both components A and B were found in the proxy sources, but in different proportions;

3) if only one ancestry component in the target was shared with the 2 proxy sources, then the model was considered unsupported;

4) ancestry components in the target that are absent in any of the sources were ignored since 3-way and more complex admixture histories are common in the set of random admixture graphs explored here (see examples of such topologies in Supplementary Fig. 6k–m);

5) ancestry components in a proxy source that are absent in the target were also ignored since a proxy source may not be fully cladal with the real source (see examples of such topologies in Supplementary Fig. 6i, j, and m).

These rules were designed to reproduce typical reasoning of an archaeogeneticist interpreting *ADMIXTURE* results. Observing a pattern of ancestry components in the target group and proxy sources compatible with the admixture model "target = proxy source$_1$ + proxy source$_2$" for one K value was enough for declaring that the model is supported by the *ADMIXTURE* analysis. This condition was motivated by an observation that models supported at one K value only were equally common among FP and TP *qpAdm* models (10 and 13%, respectively, across simulation setup nos. 1, 2, 4, and 5). Models supported at 4 or more K values were more common among TP *qpAdm* models (3.3% of FP and 12.6% of TP models across the 4 simulation setups).

## SSL histories: simulations

The *slendr* v. 0.7.2.9000 R package (Petr *et al.* 2023) with the *msprime* v. 1.2.0 backend was used for the stepping stone simulations. Sixty-four panmictic demes (with stable effective sizes of 1,000 diploid individuals) arose via multifurcation at generation 470, counted from the past. Starting at generation 471, 3 gene flow eras followed each other: "pre-last glacial maximum (LGM)" lasting 1,400 generations, "LGM" lasting 350 generations, and "post-LGM" lasting 770 generations. Three diploid individuals were sampled from each deme at the end of the simulation ("present"), at 100, and 300 generations before "present." The demes were arranged on a hexagonal lattice (demes were placed at vertices of triangles tiling the plane; not to be confused with hexagonal or honeycomb tiling), forming an approximately circular landscape, where each deme was connected by bidirectional gene flows to 3–6 neighboring demes (Fig. 4). Gene flow intensities were sampled randomly (in each direction independently) from either uniform or normal distributions, generating 4 types of

landscapes with the following approximate per-generation gene flow intensity ranges: $10^{-5}$ to $10^{-4}$, $10^{-4}$ to $10^{-3}$, $10^{-3}$ to $10^{-2}$, $10^{-5}$ to $10^{-2}$ (the values stand for percentage of individuals in a generation coming from a given neighboring deme; see Supplementary Table 6 for details). In the case of the "$10^{-5}$ to $10^{-2}$" landscape, normal distributions centered at 0 were used (see Supplementary Table 6 for their parameters). Random sampling of the gene flow intensity distributions was done at the beginning of each era, except for the "$10^{-3}$ to $10^{-2}$" landscape, where sampling was repeated 5 times per era, at equal intervals of time. Ten independent simulation replicates were generated for each landscape type, with gene flow intensities resampled for each replicate. Genome size was 500 Mb, recombination rate was simulated at $1 \times 10^{-8}$ per nt per generation, and mutation rate was $1.25 \times 10^{-8}$ per nt per generation.

There are so many ways of "unfolding" an initial lattice state via a serial founder process represented as a bifurcating tree (Estavoyer and François 2022) that we decided to avoid this additional level of complexity in our study and initialized the lattice simulations with a multifurcation, but deep in the past, at 2,220 generations before the oldest sampling time point. In this way we focus on genetic structure generated by gene flow networks on SSL instead of structure generated by phylogenetic processes. If gene flow intensities on SSL are relatively high, traces of the initial multifurcation are lost by the beginning of the sampling epoch, but this is not true for simulations with the lowest per-generation gene flow intensities (the "$10^{-5}$ to $10^{-4}$" landscapes, see below), as demonstrated by our analysis of 1-way *qpAdm* models interpreted as cladality tests (Supplementary Text 5 and Fig. 21).

## SSL histories: *qpAdm* protocols with randomized landscape sampling

Four *qpAdm* protocols were applied to SSL: distal and proximal nonrotating, distal, and proximal rotating. For constructing these *qpAdm* experiments, demes were sampled randomly from the landscapes, keeping their original sample size of 3 diploid individuals. For each protocol, 36 composite feasibility criteria and 2 deme sampling densities were tested: 13 or 18 demes and a target per a rotating *qpAdm* experiment, 10 or 15 proxy sources, and 10 or 15 "right" demes and a target per a nonrotating experiment. For brevity, we refer to these sampling densities as "13 demes" or "18 demes" for all types of protocols. Here, we describe random deme sampling rules for the 4 protocols, and they are also illustrated in Supplementary Fig. 16:

1) Distal nonrotating. Target demes were sampled at "present," and for each target, 2 alternative sets of "right" groups and proxy sources were generated: 10 (or 15, depending on the sampling density) "right" demes were from 300 generations before present (ca. 8700 years in the case of humans), and 10 or 15 proxy source demes were from 100 generations before present (ca. 2900 years in the case of humans).

2) Proximal nonrotating. Ten or fifteen "right" demes were sampled at 300 generations before present, 10 or 15 proxy source demes were sampled at 100 generations and at "present" (demes from the 2 dates were combined before sampling), and a target deme was sampled from either the 100- or 300-generation sampling point.

3) Distal rotating. Target demes were sampled at "present," and for each target 2 alternative rotating sets were generated: 13 or 18 "right"/source demes were sampled from 300

and 100 generations ago (demes from the 2 dates were combined before sampling).

4) Proximal rotating. A rotating set of 14 or 19 demes was sampled from all the 3 time points combined, and a single target was sampled from that subset, but 300-generation-old targets were not allowed.

The *qpAdm* experiments described above consisted of all possible 1-, 2-, 3-, and 4-way models for one target (1-, 2-, and 3-way models for proxy sources were also counted toward some feasibility criteria as described below). Before each *qpAdm* test, per-block $f_2$-statistics were calculated with *ADMIXTOOLS* v.2.0.0 (Maier *et al.* 2023) for the selected set of demes under the following settings: "*maxmiss* = 0" (no missing data allowed at the group level), "*blgsize* = 4000000" (genome block size in nt), "*minac2* = *FALSE*," and "*adjust_pseudohaploid* = *FALSE*" (the latter 2 settings have their default values intended for high-quality diploid data). *QpAdm* tests were performed in *ADMIXTOOLS* v.2.0.0 under the default settings, except for the "*fudge_twice* = *TRUE*" setting designed to make numerical results more similar to those of classic *ADMIXTOOLS* (Patterson *et al.* 2012). Here, we describe 36 composite feasibility criteria (arranged from more to less stringent) tested on SSLs, which are variations on the criterion used for the work on AGS histories (Box 1):

1) Admixture fraction estimates (EAF) $\pm 2$ SE $\in (0, 1)$; *P*-value threshold for *n*-way models is either 0.05 or 0.5; all "trailing" simpler models (Box 1) are rejected according to the 0.001 *P*-value threshold.

2) EAF $\pm 2$ SE $\in (0, 1)$; *P*-value threshold for *n*-way models is 0.001, 0.01 (this version of the criterion matches the main version applied to AGS histories), 0.05, 0.1, or 0.5; all "trailing" simpler models are rejected according to the same *P*-value thresholds.

3) EAF $\in (0, 1)$; *P*-value threshold for *n*-way models is either 0.05 or 0.5; all "trailing" simpler models are rejected according to the 0.001 *P*-value threshold.

4) EAF $\in (0, 1)$; *P*-value threshold for *n*-way models is 0.001, 0.01, 0.05, 0.1, or 0.5; all "trailing" simpler models are rejected according to the same *P*-value thresholds.

5) EAF $\in (-0.3, 1.3)$; *P*-value threshold for *n*-way models is either 0.05 or 0.5; all "trailing" simpler models are rejected according to the 0.001 *P*-value threshold.

6) EAF $\in (-0.3, 1.3)$; *P*-value threshold for *n*-way models is 0.001, 0.01, 0.05, 0.1, or 0.5; all "trailing" simpler models are rejected according to the same *P*-value thresholds.

7) EAF $\pm 2$ SE $\in (0, 1)$; *P*-value threshold is 0.001, 0.01, 0.05, 0.1, or 0.5; "trailing" simpler models are not checked.

8) EAF $\in (0, 1)$; *P*-value threshold is 0.001, 0.01, 0.05, 0.1, or 0.5; "trailing" simpler models are not checked.

9) EAFs $\in (-0.3, 1.3)$; *P*-value threshold is 0.001, 0.01, 0.05, 0.1, or 0.5; "trailing" simpler models are not checked.

If *qpAdm* results were interpreted at the level of experiments, but not models, for declaring a positive result, it was required that all simpler models for the same target (with all possible source combinations) are rejected according to one of the feasibility criteria described above. Models at a complexity level where the first fitting model was found were binned into rejected (nonfitting) and nonrejected (fitting) and ranked by Euclidean distances to ideal symmetric models (of the same complexity) in the space of optimality metrics (Fig. 4). We defined "outcomes of experiments that would be misleading in practice" as satisfying the following condition at the model complexity level where the first fitting model is found

for the target: minimal Euclidean distance$_{\text{nonrejected}}$–minimal Euclidean distance$_{\text{rejected}}$ $> \sqrt{8}$, which is a diagonal of a right triangle with legs = 2 (corresponding to spatial distance of 2 or to angular distance of 40°, 26.6°, or 20° for 2-, 3-, and 4-way models, respectively). The reasoning behind this approach is as follows: we hope to accept models that are not much worse in spaces of optimality metrics than the models we reject. The spatial distance of 2 and the angular distance of 40° are substantial in the context of the simulated landscapes with maximal distance = 9 and maximal angle between sources in a 2-way model = 180°.

This criterion based on Euclidean distance to the ideal symmetric model was applied to any experiment producing 2-way and more complex fitting models, and in the case of experiments yielding 1-way fitting models the 2D space of model optimality metrics does not exist, and a simpler condition defined "bad" outcomes of experiments: *minimal ST distance in the nonrejected model group > minimal ST distance in the rejected model group*. For 2-way and more complex models with at least one source located at the same position as the target (distance = 0), min. STS angles were undefined, and thus distances in the space of min. STS angles and maximal or average ST distance were undefined for them too. Fitting models of this sort were counted toward positive results at a certain complexity level, but not toward experiments with misleading outcomes.

Forty randomized independent experiments progressing to 4-way admixture models (irrespective of model feasibility assessments) were performed per 1 simulation replicate, *qpAdm* protocol (distal and proximal, rotating, and nonrotating), and landscape sampling density (13 or 18 demes) (12,800 independent experiments in total). For the most realistic landscape type, that with per-generation gene flow intensities ranging from ca. $10^{-5}$ to $10^{-2}$, 160 independent experiments progressing to 3-way models were performed per 1 simulation replicate, *qpAdm* protocol, and landscape sampling density (12,800 experiments in total). For the 3 other landscape types, 40 independent experiments progressing to 3-way models were performed per 1 simulation replicate, *qpAdm* protocol, and landscape sampling density (9,600 experiments in total). The former set of experiments progressing to 4-way models was used for all the analyses in the study, and the latter 2 sets were employed as additional data for comparing estimated and predicted admixture fractions only (Supplementary Figs. 39 and 40).

Spearman's or Pearson's correlation coefficients and statistical significance of their difference from 0 were calculated in the spaces of EAF or SE vs model *P*-values (Supplementary Tables 1c, 7, 8, and 12), or in the spaces of optimality metrics (Supplementary Table 11), using the *cor.test* function from the R package *stats* v. 4.3.0. Two-sided significance tests were used: for Spearman's test, *P*-values were computed via the asymptotic *t* approximation and subjected to the Bonferroni correction. The same approaches were used for all applications of Spearman's (Supplementary Tables 1c, 7, 8, and 11–13) or Pearson's (Supplementary Figs. 14, 39, and 40 and Table 15) correlation coefficient in this study.

## SSL histories: *qpAdm* protocols with systematic landscape sampling

For *qpAdm* tests based on systematic landscape sampling, we used the landscapes (10 simulation iterations) with per-generation gene flow intensities ranging between approximately $10^{-5}$ and $10^{-2}$. On this landscape, we tested 2 protocols (Supplementary Fig. 16): distal rotating (in each experiment, a target was selected from demes sampled at "present", and 16 demes sampled 100 generations before "present" formed a rotating set) and distal nonrotating (in each experiment, a target was

selected from demes sampled at "present," 16 demes sampled 100 generations before "present" were used as proxy sources, and 16 demes sampled 300 generations before "present" were used as "right" groups).

Deme no. 32, occupying a central position on the landscape, was selected as a principal target. In the case of the rotating protocol, 16 demes forming a rotating set were selected exclusively from the 1st and 3rd circles of demes surrounding the target (Supplementary Fig. 16). We tested all rotating sets of this class including from 0 to 6 demes from the 1st circle (the nearest neighbors of the target deme). In total, 64 experiments were performed for each simulation replicate. Since there was only 1 experiment per simulation replicate with 0 or 6 demes from the 1st circle, we repeated experiments of this type (with 0 and 6 demes from the 1st circle) for 9 other target demes located in the center of the landscape (if it was possible to find a full 3rd circle for them). The same experiments were reproduced in the nonrotating version: all the demes from the 16-deme rotating sets were used as proxy sources, and the "right" set was identical in all the nonrotating experiments and included 16 demes from the 3rd circle (there were from 17 to 18 demes in that circle in total, depending on the target).

Above we described a "standard systematic setup" (presented or used for generating the results visualized in Figs. 4, 9 and 10 and Supplementary Figs. 16, 31, 32, 35, 37, and 39 and Tables 7–9); however, other configurations of demes were tested too: (1) 16 "right" groups coming from circles at different distances from the target and proxy sources in a distal nonrotating protocol coming alternatively from the 1st, 2nd, 3rd, or 4th circles around the target (Supplementary Fig. 29) and (2) 16 "right" groups coming from circles at different distances from the target and all 64 demes sampled at 100 generations ago included as alternative proxy sources in a distal nonrotating protocol (Supplementary Figs. 33, 34, and 36 and Table 12).

Before each *qpAdm* test, per-block $f_2$-statistics were calculated with *ADMIXTOOLS* v.2.0.0 (Maier *et al.* 2023) for the selected set of demes under the following settings: "*maxmiss* = 0," "*blgsize* = 4000000," "*minac2* = FALSE," and "*adjust_pseudohaploid* = FALSE.". *QpAdm* tests were performed under the default settings, except for the "*fudge_twice* = TRUE" setting designed to make numerical results more similar to those of classic *ADMIXTOOLS* (Patterson *et al.* 2012).

Systematic experiments progressed from simpler to more complex models (from 1- to 4-way models) until at least one feasible model was encountered. Unlike the randomized experiments which always included tests of all possible models from 1- to 4-way, systematic experiments were stopped and did not progress all the way to 4-way models if a simpler fitting model was encountered. The following composite feasibility criterion was used: EAF ∈ (0, 1); *P*-value threshold is 0.01; *P*-values of "trailing" models were not assessed since all simpler models for the target had to be rejected for an experiment to progress.

## Probability density curves for radiocarbon and calendar dates

Probability density curves for published radiocarbon and calendar dates were constructed in *OxCal v.4.4*. For calendar dates, we used the *C-Simulate* function in *OxCal v.4.4* for simulating normally distributed dating methods, taking the average calendar date as a median and the length of the timespan as a 95% confidence interval. For radiocarbon dates, we used calibration based on the IntCal20 calibration curve. Probability densities were summarized using the *Sum* function in *OxCal v.4.4* for each of the 3 groups of

individuals, those included in the "left," "right," and "target" population sets in at least one of the published *qpAdm* models (Narasimhan *et al.* 2019; Lazaridis *et al.* 2022) and then plotted together.

## Data availability

The Supplementary Materials for this paper are available at GSA FigShare: https://doi.org/10.25386/genetics.28334912 (the file includes detailed descriptions of all the materials, Supplementary Text 1–6, Supplementary Figs. 1–40, and Supplementary Tables 1–15). Principal software packages used in the study are *slendr v. 0.7.2.9000* (Petr *et al.* 2023; https://github.com/bodkan/slendr), *msprime v. 1.1.1* and *v. 1.2.0* (Baumdicker *et al.* 2022; https://github.com/tskit-dev/msprime), and *ADMIXTOOLS 2* (Maier *et al.* 2023; https://github.com/uqrmaie1/admixtools). Other codes for generating the simulated genetic data, running *qpAdm* protocols, and visualization are available at https://github.com/flegontovlab/qpadm_paper under the open-source MIT license (https://opensource.org/license/mit). Input data for all the scripts are available in the respective folders at https://github.com/flegontovlab/qpadm_paper, and larger input files (sets of *qpAdm* results on simulated data) are available at https://doi.org/10.5281/zenodo.14966948.

## Acknowledgments

Computational resources were provided by the e-INFRA CZ project (ID: 90254), supported by the Ministry of Education, Youth and Sports of the Czech Republic and by the ELIXIR-CZ project (ID: 90255), part of the international ELIXIR infrastructure.

## Funding

OF, PC, and PF were supported by the Czech Science Foundation (project no. 21-27624S). UI, JK, and LAV were supported by the Czech Ministry of Education, Youth and Sports (program ERC CZ, project no. LL2103). PF was supported by the Czech Ministry of Education, Youth and Sports ERD Funds (project OPVVV 16_013/0001775). PF and LAV were also supported by the John Templeton Foundation (grant no. 61220) and by a gift from Jean-Francois Clin. CDH and MPW were funded by the National Institutes of Health under award number R35GM146886.

## Conflicts of interest

The author(s) declare no conflicts of interest.

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

*Editor: N. Barton*