## [Peer Review File · Genetics]

Performance of qpAdm-based screens for genetic admixture on admixture-graph-shaped histories and stepping-stone landscapes

Olga Flegontova, Ulaş Işıldak, Eren Yüncü, Matthew Williams, Christian Huber, Jan Kočí, Leonid Vyazov, Piya Changmai, and Pavel Flegontov

NOTE: The reviews and decision letters are unedited and appear as submitted by the reviewers.

In extremely rare instances and as determined by a Senior Editor or the EIC, portions of a review may be redacted. If a review is signed, the reviewer has agreed to no longer remain anonymous.

The review history appears in chronological order.

Review Timeline:

Submission Date:	2023-10-18
Editorial Decision:	2024-02-09
Resubmission Received:	2024-08-22
Editorial Decision:	2024-12-18
Resubmission Received:	2025-02-03
Editorial Decision:	2025-02-22
Revision Received:	2025-03-08
Accepted:	2025-03-11

February 8, 2024

GENETICS-2023-306568

False discovery rates of qpAdm-based screens for genetic admixture

Dear Dr. Flegontov:

An expert in the field has reviewed your manuscript, and I have read it as well and discussed it with multiple colleagues on the GENETICS editorial board. We all agree your manuscript is timely and an important assessment of a popular method, qpAdm. While your manuscript is not currently acceptable for publication in GENETICS, we would welcome a substantially revised manuscript that addresses the reviewer's comments. You can read the review at the end of this email.

We look forward to receiving your revised manuscript. Please let the editorial office know approximately how long you expect to need for revisions.

Upon resubmission, please include:

1. A clean version of your manuscript;
2. A marked version of your manuscript in which you highlight significant revisions carried out in response to the major points raised by the editor/reviewers (track changes is acceptable if preferred);
3. A detailed response to the editor's/reviewers' feedback and to the concerns listed above. Please reference line numbers in this response to aid the editor and reviewers.

Your paper may be sent back out for review.

Additionally, please ensure that your resubmission is formatted for GENETICS
<https://academic.oup.com/genetics/pages/general-instructions>

Follow this link to submit the revised manuscript: Link Not Available

Sincerely,

Sohini Ramachandran
Associate Editor
GENETICS

Approved by:
Nicholas Barton
Senior Editor
GENETICS

Reviewer #1 (Comments for the Authors (Required)):

[I want to disclose that I also reviewed the companion qpAdm study by Williams et al. which is under revision at the same time. Given that both papers share co-authors and have thus likely been worked on in parallel, some of my comments will apply to both papers and might be present in both of my reviews.]

In the presented manuscript, the authors evaluate the limits of the popular qpAdm method (and various "inference protocols" utilising it) for inference of ancient gene-flow events. They investigate the ability of qpAdm to infer gene-flow parameters across a large number of randomly generated complex admixture graph-based demographic histories, evaluate under which conditions can qpAdm potentially mislead users, and provide a set of warnings, guidelines, and recommendations for future studies.

This work is very timely and important. The qpAdm method has established itself as a workhorse of aDNA-adjacent population genetic research and is becoming more and more popular as the aDNA field continues to grow. However, the scale of our understanding of the limits and applicability of qpAdm in various circumstances (particularly for complex models) has been severely lagging behind the scale of popularity in the field. The presented manuscript aims to do fix this problem and provides - at long last - an important evaluation of the limits and power of qpAdm from a rigorous population genetic/statistical perspective. I have no doubts the paper will be of a huge service to the entire aDNA community and will provide an important set of decision guidelines in future qpAdm applications.

I don't have critical comments w.r.t. methodology or statistical design of the study. It's clear that the authors have thought carefully about the work and that it stems from extensive practical experience with qpAdm. However, I have a few comments which I would like to see being addressed or implemented, intended to help with making the paper more digestible, easier interpret for the general audience of aDNA field, and easier to put into practice in future work.

First, I have to applaud the authors for the extremely clear and educational introduction section. The topic of f-statistics and related methodology has been extensively covered in great mathematical detail by two well-known studies (Patterson et al. 2012; Peter 2016). However, these two papers are not exactly an easy reading, especially for an average aDNA researcher who generally doesn't have extensive mathematical background. The introductory section is quite long, but is written so clearly that it might very well end up forming a classroom-level material on the topic in the future.

Major comments

=====

On lines 405-408, the authors write that they simulated genomes composed of three or ten 100 Mbp chromosomes (i.e. 300 Mb and 1000 Mb genomes total) across a certain number of replicates, with the idea of testing the qpAdm performance as a function of amounts of available data, particularly in the context of SNP capture missingness.

This made me think about jackknifing, significance of f-statistics (Z scores) and, by extension, also qpAdm "feasibility" (p-values and admixture proportions) depending on the sequence length: For instance, if I simulate replicates of a 1Mb, 10Mb, 100Mb chromosome and compute an f4 statistic, I will get f4 values which will be less noisy the more sequence is simulated (and more accurate admixture inferences). I would expect the same should apply even for qpAdm, with an increase in power with larger sequence lengths, purely by reducing statistical noise. In other words, simulating a ~1 million SNPs "capture set" by subsampling from different amounts simulated sequence (here 300 Mb, 1000 Mb vs potentially all autosomes) should result in that 1M SNPs with different linkage and (I presume) different significance levels of analyses from those simulations.

In order to make the results more comparable to the behavior of real data, wouldn't it be more accurate to simulate replicates from a single simulation of all 22 autosomes for each topology (about 3000 Mb genomes), and then perform subsampling to the desired numbers of SNPs (and even individual missingness) from those whole-genome simulations? Simulating 3x100Mb and 10x100Mb genomes separately seems a little arbitrary and it's not clear to me whether the f-statistics/qpAdm noisiness expected in real data will be comparable to these simulations.

If my reasoning above is not correct, I think it would be useful to describe the motivation for the more complex 3x100Mb and 100xMb simulations (and further subsampling to different SNP counts and missingness across those two sets), rather than simulating a truly whole-genome simulation of all autosomes and subsetting on those. After all, simulating all autosomes with msprime is no longer an issue, even for many replicate simulations.

Reading the result section titled "False positive models" I wondered if the extensive one-page descriptions of all of the seven ill-fitted models is absolutely necessary and if it won't be hard to digest it is for the reader. Perhaps this is compounded by the fact that those descriptions (currently spread over seven pages of continuous text!) are tricky to follow without carefully tracing the graphs plotted separately in the supplementary materials. I think this is a "less is more" situation, and that this result section would be easier to read if the authors compromised by selecting detailed descriptions of a few of the most instructive misfitted graphs to keep in the main text and accompany those with figures from the supplementary (i.e. Fig. S4a-...) directly in the main text. At the very least, each paragraph describing each individual misfitted model should be preceded by a header explaining the nature of the problem - one sentence describing what's wrong in the given graph that trips up qpAdm.

As a suggestion, perhaps the authors could select complex models comparable to each of the four simpler Models 1-4 in their Williams et al. companion paper which could allow useful comparisons between the conclusions reached by both manuscripts (after all, they are certainly intended to be read and learned from in tandem)?

lines 800-802: "We stress that whenever two-way models were feasible for targets modelled as more complex mixtures, they were nevertheless considered as TP in our subsequent analyses."

If I understood this sentence correctly, if a hypothetical target X would be simulated as mixture of three sources $X = A+B+C$ and qpAdm would judge a mixture such as $X=A+C$ as feasible, this would be considered as true positive match? Why should this be

a correct classification? It seems to me that this makes things a little easier on qpAdm than it should be? Why not consider inference of $X=A+B+C$ as TP and other, too simple models, not?

[My confusion stems from the assumption that analogously, in a simpler simulation case, if another hypothetical target Y would be simulated as $Y=A+B$ and a model $Y=B$ would be judged feasible by qpAdm, surely this would be considered an incorrect "fit"?]

On lines 778-779 the authors write "Finally, we consider examples of targets that were simulated as three-way mixtures, but for whom two-way qpAdm models were often feasible. In this case rejection of such a two-way model is considered a desired outcome".

Doesn't this contradict what's written on lines 800-802 (commented on just above) in terms of considering two-way models (from actual multi-way mixtures) as true positive? Perhaps a rephrasing would help clarify things.

lines 826-830: "[...] a technical limitation: the process of model classification into true and false ones cannot be fully automated since it requires careful interpretation of the topology and simulated admixture proportions."

Can the authors briefly elaborate in the text why this cannot be done from a technical perspective? At a casual glance after looking through the documentations of admixtools2 and msprime, it seems that:

a) Because admixtools2 function `random_admixturegraph` is used to generate random graph R objects to use as a basis for simulation, it should be possible to traverse this graph structure from a given target leaf "upwards" to find its sources?

Alternatively:

b) Any given msprime Demography Python object can be similarly converted to a graph data structure via its `to_demes` method, presumably allowing to do the same?

If this is truly impossible to do for technical reasons, perhaps the authors could briefly elaborate why.

lines 1245-1247: In point # 2 of best practices, the authors recommend using ADMIXTURE in tandem with qpAdm. However (as mentioned elsewhere in the manuscript), I don't think that ADMIXTURE is universally applicable for all kinds of temporal sampling structure or data quality (low coverage data, high missingness, etc.)?

In case that both methods can in fact handle equally poor data and this is already shown in the literature, it would be good to provide references. On the other hand, if ADMIXTURE is more demanding than qpAdm in terms of data quality needed, this should be also briefly discussed here as to avoid leading readers interested in improving the power of qpAdm via ADMIXTURE into potential dead ends.

I understand that ADMIXTOOLS2 has become increasingly more popular alternative to the original ADMIXTOOLS. However, I don't think ADMIXTOOLS2 is a true replacement to the original, as there are studies being published which still use the original C implementation which is continuing to be maintained. Do both versions give completely numerically identical results? I think a simple scatterplot-like comparison of distributions of p-values and admixture proportions computed by ADMIXTOOLS2 vs ADMIXTOOLS across a series of random simulations of eigenstrat data would be helpful. That way readers still using "old" ADMIXTOOLS will know they can transfer lessons from this manuscript to their workflows.

lines 462-463: They authors say they focused on one- and two-way admixture models. Given the limits on qpAdm accuracy on complex described in the manuscript, can they make a recommendation with regards to modelling even more complex mixtures such as three-way admixture models? My understanding of the situation after reading the manuscript is that this should perhaps be entirely avoided unless the underlying graph model topological structure can be reliably established, ideally even proven via explicit verification of the assumed graph models against empirical f-statistics via simulation (or at least showing that the underlying assumed graph cannot be rejected with qpAdm or similar). I think this point deserves an item in the manuscript's best practice list.

The authors investigated models which only involve "pulse" admixture events. I agree that it's appropriate to focus on simpler models in this study, but I also think it's fair to say that it's quite likely that most admixture events in human history occurred over extended periods of time. In order to help extrapolate the results presented in this manuscript even closer to real-world scenarios and data, it might be good to compare the power of qpAdm inference for a series of pairs of random models in which one model in the pair would contain a pulse-like admixture and another model would feature admixture spread over a longer timespan (at least admixtures into the target lineage). Perhaps a simple scatterplot showing p-values and admixture weights from qpAdm inferences between both models in the pair across the series of, say, 50 or 100 simulations would be sufficient? Does non-pulse admixture simply increase a qpAdm inference noise?; does it lead to some kind of bias?; does the number of feasible models decrease? should empirical qpAdm results be interpreted with even more caution than based on the conclusions presented here? - these are all questions that readers could immediately get insight into if this analysis is presented.

The authors recommend running ADMIXTURE to increase reliability of qpAdm analyses. The companion paper by Williams et al. recommends computing f3 as a confirmation and ranking tool for plausible qpAdm models. Should both be used where applicable? How could increasing qpAdm power with f3 help compare to using ADMIXTURE and vice versa? A quick note on this in the discussion would be helpful.

line 1384: "4-Mbp-sized genome blocks"

How does the size of genome blocks used for jackknifing affect the results of a qpAdm analysis? If admixture is recent, ancestry haplotypes (and extend of linkage between admixed SNPs in a target) will be larger, so blocks overlapping those haplotypes will be entirely formed by SNPs linked on those haplotypes. If admixture is ancient, ancestry haplotypes will be shorter, which will also show at the level of blocks (depending on the block sizes). As such, does the block size have any influence on qpAdm inferences, depending on the age of admixture being modelled? If there is such an effect, perhaps showing a simple figure demonstrating p-values and inferred admixture proportions as a function of block size would be useful. If there's not an effect and this is of no concern at all, a reference to the relevant literature would be helpful.

Minor comments

=====

Is the extensive description of the setups of two qpAdm rotating protocols (Narasimhan et al. 2019; Lazaridis et al. 2022) in the first "Results" section titled "An overview of published rotating" (line 262 and below) really required? I'm talking about the exhaustive descriptions listing the 15+ populations used by each of the papers, etc. The section is under the "Results" heading but doesn't actually contain any novel results, as far as I can tell? I really appreciate the figure with age distributions of samples from the two example papers (i.e., Figure 2), but I think the figure could very well stay in the main text as it is (because it is very instructive!) and simply be referred to from the relevant places in the introduction which discuss the qpAdm rotating protocols. The summary of setups used by the two example studies could then be moved to the supplement because it doesn't seem like it adds much to the main text itself and isn't, in fact, referred to in a significant way. As an added benefit, this would make the manuscript more focused with a more concentrated emphasis on the important results and lessons because the paper is already quite long.

lines 723-725: "[...] not satisfying the topological criteria of false models listed above."

Given that this sentence is on page 30, it was a bit hard on first reading to understand what is meant by "above". The authors might consider writing a short reminder on the criteria mentioned (not in full, of course), perhaps in parenthesis after the word "above" to help readers with scanning the paper for details.

line 93 - "to prove that a certain cline spotted in PC space is a result of"
I would avoid the word "prove" here and rather write something like "to test for the possibility that [...] might be a result of".

line 1267: I'm not sure if the paragraph numbered # 6 belongs as an item in the list of best practices? At least reading this paragraph doesn't really seem to contain any best practice and it looks more like a formatting issue.

Reviewer #1 (Comments for the Authors):

[I want to disclose that I also reviewed the companion qpAdm study by Williams et al. which is under revision at the same time. Given that both papers share co-authors and have thus likely been worked on in parallel, some of my comments will apply to both papers and might be present in both of my reviews.]

In the presented manuscript, the authors evaluate the limits of the popular qpAdm method (and various "inference protocols" utilising it) for inference of ancient gene-flow events. They investigate the ability of qpAdm to infer gene-flow parameters across a large number of randomly generated complex admixture graph-based demographic histories, evaluate under which conditions can qpAdm potentially mislead users, and provide a set of warnings, guidelines, and recommendations for future studies.

This work is very timely and important. The qpAdm method has established itself as a workhorse of aDNA-adjacent population genetic research and is becoming more and more popular as the aDNA field continues to grow. However, the scale of our understanding of the limits and applicability of qpAdm in various circumstances (particularly for complex models) has been severely lagging behind the scale of popularity in the field. The presented manuscript aims to do fix this problem and provides - at long last - an important evaluation of the limits and power of qpAdm from a rigorous population genetic/statistical perspective. I have no doubts the paper will be of a huge service to the entire aDNA community and will provide an important set of decision guidelines in future qpAdm applications.

I don't have critical comments w.r.t. methodology or statistical design of the study. It's clear that the authors have thought carefully about the work and that it stems from extensive practical experience with qpAdm. However, I have a few comments which I would like to see being addressed or implemented, intended to help with making the paper more digestible, easier interpret for the general audience of aDNA field, and easier to put into practice in future work.

First, I have to applaud the authors for the extremely clear and educational introduction section. The topic of f -statistics and related methodology has been extensively covered in great mathematical detail by two well-known studies (Patterson et al. 2012; Peter 2016). However,

these two papers are not exactly an easy reading, especially for an average aDNA researcher who generally doesn't have extensive mathematical background. The introductory section is quite long, but is written so clearly that it might very well end up forming a classroom-level material on the topic in the future.

We appreciate the positive opinion on our work, and we have completely revamped Results and Discussion in this revision to address the reviewer's comments and to make the paper even more useful in guiding future work in archaeogenetics. To reflect the broadened scope of this revision, we have renamed the paper from "False discovery rates of qpAdm-based screens for genetic admixture" to "Performance of qpAdm-based screens for genetic admixture on admixture-graph-shaped histories and stepping-stone landscapes".

Major comments

=====

On lines 405-408, the authors write that they simulated genomes composed of three or ten 100 Mbp chromosomes (i.e. 300 Mb and 1000 Mb genomes total) across a certain number of replicates, with the idea of testing the qpAdm performance as a function of amounts of available data, particularity in the context of SNP capture missingness.

This made me think about jackknifing, significance of f-statistics (Z scores) and, by extension, also qpAdm "feasibility" (p-values and admixture proportions) depending on the sequence length: For instance, if I simulate replicates of a 1Mb, 10Mb, 100Mb chromosome and compute an f4 statistic, I will get f4 values which will be less noisy the more sequence is simulated (and more accurate admixture inferences). I would expect the same should apply even for qpAdm, with an increase in power with larger sequence lengths, purely by reducing statistical noise. In other words, simulating a ~1 million SNPs "capture set" by subsampling from different amounts simulated sequence (here 300 Mb, 1000 Mb vs potentially all autosomes) should result in that 1M SNPs with different linkage and (I presume) different significance levels of analyses from those simulations.

In order to make the results more comparable to the behavior of real data, wouldn't it be more accurate to simulate replicates from a single simulation of all 22 autosomes for each topology (about 3000 Mb genomes), and then perform subsampling to the desired numbers of SNPs (and even individual missingness) from those whole-genome simulations? Simulating 3x100Mb and

10x100Mb genomes separately seems a little arbitrary and it's not clear to me whether the f-statistics/qpAdm noisiness expected in real data will be comparable to these simulations.

If my reasoning above is not correct, I think it would be useful to describe the motivation for the more complex 3x100Mb and 100xMb simulations (and further subsampling to different SNP counts and missingness across those two sets), rather than simulating a truly whole-genome simulation of all autosomes and subsetting on those. After all, simulating all autosomes with msprime is no longer an issue, even for many replicate simulations.

We agree with the reviewer that for comparing our results with the archaeogenetic literature it is better to simulate 3000Mb-sized genomes, and we did that for this revision. We have repeated all the core analyses (based on admixture-graph-shaped simulations) on 300Mb-, 1000Mb-, and 3000Mb-sized genomes, and now we highlight the 3,000-Mbp simulations in Figures 5 and 6 and in various supplemental figures. We have also increased the time depth of the 3,000-Mbp simulations from 800 to 3000 generations in order to match F_{ST} values typical for humans (Fig. S3). Some results such as the case studies in Suppl. text 2 and explorations of PCA and ADMIXTURE in Suppl. texts 3 and 4 were not updated on the latest 3,000-Mbp simulations, and we explicitly mention in the text if that is the case. We hope this is not a big problem since now the main text is not focused on these lines of analysis.

Since the jackknifing procedure implemented at the f_4 -statistic calculation step “takes care” of non-independent SNPs, we think that subsetting outputs of each simulation to the same number of sites is not necessary. Random subsetting and non-random ascertainment may introduce their own problems which we explored in another study (Flegontov et al. 2023, PLoS Genetics).

Simulating 3x100Mb and 10x100Mb genomes separately seems a little arbitrary and it's not clear to me whether the f-statistics/qpAdm noisiness expected in real data will be comparable to these simulations.

We also did not intend to reproduce the data quality and features typical for real data very closely: this task is just very hard due to the large number of variables involved and problems in estimating these variables on real data. Instead, we wanted to investigate performance trends with increasing amounts of data and at drastically different quality levels (no missing data vs. 5% to 95% missing data per individual), and for that 10x difference in the amount of data is enough, in our opinion. We address these caveats in the last paragraphs of Suppl. text 4 where a part of the AGS-focused discussion was moved.

Reading the result section titled "False positive models" I wondered if the extensive one-page descriptions of all of the seven ill-fitted models is absolutely necessary and if it won't be hard to digest it is for the reader. Perhaps this is compounded by the fact that those descriptions (currently spread over seven pages of continuous text!) are tricky to follow without carefully tracing the graphs plotted separately in the supplementary materials. I think this is a "less is more" situation, and that this result section would be easier to read if the authors compromised by selecting detailed descriptions of a few of the most instructive misfitted graphs to keep in the main text and accompany those with figures from the supplementary (i.e. Fig. S4a-...) directly in the main text. At the very least, each paragraph describing each individual misfitted model should be preceded by a header explaining the nature of the problem - one sentence describing what's wrong in the given graph that trips up qpAdm.

As a suggestion, perhaps the authors could select complex models comparable to each of the four simpler Models 1-4 in their Williams et al. companion paper which could allow useful comparisons between the conclusions reached by both manuscripts (after all, they are certainly intended to be read and learned from in tandem)?

We agree that the seven-page descriptions of various admixture graph topologies leading to problematic (or problem-free) qpAdm results (the section "Case studies illustrating false and true feasible qpAdm models", sub-sections "False positive models" and "True positive models") are probably out of place in the main text. We have moved all the FP and TP case studies, except for the one presented in Fig. 4, to Suppl. text 2. We did that to also free up space in the main text for the new content on stepping-stone landscapes. Because of that lack of space in the main text and the changed focus of the paper, we also chose not to discuss our randomized simulations similar to models 1-4 from Williams et al. 2024.

lines 800-802: "We stress that whenever two-way models were feasible for targets modelled as more complex mixtures, they were nevertheless considered as TP in our subsequent analyses."

If I understood this sentence correctly, if a hypothetical target X would be simulated as mixture of three sources $X = A+B+C$ and qpAdm would judge a mixture such as $X=A+C$ as feasible, this would be considered as true positive match? Why should this be a correct classification? It seems to me that this makes things a little easier on qpAdm than it should be? Why not consider inference of $X=A+B+C$ as TP and other, too simple models, not?

[My confusion stems from the assumption that analogously, in a simpler simulation case, if another hypothetical target Y would be simulated as $Y=A+B$ and a model $Y=B$ would be judged feasible by qpAdm, surely this would be considered an incorrect "fit"?]

Stricter definitions of TP models are possible, but we applied this lenient approach since in the archaeogenetic literature sampling of ancient individuals is often poor, and thus the task is not to find all groups which participated in the history of a given target group, but to identify at least predominant ancestry sources with some degree of precision. Likewise, in our admixture-graph-shaped simulated histories sampled at branch tips, many groups that could act as proxy sources for ancient lineages were not sampled. We have added this clarification to the text on lines 505-510. As also stated in the Discussion (lines 1621-1626), we were interested explicitly in false detections of gene-flow, but not in false omissions of detection.

On lines 778-779 the authors write "Finally, we consider examples of targets that were simulated as three-way mixtures, but for whom two-way qpAdm models were often feasible. In this case rejection of such a two-way model is considered a desired outcome".

Doesn't this contradict what's written on lines 800-802 (commented on just above) in terms of considering two-way models (from actual multi-way mixtures) as true positive? Perhaps a rephrasing would help clarify things.

Indeed, there is some contradiction here, and we have changed "desired outcome" to "ideal outcome" (see line 290 in Suppl. text 1).

lines 826-830: "[...] a technical limitation: the process of model classification into true and false ones cannot be fully automated since it requires careful interpretation of the topology and simulated admixture proportions."

Can the authors briefly elaborate in the text why this cannot be done from a technical perspective? At a casual glance after looking through the documentations of admixtools2 and msprime, it seems that:

a) Because admixtools2 function random_admixturegraph is used to generate random graph R objects to use as a basis for simulation, it should be possible to traverse this graph structure from a given target leaf "upwards" to find its sources? Alternatively:

b) Any given msprime Demography Python object can be similarly converted to a graph data structure via its to_demes method, presumably allowing to do the same?

If this is truly impossible to do for technical reasons, perhaps the authors could briefly elaborate why.

The language we used ("cannot be automated") is indeed too strong, and we softened it by saying that automation is difficult (lines 710 and 1634). We plan to add various automated analyses of admixture graph topologies to the ADMIXTOOLS 2 package in the future. Given that the scope of the paper now includes both admixture-graph like histories (where classification of qpAdm models into false and true is hard but possible) and stepping-stone landscapes where such a binary classification is impossible in principle, we believe that investing too much time in the automation task is unwarranted.

lines 1245-1247: In point # 2 of best practices, the authors recommend using ADMIXTURE in tandem with qpAdm. However (as mentioned elsewhere in the manuscript), I don't think that ADMIXTURE is universally applicable for all kinds of temporal sampling structure or data quality (low coverage data, high missingness, etc.)?

In case that both methods can in fact handle equally poor data and this is already shown in the literature, it would be good to provide references. On the other hand, if ADMIXTURE is more demanding than qpAdm in terms of data quality needed, this should be also briefly discussed here as to avoid leading readers interested in improving the power of qpAdm via ADMIXTURE into potential dead ends.

Indeed, we have not tested the performance of ADMIXTURE on all types of data in this study (low and high quality, admixture-graph-shaped histories and stepping-stone landscapes), and thus our recommendation was rephrased as a tentative suggestion (lines 1713-1717), and all the results on the "admixture screening pipelines" combining qpAdm with PCA and/or ADMIXTURE were moved to Suppl. text 4 to save space in the main text for the results based on simulated stepping-stone landscapes (see below)

I understand that ADMIXTOOLS2 has become increasingly more popular alternative to the original ADMIXTOOLS. However, I don't think ADMIXTOOLS2 is a true replacement to the original, as there are studies being published which still use the original C implementation which is continuing to be maintained. Do both versions give completely numerically identical results? I think a simple scatterplot-like comparison of distributions of p-values and admixture proportions computed by ADMIXTOOLS2 vs ADMIXTOOLS across a series of random simulations of eigenstrat data would be helpful. That way readers still using "old" ADMIXTOOLS will know they can transfer lessons from this manuscript to their workflows.

We thank the reviewer for this suggestion, and we have added a small table (Table S14) comparing the numerical results of ADMIXTOOLS 2 vs. ADMIXTOOLS on noisy data from the latest 3,000-Mbp admixture-graph shaped simulations to the Methods section. Thus, we explored the most realistic AGS simulations available to us. The results are not identical, but very similar: Pearson's $r = 0.99$ for non-logarithmic p-values (this analysis prioritizes high p-values) and 0.91 for logarithmic estimated admixture fractions, and lower for logarithmic SE and logarithmic p-values ($0.7 - 0.8$). As we show elsewhere, p-values reported by ADMIXTOOLS 2 are in general very variable from simulation to simulation and from one data subsampling replicate to another (Fig. S7).

lines 462-463: The authors say they focused on one- and two-way admixture models. Given the limits on qpAdm accuracy on complex graphs described in the manuscript, can they make a recommendation with regards to modelling even more complex mixtures such as three-way admixture models? My understanding of the situation after reading the manuscript is that this should perhaps be entirely avoided unless the underlying graph model topological structure can be reliably established, ideally even proven via explicit verification of the assumed graph models against empirical f-statistics via simulation (or at least showing that the underlying assumed graph cannot be rejected with qpAdm or similar). I think this point deserves an item in the manuscript's best practice list.

As we discuss in the revised manuscript, exploring three-way and more complex models is likely more risky for AGS and definitely less accurate for SSL simulations than focusing on two-way models only (lines 1776-1793), but we actually recommend exploring a range of model complexities for each target and not stopping at the simplest fitting model in the latter case since gene-flow networks are much more complex on SSL. In general, we have rewritten the Discussion and removed the "Best practices" section, placing the few advice we give in the Discussion.

The authors investigated models which only involve "pulse" admixture events. I agree that it's appropriate to focus on simpler models in this study, but I also think it's fair to say that it's quite likely that most admixture events in human history occurred over extended periods of time. In order to help extrapolate the results presented in this manuscript even closer to real-world scenarios and data, it might be good to compare the power of qpAdm inference for a series of pairs of random models in which one model in the pair would contain a pulse-like admixture and another model would feature admixture spread over a longer timespan (at least admixtures into the target lineage). Perhaps a simple scatterplot showing p-values and admixture weights from qpAdm inferences between both models in the pair across the series of, say, 50 or 100 simulations would be sufficient? Does non-pulse admixture simply increase a qpAdm inference noise?; does it lead to some kind of bias?; does the number of feasible models decrease? should empirical qpAdm results be interpreted with even more caution than based on the conclusions presented here? - these are all questions that readers could immediately get insight into if this analysis is presented.

This suggestion is in the spirit of our ongoing research program where we explore performance of the "standard archaeogenetic toolkit" (PCA, ADMIXTURE, f-statistics, qpAdm, admixture graphs) on "isolation-by-distance landscapes" approximated by two-dimensional stepping-stone simulations with non-uniform randomized gene flow intensities. That is why we decided to restructure our manuscript substantially to add a series of results on qpAdm applied to such "genetic landscapes". Since all demes on genetic landscapes are connected by gene flows decaying with distance, classifying admixture models into strictly false and true is impossible, and instead we assessed if qpAdm enriches the set of feasible models in demes close to the target and/or arranged symmetrically with respect to the target. Ideally, for ranking proxy sources we should have used clouds of ancestor density for targets (Grundler et al. 2024) instead of spatial source-target distances and angles, as discussed in the text (lines 843-859, 1661-1674), but we'd prefer to leave this more precise and technically challenging analysis for another study.

We explored both systematically and randomly picked sets of reference demes and targets, qpAdm model complexities from 1 to 6 sources, four kinds of qpAdm protocols (proximal and distal, rotating and non-rotating), four gene-flow intensity ranges and two deme sampling densities on the landscapes, and 36 sets of qpAdm model feasibility criteria. Thus, our exploration of the space of simulations and qpAdm protocol parameters is deeper on genetic landscapes than on graph-shaped simulations. This was possible simply because manual analysis of admixture graph topologies and classification of models into true and false was not needed. However, to keep the manuscript reasonably short, we did not explore the performance of PCA, ADMIXTURE, and multi-method admixture-inference pipelines on genetic landscapes (conversely, these methods were tested on AGS simulations, now presented in the supplement).

Of course, we can imagine a continuum between admixture-graph-shaped histories and completely uniform stepping-stone landscapes, but exploring many points on this continuum is unfeasible. That is why we chose to compare performance of qpAdm on these two extremes and did not explore slightly modified admixture graphs where some gene flows are protracted. Moreover, qpAdm performance in the case of an admixture-graph-shaped history with a protracted gene flow was already explored by Harney et al. (2021). To keep this manuscript focused, we prefer not to make strong judgements about usefulness of the admixture graph and genetic landscape paradigms for research on particular species such as humans, although it can be understood from the manuscript that we prefer the latter paradigm now. We prefer just to present these alternative paradigms and warn about implications of the uncertainty about the most appropriate paradigm for design of qpAdm protocols and interpretation of their results.

Due to the inclusion of results on genetic landscapes, substantial modification of the manuscript layout was done: many parts of Results and Discussion on admixture graph histories were moved to SI, and the remaining core analyses were largely re-written to harmonize the presentation in the AGS and SSL sections. We also analyzed from the SSL perspective a set of ~420,000 published qpAdm models based on real archaeogenetic data (Zeng et al. 2023) and interpreted the resulting patterns using insights from the extensive exploration of simulated SSL data. We hope that the new manuscript will be even more useful for the archaeogenetic community, although its main message is more pessimistic: temporal stratification delivers relatively accurate models on AGS histories (at least in the case of simple two-way models), but we are rarely sure an AG is an adequate historical model. On SSL, in certain conditions accurate two-way models are achievable too, but in this case, they are simplistic too much, and complex qpAdm models found on SSL are largely inaccurate, as we show.

The authors recommend running ADMIXTURE to increase reliability of qpAdm analyses. The companion paper by Williams et al. recommends computing f3 as a confirmation and ranking tool for plausible qpAdm models. Should both be used where applicable? How could increasing qpAdm power with f3 help compare to using ADMIXTURE and vice versa? A quick note on this in the discussion would be helpful.

Since the scope of the paper was broadened and the ADMIXTURE/PCA/f3 results were moved to SI, we decided not to add this discussion and additional analyses.

line 1384: "4-Mbp-sized genome blocks"

How does the size of genome blocks used for jackknifing affect the results of a qpAdm analysis? If admixture is recent, ancestry haplotypes (and extent of linkage between admixed SNPs in a target)

will be larger, so blocks overlapping those haplotypes will be entirely formed by SNPs linked on those haplotypes. If admixture is ancient, ancestry haplotypes will be shorter, which will also show at the level of blocks (depending on the block sizes). As such, does the block size have any influence on qpAdm inferences, depending on the age of admixture being modelled? If there is such an effect, perhaps showing a simple figure demonstrating p-values and inferred admixture proportions as a function of block size would be useful. If there's not an effect and this is of no concern at all, a reference to the relevant literature would be helpful.

The influence of block size on qpAdm performance was quite thoroughly explored by Harney et al. (2021): “These results are consistent with theoretical expectations, as we expect that when the block size is too small there will be correlation between SNPs across different blocks that is uncorrected. Conversely, when the block size is too large, the standard error of the f4 statistics used in qpAdm calculations may be poorly estimated. Despite the observation of biased P-value distributions, qpAdm appears relatively robust to the selected block jackknife size, as biases were only observed in cases where the block size was either 50x smaller or 20x larger than the default block jackknife size.”

Since the default block size in qpAdm is 0.05 Morgans = 5 cM or ~5 Mbp in humans (and since we used a recombination rate similar to that estimated for humans), we believe that the 4 Mbp block size we have chosen is well within the range of robust performance. In this revision, we have cited these results by Harney et al. (2021) in the Methods section, see lines 1995-1998.

Minor comments

=====

Is the extensive description of the setups of two qpAdm rotating protocols (Narasimhan et al. 2019; Lazaridis et al. 2022) in the first "Results" section titled "An overview of published rotating" (line 262 and below) really required? I'm talking about the exhaustive descriptions listing the 15+ populations used by each of the papers, etc. The section is under the "Results" heading but doesn't actually contain any novel results, as far as I can tell? I really appreciate the figure with age distributions of samples from the two example papers (i.e., Figure 2), but I think the figure could very well stay in the main text as it is (because it is very instructive!) and simply be referred to from the relevant places in the introduction which discuss the qpAdm rotating protocols. The summary of setups used by the two example studies could then be moved to the supplement because it doesn't seem like it adds much to the main text itself and isn't, in fact, referred to in a significant way. As an added benefit, this would make the manuscript more focused with a more concentrated emphasis on the important results and lessons because the paper is already quite long.

Following this suggestion, we have moved the whole section “An overview of published rotating and model competition qpAdm protocols” to Suppl. text 1 and kept Fig. 2 in the main text.

lines 723-725: "[...] not satisfying the topological criteria of false models listed above."

Given that this sentence is on page 30, it was a bit hard on first reading to understand what is meant by "above". The authors might consider writing a short reminder on the criteria mentioned (not in full, of course), perhaps in parenthesis after the word "above" to help readers with scanning the paper for details.

Since, following another suggestion, we have moved to the supplement nearly all the text separating the list on lines 484-495 and this reference to it on line 503, repeating the list here is no longer necessary.

line 93 - "to prove that a certain cline spotted in PC space is a result of"

I would avoid the word "prove" here and rather write something like "to test for the possibility that [...] might be a result of".

We have rephrased as suggested, see lines 101-104.

line 1267: I'm not sure if the paragraph numbered # 6 belongs as an item in the list of best practices? At least reading this paragraph doesn't really seem to contain any best practice and it looks more like a formatting issue.

Indeed, paragraph #6 was included in the best practices list by mistake, and now the whole Discussion section has been restructured.

December 18, 2024

GENETICS-2024-307000

Performance of qpAdm-based screens for genetic admixture on admixture-graph-shaped histories and stepping-stone landscapes

Dear Dr. Flegontov:

I apologise for my slow handling of your paper - the original reviewer was happy with your response to his concerns, but did not feel able to review the substantial amount of new material. So, we needed to find a new reviewer, who is very positive. I agree with the reviewer that it is crucial that statistical methods are properly validated, especially when they have become widely used.

I am pleased to inform you that, with some revisions, your work is potentially suitable for publication in GENETICS. I do not expect to send the revision out for review. The reviewer has comments and concerns that need to be addressed in a revised manuscript. You can read their review at the end of this email.

It is most important that you address the following in your resubmission:

- the 'pruning' of alleles in LD needs to be justified, since such associations surely include signals of admixture (L2122).
- The description of the EAF estimator and the spatial lattice need to be clarified.

Most important, the MS is exceptionally long. This is to some degree necessary, because it examines the performance of qpADM against a wide range of models. However, as well as being long, the paper is quite demanding to read. When you revise, please try to give the reader a clear path through the MS, and consider which material is essential for the main text, or could instead be relegated to supplementary material. Having said this, I do realize that much of the material is necessary, and should be in the main text - but it should nevertheless be possible to make your work more accessible.

We look forward to receiving your revised manuscript. Please let the editorial office know approximately how long you expect to need for revisions.

Upon resubmission, please include:

1. A clean version of your manuscript;
2. A marked version of your manuscript in which you highlight significant revisions carried out in response to the major points raised by the editor/reviewers (track changes is acceptable if preferred);
3. A detailed response to the editor's/reviewers' comments and to the concerns listed above. Please reference line numbers in this response to aid the editors.

Additionally, please ensure that your resubmission is formatted for GENETICS.

<https://academic.oup.com/genetics/pages/general-instructions>

Follow this link to submit the revised manuscript: Link Not Available

Sincerely,

Nick Barton
Senior Editor
GENETICS

Approved by:
Howard Lipshitz
Editor in Chief
GENETICS

Reviewer #2 :

General comments

The work is timely and important. This kind of analysis is extremely work intensive and the results are complex to express. I am reminded of the Nested Clade Analysis (NCA) debacle. Two research groups spent years of effort to measure NCA error rates (Beaumont and Panchal 2008, Knowles 2008) because an untested method had gone viral. That effort was greatly hampered

because the NCA decision process could not be automated (cf L707-708 "this frequency of assumption violations was not quantified in our study since it requires another layer of manual topological analysis.").

I congratulate the authors on having done a good job of expressing complex results. I have a long list of very minor typos and suggestions for wording, but no issues sufficiently important that they should slow the publication of the work.

There are, however (as always) a few issues which the authors should try to resolve before publication, and that should not take much effort.

- 1) The description of the spatial lattice needs clarified
- 2) The description of the qpAdm EAF estimator needs clarified
- 3) There is one passage (L1804-1811) which appears self-contradictory.

These first two crop up several times in the minor issues below.

Finally, in the Methods(L2122,L2131) it would be good to have some justification for data reduction by removal of sites found to be in admixture linkage disequilibrium, rather than data reduction by random subsetting. Admixture generates linkage disequilibrium genome-wide, and so it would seem this data reduction step is preferentially removing signal that should be available for inference. As with the contradictory passage, these are elaborated on below.

Stuart J.E. Baird
Studenec 18th November 2024

Minor issues

L36

"In this situation only a specific combination of landscape properties and feasibility criteria allows to efficiently reject highly asymmetric non-optimal models most abundant in random deme sets." ->

"In this situation only a specific combination of landscape properties and feasibility criteria allows highly asymmetric non-optimal models most abundant in random deme sets to be efficiently rejected."

["allows to" is missing a noun phrase before the infinitive phrase]

L54

"only a few studies were devoted to testing" ->
"only a few studies have been devoted to testing"

L116

"interpreting deviations of D- and f4-statistics from 0 (or lack thereof) becomes hardly possible if both branch pairs are connected by detectable gene flows, a typical situation on isolation-by-distance landscapes"

[Here "hardly possible" is unclear. "Almost impossible" is different from "possible, given hard work"]

Box1

SSL entry:

"here we describe an implementation used in this study. A stepping-stone landscape is originally "unfolded" from a single founder deme via a serial founder process that can be represented by a bifurcating tree (Estavoyer and Francois 2022), or via multifurcation, i.e. star radiation. This stage is followed by a gene-flow era represented as an undirected graph connecting demes, where each edge represents a bidirectional gene flow, and gene-flow intensities are allowed to differ in the forward and reverse directions. Gene flows on this undirected graph happen in one or more epochs, with unidirectional gene-flow intensities sampled randomly from uniform or non-uniform distributions. In this study all landscapes are based on the triangular lattice, and node degree varies from 3 to 6."

See also Methods LL2173-2174

"The demes were arranged on a triangular lattice, forming an approximately circular landscape, where each deme was connected by bidirectional gene flows to 3 to 6 neighboring demes (Fig. 7)."

I am familiar with modelling geneflow on the triangular lattice (Baird and Santos 2010): it seems a strange choice here because indexing of the graph connecting demes in the gene flow era must be more complex than indexing of a traditional square lattice. The reader needs a line explaining why a triangular lattice was chosen, and how nodes of degree higher than 3 are connected (assuming degree 3 is nearest-neighbour). This last needs explanation especially as L844-845 "gene flows between non-neighboring demes were not allowed"... yet in the traditional notion of a triangular tiling of the plane, a node can only have 3 direct neighbours.

Box1

proxy (ancestry) source entry:

"assumed to be cladal with one of true ancestry sources"->

"assumed to be cladal with one of the true ancestry sources"

L205

"This new type of qpAdm protocols, termed "rotating" protocol"->

"This new type of qpAdm protocol, termed the "rotating" protocol"

L222-226

"As an additional criterion of a fitting model, all inferred admixture proportions (see[...]), or proportions {plus minus} 2 standard errors (Narasimhan et al. 2019), may be required to lie between 0 and 1."

[This reads strangely. To require a proportion to lie between 0 and 1 is not a restriction. Further: what admixture proportion is returned in the case of no information? When software is required to return an estimate irrespective of information available, the default is to return a mid-interval estimate: 1/2. For example, this is the case for STRUCTURE K=2 q estimates: Intermediate and Unsupported estimates can only be distinguished by the confidence intervals STRUCTURE places round them. In the no information case the confidence interval is 0-1. This suggests only the (Narasimhan et al. 2019) approach above has any hope of ensuring there is support for admixture].

L271

"a particular type of SNP ascertainment: selecting sites" ->

"a particular type of SNP ascertainment bias: selecting sites"

L287-290

"Finding a feasible two-way or more complex admixture model for a target is often interpreted as solid evidence for gene flow, especially if PCA and ADMIXTURE methods confirm the same signal. Thus, qpAdm protocols are used in fact as formal tests for admixture, whereas the latter two methods are not formal tests."

[Given the rather arbitrary and adhoc test criteria that have been described (egg L222-226 comment above), it seems a stretch to suggest qpAdm can be used to make a formal test... I note that your following paragraph goes no to say essentiall this. Perhaps change the above text to: "... are used as supposed formal tests..."]

L330

"Below we explore on simulated AGS histories"->

"Below we explore simulated AGS histories"

L352

"where target group's sampling date"->

"where the target group's sampling date"

L397

"simulated graph six most ancient groups were selected"->

"simulated graph the six most ancient groups were selected"

L522-252

"admixture fractions estimated by qpAdm (abbreviated as EAF) [...]. Since qpAdm EAF vary from <<-1 to >>1...."

[This needs explained: how can an admixture proportion (by definition on (0,1) vary from <<-1 to >>1 ? To interpret the work done here, the reader needs to know what is actually being discussed]

L619

"in target's history"->

"in a target's history"

L625-627

"occasionally fitting and false is significant too (Table S1b). Statistically significant trends in the same direction are observed for distal models too, but in this case median"->

"occasionally fitting and false is also significant (Table S1b). Statistically significant trends in the same direction are observed for distal models aswell, but in this case median"

[too many toos]

L675-678

"A large fraction of false proximal models emerges as fitting due to false rejections of one-way models because of violations of the topological assumptions (Fig. 1), for instance, due to "left-to-right" gene flows, and that prompts the investigator to test more complex two-way models,"

"and that prompts the investigator" ->

"and these prompt the investigator"

[false rejections (plural) prompt the investigator]

L824

"but leave other landscapes out"->

"but leaving other landscapes out"

L889

"the current sampling situation for the Paleolithic humans"->

"the current sampling situation for Paleolithic humans"

L906-907

"Thus, there exists a limit to model complexity that makes sense to explore with qpAdm on SSL"

[While true, this sentence contains no information about qpAdm, as there exists a limit to model complexity that makes sense to explore with any tool and finite data. And data is always finite. What would be informative is an indication of where the complexity limit lies for people wish to use qpAdm].

L987

"that part of the space is populated by most symmetric models" ->

(A) "that part of the space is populated by the most symmetric models"

OR

(B) "that part of the space is populated by most of the symmetric models"

[Inclusion of article "the" actually makes a difference in meaning here! (so rare!)]

In the current version some symmetric models populate other parts of the space.

In my suggested version (A) those models which are most symmetric, populate that part of the space, but my understanding is symmetry of the models is binary yes/no. 'most' then would not be appropriate to describe symmetry. So I favour (B)]

L995

"tests also became popular" -> "tests have also become popular" (I believe these tests still are popular).

L1003

". And frequency..." -> ". Frequency"

L1055

"are populated by models highly unevenly" ->

"are highly unevenly populated by models"

1100

"protocols on the real-world archaeogenetic sampling"->

"protocols on real-world archaeogenetic sampling"

1148-1149

"following this approach, it is hardly possible to find a single best model complexity level for a target"

[Again "hardly possible" is ambiguous in English See L116 comment. See also "hardly achievable" on L1149 - difficult to

achieve?]

1233

"the relaxed conditions on EAF do not allow to reject efficiently the highly asymmetric models" ->
"the relaxed conditions on EAF do not allow the highly asymmetric models to be efficiently rejected"

[cf comment L36, see also L1237,L1251, L1410, L1536, 1681]

1587-1590

"To be precise, on the "10-5 to 10-2" SSL the pre-qpAdm and post-qpAdm distributions of models in the optimality metric space are uncorrelated for the simplest, two-way, models (Figs. 9, 14, Table S11), but even this is not true in the case of the real data (Fig. 14)."

["but even this is untrue in the case of the real data"... even non-correlation is untrue for real data? ...rewording necessary]

1611

"of chosen complexity classes only (one- and two-way)" ->
"of chosen complexity classes (only one- and two-way)"

1618-1620

"This problem was deliberately left out since in the literature more attention is paid to interpretation of "fitting" than of rejected qpAdm models."

[as this is in the Discussion, the authors should feel free to express an opinion on whether the literature's attention is well directed on the issue]

L1656-1657

"compared to the real population history of humans and other species as we understand it" ->
"compared to the real population histories of humans and other species as we understand them"

L1712-1716

"Another way of radically improving FDR of qpAdm protocols applied to AGS histories is combining qpAdm with an unsupervised ADMIXTURE analysis (Suppl. text 4), however we have not tested ADMIXTURE on low-quality data, and supervised ADMIXTURE protocols were not tested either. Unsupervised ADMIXTURE has known problems even on high-quality data and AGS histories (Lawson et al. 2018). That is why this recommendation is still tentative."

[It seems strange to describe a tentative recommendation as a "way of radically improving FDR"]

L1764-1768

"EAF lie largely within (0, 1) and correlate with simulated gene-flow intensities only for sources located symmetrically around the target, but for asymmetric sources they vary widely both outside and, complicating the matters even further, inside (0, 1) Although this property of AF estimation is trivial from the mathematical perspective, it is not recognized in practice"

[cf comment L522-252: This needs explained. Further, returning 'proportions' outside the range (0,1) is NOT a general property of AF estimation, as there are many ways to estimate ancestry proportion that actually return estimates on (0,1) for two sources or the n-simplex for n sources (eg STRUCTURE and all its relatives and descendants). I suggest there is good reason for estimates outside(0,1) to be "not recognised in practice". I suggest it is quite possible for something to be both mathematically trivial and WRONG. Cf also Methods L1997-1998].

1780-1782

"efficient discrimination of the worst models from ideal ones is possible with a p-value threshold and the distal non-rotating (but not rotating!) protocol"

[Unclear text]

L1804-1811

"We showed that in this situation robust rejection of highly asymmetric complex models with distant sources (with average ST distance greater or approximately equal to the landscape radius) is possible only for very specific combinations of landscape sparsity and model feasibility criteria, but models of that sort are by far the most common in random sets of demes drawn from both simulated landscapes (Figs. 9 and S20b, Table S11) and real archaeogenetic data (Fig. 14). Furthermore, this optimal combination of sparsity and feasibility criteria we found (EAF {plus minus} 2 SE \in (0, 1) and the "10-3 to 10-2" landscapes) is very uncommon in practice"

[

- A) robust rejection is possible only for very specific combinations
- B) but models of that sort are by far the most common in sim and real data
- C) Furthermore this optimal combination is very uncommon in practice

This text appears self-contradictory. Re-word.

]

1869-1870

"and at even higher levels of sparsity the method stops working completely (at least for our simulations where all demes arise via a multifurcation)."

[Is the suggestion here that all demes arising from a multifurcation is not realistic? - it does seem a strange way to start a lattice simulation - perhaps give the reader more info].

L1997-1998

"Inferred admixture proportions in our analysis were not constrained between 0 and 1."

[The reader needs to know why and how. See comments L522-252, L1764-1768]

L2122 "linked sites were pruned with PLINK v.2.00a3LM (Chang et al. 2015)"

[Sites on the same chromosome (linked) were pruned? That would be strange. Perhaps what is meant here is: sites with allelic states in statistical association were pruned. This would make more textual sense as the pruning has an r^2 argument (and sometimes in the literature statistical association (linkage disequilibrium) is confounded with being linked)... But would pruning sites in LD make any sense in terms of inference? What is the justification? Admixture generates LD genome wide (Baird 2015) and is therefore a potentially important part of the signal in the simmed data... why prune this signal out? If a data reduction step is necessary before PCA, a random thinning would seem more appropriate]

L2131 "Prior to the analysis, linked sites were pruned with PLINK v.2.00a3LM (Chang et al. 2015)"

[See comment L2122... Admixture generates LD genome wide and is therefore a potentially important part of the signal in the simmed data... why prune this signal out? If a data reduction step is necessary before ADMIXTURE analysis, a random thinning would seem more appropriate]

References

Baird, S. J. (2015). "Exploring linkage disequilibrium." *Molecular ecology resources* 15(5): 1017-1019.

Baird, S. J. E. and F. Santos (2010). "Monte Carlo integration over stepping stone models for spatial genetic inference using approximate Bayesian computation." *Molecular Ecology Resources* 10(5): 873-885.

Beaumont, M. A. and M. Panchal (2008). "On the validity of nested clade phylogeographical analysis." *Molecular Ecology* 17(11).

Knowles, L. L. (2008). "WHY DOES A METHOD THAT FAILS CONTINUE TO BE USED?" *Evolution* 62(11): 2713-2717.

Reviewer #2:

General comments

The work is timely and important. This kind of analysis is extremely work intensive and the results are complex to express. I am reminded of the Nested Clade Analysis (NCA) debacle. Two research groups spent years of effort to measure NCA error rates (Beaumont and Panchal 2008, Knowles 2008) because an untested method had gone viral. That effort was greatly hampered because the NCA decision process could not be automated (cf L707-708 "this frequency of assumption violations was not quantified in our study since it requires another layer of manual topological analysis.").

I congratulate the authors on having done a good job of expressing complex results. I have a long list of very minor typos and suggestions for wording, but no issues sufficiently important that they should slow the publication of the work.

There are, however (as always) a few issues which the authors should try to resolve before publication, and that should not take much effort.

- 1) The description of the spatial lattice needs clarified*
- 2) The description of the qpAdm EAF estimator needs clarified*
- 3) There is one passage (L1804-1811) which appears self-contradictory.*

These first two crop up several times in the minor issues below.

Finally, in the Methods(L2122,L2131) it would be good to have some justification for data reduction by removal of sites found to be in admixture linkage disequilibrium, rather than data reduction by random subsetting. Admixture generates linkage disequilibrium genome-wide, and so it would seem this data reduction step is preferentially removing signal that should be available for inference. As with the contradictory passage, these are elaborated on below.

Stuart J.E. Baird

Studenec 18th November 2024

We are grateful to the reviewer for this positive assessment of our work. Since all the issues listed above are also mentioned in the specific comments, we respond to them below.

Minor issues

L36

"In this situation only a specific combination of landscape properties and feasibility criteria allows to efficiently reject highly asymmetric non-optimal models most abundant in random deme sets."

->

"In this situation only a specific combination of landscape properties and feasibility criteria allows highly asymmetric non-optimal models most abundant in random deme sets to be efficiently rejected."

["allows to" is missing a noun phrase before the infinitive phrase]

The Abstract where this sentence comes from was re-written and the sentence was removed.

L54

"only a few studies were devoted to testing" ->

"only a few studies have been devoted to testing"

The mistake in grammar was fixed (see line 100; all the line numbers here and below refer to the version with tracked changes).

L116

"interpretating deviations of D- and f4-statistics from 0 (or lack thereof) becomes hardly possible if both branch pairs are connected by detectable gene flows, a typical situation on isolation-by-distance landscapes"

[Here "hardly possible" is unclear. "Almost impossible" is different from "possible, given hard work"]

The sentence was clarified by choosing the first alternative, "almost impossible" (see line 206).

Box1

SSL entry:

"here we describe an implementation used in this study. A stepping-stone landscape is originally "unfolded" from a single founder deme via a serial founder process that can be represented by a bifurcating tree (Estavoyer and Francois 2022), or via multifurcation, i.e. star radiation. This stage is followed by a gene-flow era represented as an undirected graph connecting demes, where each edge represents a bidirectional gene flow, and gene-flow intensities are allowed to differ in the forward and reverse directions. Gene flows on this undirected graph happen in one or more epochs, with unidirectional gene-flow intensities sampled randomly from uniform or non-uniform distributions. In this study all landscapes are based on the triangular lattice, and node degree varies from 3 to 6."

See also Methods LL2173-2174

"The demes were arranged on a triangular lattice, forming an approximately circular landscape, where each deme was connected by bidirectional gene flows to 3 to 6 neighboring demes (Fig. 7)."

I am familiar with modelling geneflow on the triangular lattice (Baird and Santos 2010): it seems a strange choice here because indexing of the graph connecting demes in the gene flow era must be more complex than indexing of a traditional square lattice. The reader needs a line explaining why a triangular lattice was chosen, and how nodes of degree higher than 3 are connected (assuming degree 3 is nearest-neighbour). This last needs explanation especially as L844-845 "gene flows between non-neighboring demes were not allowed"... yet in the traditional notion of a triangular tiling of the plane, a node can only have 3 direct neighbours.

Nodes (demes) in the lattice were located at vertices of triangles tiling the plane, and in the unbounded case each node has 6 nearest neighbors. This lattice is sometimes called hexagonal, sometimes triangular, where the confusion probably comes from. We have clarified in **Box 1**, in Results (line 500) and in Methods (line 3387) that we used a type of lattice formed by triangles tiling the plane and that it should not be confused with hexagonal (honeycomb) tiling:

"In this study all SSL are based on finite hexagonal lattices where demes are located at vertices of triangles tiling the plane (not be confused with honeycomb tiling), and node degree varies from 3 to 6."

The lattice is also shown in **Fig. 4** presenting our SSL simulation setup. This type of lattice is frequently used in the spatial genetics literature (see, e.g., Petkova et al. 2016, Visualizing spatial population structure with estimated effective migration surfaces. *Nature Genetics*; the term "triangular grid" is used in that study).

Box1

proxy (ancestry) source entry:

"assumed to be cladal with one of true ancestry sources"->

"assumed to be cladal with one of the true ancestry sources"

The mistake in grammar was fixed.

L205

"This new type of qpAdm protocols, termed "rotating" protocol"->

"This new type of qpAdm protocol, termed the "rotating" protocol"

The mistakes in grammar were fixed, see line 326.

L222-226

"As an additional criterion of a fitting model, all inferred admixture proportions (see[...]), or proportions {plus minus} 2 standard errors (Narasimhan et al. 2019), may be required to lie between 0 and 1."

[This reads strangely. To require a proportion to lie between 0 and 1 is not a restriction].

Although it is possible to restrict admixture proportion estimates reported by the *qpAdm* algorithm to $[0, 1]$, this is not its default behavior, and this condition is usually included in model feasibility criteria instead: "Models are deemed implausible if their estimated admixture proportions fall outside the biologically relevant range (0–1) or if they are rejected statistically by having a small *P*-value." (Harney et al. 2021, Assessing the performance of *qpAdm*: a statistical tool for studying population admixture. *Genetics*, 217(4), iyaa045; this study is co-authored by Nick Patterson, the main developer of *qpAdm* and *Admixtools*). See also a justification for this unusual property of the *qpAdm* algorithm in the same study:

"This stepping-stone model can also be used to highlight an interesting feature of qpAdm, which is that admixture proportion estimates that fall outside the bounds of 0–1 may also be informative about the history of the population being modeled. It has previously been suggested that in cases where the estimated admixture proportion exceeds 1, this is indicative of the target population falling in a more extreme position along a genetic cline than either of the modeled source populations (Lazaridis et al. 2017). We confirm this to be true by attempting to model population 1 as the product of admixture between source populations 2 and 3 (Supplementary Figure S14). In this model, an estimated alpha of 1 would indicate that population 1 could be modeled as deriving 100% of its ancestry from population 2. Instead, we observe that all of the estimates of alpha all fall outside the bounds of 0–1, instead centering around 2, supportive of population 1's more extreme position along the genetic cline that also includes populations 2 and 3."

As another example from the literature, we quote Speidel et al. 2025, a study we re-visited in section 7: "we remove models with infeasible admixture proportions", which means that models with admixture fraction estimates outside $[0,1]$ were not shown in that paper (see lines 2494-2497). Thus, we followed the standard practice in our study. We've added a clarification in the Introduction and Results:

lines 355-358:

"We note that estimated admixture fractions (abbreviated as EAF) not restricted to $[0, 1]$ are reported by qpAdm by default, and this setting is used in a great majority of published studies (see Harney et al. 2021 for a justification of this unusual property of the algorithm)."

lines 810-812:

"Since estimated admixture fractions reported by qpAdm may vary from $\ll -1$ to $\gg 1$ (see a justification for this unusual property of the algorithm in Harney et al. 2021)..."

*Further: what admixture proportion is returned in the case of no information? When software is required to return an estimate irrespective of information available, the default is to return a mid-interval estimate: 1/2. For example, this is the case for STRUCTURE $K=2$ *q* estimates: Intermediate*

and Unsupported estimates can only be distinguished by the confidence intervals STRUCTURE places round them. In the no information case the confidence interval is 0-1.

If data quality is poor, EAF standard errors reported by *qpAdm* tend to grow indeed (Harney et al. 2021).

This suggests only the (Narasimhan et al. 2019) approach above has any hope of ensuring there is support for admixture

This is actually one of the main conclusions of our work: a requirement that $EAF \pm 2 SE$ is within $(0, 1)$ following Narasimhan et al. 2019 is very important for reducing FPR of hence FDR of high-throughput *qpAdm* protocols, and our exploration of dozens of composite model feasibility criteria on various SSL simulations supports this statement. This point is mentioned in the Abstract (lines 46-47), and at several places in the text.

Lines 1721-1725:

"Since pre-study odds fall rapidly with increasing model complexity, to keep FDR for three-way and more complex models below 50%, both FPR and FNR must remain relatively low for these model classes (equation 1), but this is achievable only in the case of densely sampled landscapes and stringent conditions on EAF ($EAF \pm 2 SE \in (0, 1)$)"

Lines 1782-1787:

"A key observation is that the most optimal qpAdm setup (the densest, " 10^{-3} to 10^{-2} ", landscape, $EAF \pm 2 SE \in (0, 1)$, p -value ≥ 0.001 , "trailing" models not checked) found in the previous section indeed results in the lowest fractions of positive experiments with "bad" outcomes, as compared to other conditions tested."

Lines 2755-2760:

"Furthermore, this optimal combination of sparsity (" 10^{-3} to 10^{-2} " landscapes) and feasibility criteria we found is very uncommon in practice: restricting both "right" and "left" deme sets to a sub-continental region, using a low p -value threshold such as ≥ 0.001 , and requiring that $EAF \pm 2 SE \in (0, 1)$ are very rare approaches, to our knowledge. For example, in Speidel et al. 2025, a study we re-visited, the former condition was satisfied, but the latter two were not."

L271

"a particular type of SNP ascertainment: selecting sites" ->

"a particular type of SNP ascertainment bias: selecting sites"

Corrected, see line 405.

L287-290

"Finding a feasible two-way or more complex admixture model for a target is often interpreted as solid evidence for gene flow, especially if PCA and ADMIXTURE methods confirm the same signal. Thus, qpAdm protocols are used in fact as formal tests for admixture, whereas the latter two methods are not formal tests."

[Given the rather arbitrary and adhoc test criteria that have been described (egg L222-226 comment above), it seems a stretch to suggest qpAdm can be used to make a formal test... I note that your following paragraph goes no to say essentially this. Perhaps change the above text to: "... are used as supposed formal tests..."]

The text was corrected as suggested, see lines 427-428.

L330

"Below we explore on simulated AGS histories"->

"Below we explore simulated AGS histories"

We have rephrased as follows (lines 574-575): *"Below we explore, relying on simulated AGS histories, performance (mainly FDR) of qpAdm protocols representing the spectrum of protocols used in the literature."*

L352

"where target group's sampling date"->

"where the target group's sampling date"

The mistake in grammar was fixed, see line 595.

L397

"simulated graph six most ancient groups were selected"->

"simulated graph the six most ancient groups were selected"

The mistake in grammar was fixed, see lines 649-650.

L522-252

"admixture fractions estimated by qpAdm (abbreviated as EAF) [...]. Since qpAdm EAF vary from <<-1 to >>1...."

[This needs explained: how can an admixture proportion (by definition on (0,1) vary from <<-1 to >>1 ? To interpret the work done here, the reader needs to know what is actually being discussed]

Please see the discussion of this issue above (the comment for L222-226). We have added the following clarification here too (lines 810-812 in the updated version with tracked changes):

"Since estimated admixture fractions reported by qpAdm may vary from <<-1 to >>1 (see a justification for this unusual property of the algorithm in Harney et al. 2021)..."

L619

"in target's history"->

"in a target's history"

The mistake in grammar was corrected here (on line 884) and elsewhere in the text (the last entry in **Box 1**, lines 891, 3301).

L625-627

"occasionally fitting and false is significant too (Table S1b). Statistically significant trends in the same direction are observed for distal models too, but in this case median"->

"occasionally fitting and false is also significant (Table S1b). Statistically significant trends in the same direction are observed for distal models as well, but in this case median"

[too many toos]

The sentence cited here was removed in the process of shortening the paper.

L675-678

"A large fraction of false proximal models emerges as fitting due to false rejections of one-way models because of violations of the topological assumptions (Fig. 1), for instance, due to "left-to-right" gene flows, and that prompts the investigator to test more complex two-way models,"

"and that prompts the investigator" ->

"and these prompt the investigator"

[false rejections (plural) prompt the investigator]

The mistake in grammar was corrected as suggested, see lines 975-976.

L824

"but leave other landscapes out"->

"but leaving other landscapes out"

The mistake in grammar was corrected, see line 1175.

L889

"the current sampling situation for the Paleolithic humans"->

"the current sampling situation for Paleolithic humans"

The mistake in grammar was corrected, see line 1275.

L906-907

"Thus, there exists a limit to model complexity that makes sense to explore with qpAdm on SSL"

[While true, this sentence contains no information about qpAdm, as there exists a limit to model complexity that makes sense to explore with any tool and finite data. And data is always finite. What would be informative is an indication of where the complexity limit lies for people wish to use qpAdm].

We have clarified the sentence in the following way (lines 1292-1294):

"Thus, a limit to model complexity that makes sense to explore with high-throughput qpAdm on SSL is rather low in most of our simulations: just two or three sources (see more on this topic below in the paragraphs focused on FDR)."

In *section 5b* (lines 1682-1735, **Fig. 7**) and elsewhere in the updated manuscript we discuss that the limits to model complexity that makes sense to explore with high-throughput qpAdm protocols are due to growing pre-study odds and less precise EAF estimates for complex models (the latter leading to increased FNR, see lines 1614-1617, 2164, 2642-2644).

L987

"that part of the space is populated by most symmetric models" ->

(A) "that part of the space is populated by the most symmetric models"

OR

(B) "that part of the space is populated by most of the symmetric models"

[Inclusion of article "the" actually makes a difference in meaning here! (so rare!)]

In the current version some symmetric models populate other parts of the space.

In my suggested version (A) those models which are most symmetric, populate that part of the space, but my understanding is symmetry of the models is binary yes/no. 'most' then would not be appropriate to describe symmetry. So I favour (B)]

We disagree here: some admixture models in the SSL context are more symmetric than others (compare sources at a very small angular distance and those almost opposite to each other), and we try to capture that gradient with our "minimal source-target-source angle" metric (see **Box 1, Figs. 4 and 6**). That's why we prefer the suggested version (A); see lines 532-534 in the supplement where this text was moved.

L995

"tests also became popular" -> "tests have also become popular" (I believe these tests still are popular).

The mistake in grammar was corrected, see line 540 in the supplement where this text was moved.

L1003

". And frequency..." -> ". Frequency"

Corrected as suggested, see line 548 in the supplement where this text was moved (no changes are tracked in SI).

L1055

"are populated by models highly unevenly" ->

"are highly unevenly populated by models"

The mistake in grammar was corrected, see line 1377.

L1100

"protocols on the real-world archaeogenetic sampling"->

"protocols on real-world archaeogenetic sampling"

The mistake in grammar was corrected, see line 1521.

L1148-1149

"following this approach, it is hardly possible to find a single best model complexity level for a target"

[Again "hardly possible" is ambiguous in English See L116 comment. See also "hardly achievable" on L1149 - difficult to achieve?]

The sentence was clarified by choosing the phrase "almost impossible" in the former case (see line 1624) and the phrase "difficult to achieve" in the latter case (see line 1626):

"We note that, following this approach, it is almost impossible to find a single best model complexity level for a target and a single best model at that complexity level, but that is difficult to achieve in principle on SSL as we discuss in the next section."

L1233

*"the relaxed conditions on EAF do not allow to reject efficiently the highly asymmetric models" ->
"the relaxed conditions on EAF do not allow the highly asymmetric models to be efficiently rejected"*

[cf comment L36, see also L1237, L1251, L1410, L1536, 1681]

All these mistakes in grammar were corrected: see lines 431-432, 1200-1201, 1802-1803, 2861-2862,

L1587-1590

"To be precise, on the "10⁻⁵ to 10⁻²" SSL the pre-qpAdm and post-qpAdm distributions of models in the optimality metric space are uncorrelated for the simplest, two-way, models (Figs. 9, 14, Table S11), but even this is not true in the case of the real data (Fig. 14)."

["but even this is untrue in the case of the real data"... even non-correlation is untrue for real data? ...rewording necessary]

We have rephrased in the following way (lines 2527-2531): "To be precise, on the "10⁻⁵ to 10⁻²" SSL the pre-qpAdm and post-qpAdm distributions of models in the optimality metric space are uncorrelated just for the simplest, two-way, models (**Figs. 6, 11, Table S11**), but in the case of the real data these distributions are correlated even for two-way models (**Fig. 11**)."

1611

"of chosen complexity classes only (one- and two-way)" ->

"of chosen complexity classes (only one- and two-way)"

The mistake in grammar was corrected, see line 3067.

1618-1620

"This problem was deliberately left out since in the literature more attention is paid to interpretation of "fitting" than of rejected qpAdm models."

[as this is in the Discussion, the authors should feel free to express an opinion on whether the literature's attention is well directed on the issue]

To make our reasoning more transparent, we have reorganized the whole section on limitations of our AGS-focused analyses, see lines 3061-3099 (now this section opens the Methods chapter). The sentence in question was removed since we skipped classification of universally rejected models into true and false ones also for another more important reason, and we highlight it now (lines 3064-3075):

"We ... have estimated FDR instead of FPR due to an important technical limitation: the process of model classification into true and false ones is difficult to automate fully since it requires careful interpretation of the simulated topology and simulated admixture proportions (this is illustrated

by the case studies in **Fig. S6**; note that we attempted to estimate FPR and FNR for *qpAdm* results on AGS histories indirectly, see the end of *section 3*). Due to this limitation, we did not attempt to classify universal rejections of two-way models by *qpAdm* or other methods into false rejections due to violations of the topological assumptions of *qpAdm* (**Fig. 1**) and true rejections when the true admixture history of the target does not fit a two-way model."

L1656-1657

*"compared to the real population history of humans and other species as we understand it" ->
"compared to the real population histories of humans and other species as we understand them"*

The mistake in grammar was corrected, see lines 3140-3141.

L1712-1716

"Another way of radically improving FDR of qpAdm protocols applied to AGS histories is combining qpAdm with an unsupervised ADMIXTURE analysis (Suppl. text 4), however we have not tested ADMIXTURE on low-quality data, and supervised ADMIXTURE protocols were not tested either. Unsupervised ADMIXTURE has known problems even on high-quality data and AGS histories (Lawson et al. 2018). That is why this recommendation is still tentative."

[It seems strange to describe a tentative recommendation as a "way of radically improving FDR"]

To soften the statement, we have replaced "Another way of radically improving FDR" with "A potential way of improving FDR", see lines 2611-2614.

L1764-1768

"EAF lie largely within (0, 1) and correlate with simulated gene-flow intensities only for sources located symmetrically around the target, but for asymmetric sources they vary widely both outside and, complicating the matters even further, inside (0, 1) Although this property of AF estimation is trivial from the mathematical perspective, it is not recognized in practice"

[cf comment L522-252: This needs explained. Further, returning 'proportions' outside the range (0,1) is NOT a general property of AF estimation, as there are many ways to estimate ancestry proportion that actually return estimates on (0,1) for two sources or the n-simplex for n sources (eg STRUCTURE and all its relatives and descendants). I suggest there is good reason for estimates outside(0,1) to be "not recognised in practice". I suggest it is quite possible for something to be both mathematically trivial and WRONG. Cf also Methods L1997-1998].

Please see the discussion of this issue above (the comments for L222-226 and L522). We have emphasized that this property of the default *qpAdm* algorithm is unusual and cited Harney et al. 2021 several times as a justification by the *qpAdm* developers (in Introduction on lines 355-358, Results on lines 811-813, and Methods on lines 3246-3247).

1780-1782

"efficient discrimination of the worst models from ideal ones is possible with a p-value threshold and the distal non-rotating (but not rotating!) protocol"

[Unclear text]

We have removed this sentence from Discussion in the process of streamlining it, and we have rephrased relevant sentences in Results to clarify our statements (see section 6b, lines 2153-2160):

"The systematic distal non-rotating protocol with the second feasibility criterion (the p-value threshold) demonstrates attractive properties in the case of three-, four-, and six-way models: there is strong correlation between the fraction of models passing the p-value threshold and model optimality, and the percentage of truly ideal models rejected equals the p-value threshold, as expected (Fig. 9b). In contrast, such a "clean" result was not achieved with the distal rotating protocol: the fraction of models passing the p-value threshold grows with increasing model complexity, but at the same time its correlation with model optimality disappears (Fig. 9b)."

L1804-1811

"We showed that in this situation robust rejection of highly asymmetric complex models with distant sources (with average ST distance greater or approximately equal to the landscape radius) is possible only for very specific combinations of landscape sparsity and model feasibility criteria, but models of that sort are by far the most common in random sets of demes drawn from both simulated landscapes (Figs. 9 and S20b, Table S11) and real archaeogenetic data (Fig. 14). Furthermore, this optimal combination of sparsity and feasibility criteria we found (EAF {plus minus} 2 SE $\in (0, 1)$ and the "10⁻³ to 10⁻²" landscapes) is very uncommon in practice"

[A) robust rejection is possible only for very specific combinations

B) but models of that sort are by far the most common in sim and real data

C) Furthermore this optimal combination is very uncommon in practice

This text appears self-contradictory. Re-word.]

We agree that the text does not convey our message well: not admixture models as combinations of sources on the landscape, but combinations of landscape properties and model feasibility criteria are meant in statements A and C. We think that our emphasis on pre-study odds and FDR in the new manuscript version helps to improve the argument, see lines 2764-2773:

"Highly asymmetric models with distant sources are by far the most common in random sets of demes drawn from simulated landscapes (Figs. 6, S23) and real archaeogenetic data (Fig. 11): that is, pre-study odds are low (Fig. 7). Efficient rejection of these non-optimal models (with average ST distance greater or approximately equal to the landscape radius) is possible only for very specific combinations of landscape sparsity (influencing mostly power = 1 – FNR) and model feasibility criteria (influencing both FPR and FNR). Furthermore, this optimal combination of sparsity ("10⁻³ to 10⁻²" landscapes) and feasibility criteria we found is very uncommon in practice:

restricting both “right” and “left” deme sets to a sub-continental region, using a low p-value threshold such as ≥ 0.001 , and requiring that $EAF \pm 2 SE \in (0, 1)$ are very rare approaches, to our knowledge.”

1869-1870

"and at even higher levels of sparsity the method stops working completely (at least for our simulations where all demes arise via a multifurcation)."

[Is the suggestion here that all demes arising from a multifurcation is not realistic? - it does seem a strange way to start a lattice simulation - perhaps give the reader more info].

We did not mean in this sentence that the situation where all demes arise from a multifurcation is not realistic (although unrealistic it is!). We have added a clarification on lines 3438-3447 in the Methods:

*“There are so many ways of “unfolding” an initial lattice state via a serial-founder process represented as a bifurcating tree (Estavoyer and François 2022) that we decided to avoid this additional level of complexity in our study and initialized the lattice simulations with a multifurcation, but deep in the past, at 2,220 generations before the oldest sampling time point. In this way we focus on genetic structure generated by gene-flow networks on SSL instead of structure generated by phylogenetic processes. If gene-flow intensities on SSL are relatively high, traces of the initial multifurcation are lost by the beginning of the sampling epoch, but this is not true for simulations with the lowest per-generation gene-flow intensities (the “ 10^{-5} to 10^{-4} ” landscapes, see below), as demonstrated by our analysis of one-way qpAdm models interpreted as cladality tests (**Suppl. text 5** and **Fig. S21**).”*

L1997-1998

"Inferred admixture proportions in our analysis were not constrained between 0 and 1."

[The reader needs to know why and how. See comments L522-252, L1764-1768]

Please see the discussion of this issue above (the comments for L222-226 and L522). We have emphasized that this property of the default qpAdm algorithm is unusual and cited Harney et al. 2021 several times as a justification by the qpAdm developers (in Introduction on lines 355-358, Results on lines 811-813, and Methods on lines 3246-3247).

L2122 *"linked sites were pruned with PLINK v.2.00a3LM (Chang et al. 2015)"*

[Sites on the same chromosome (linked) were pruned? That would be strange. Perhaps what is meant here is: sites with allelic states in statistical association were pruned. This would make more textual sense as the pruning has an r^2 argument (and sometimes in the literature statistical association (linkage disequilibrium) is confounded with being linked)... But would pruning sites in LD make any sense in terms of inference? What is the justification? Admixture generates LD genome wide (Baird 2015) and is therefore a potentially important part of the signal in the

simmed data... why prune this signal out? If a data reduction step is necessary before PCA, a random thinning would seem more appropriate]

L2131 *"Prior to the analysis, linked sites were pruned with PLINK v.2.00a3LM (Chang et al. 2015)"*

[See comment L2122... Admixture generates LD genome wide and is therefore a potentially important part of the signal in the simmed data... why prune this signal out? If a data reduction step is necessary before ADMIXTURE analysis, a random thinning would seem more appropriate]

We indeed pruned sites with allelic states in statistical association, following recommendations from key publications applying PCA to genomic SNP data (see, for example, Privé F, et al. 2020. Efficient toolkit implementing best practices for principal component analysis of population genetic data. *Bioinformatics*. 36(16):4449-4457). In the case of LD pruning prior to ADMIXTURE analyses, that practice is recommended by the software developers themselves: *"Our model makes the further assumption of linkage equilibrium among the markers. Dense marker sets should be pruned to mitigate background linkage disequilibrium (LD). This can be done informally, by thinning the marker set according to a minimum separation criterion or by pruning markers observed to be in linkage disequilibrium on the basis of common LD summary statistics such as D' or r^2 . Neither pruning approach is a perfect remedy for linkage disequilibrium."* (Alexander et al. 2009. Fast model-based estimation of ancestry in unrelated individuals. *Genome Res*. 19(9):1655-1664).

We have clarified this in Methods on lines 3360-3365 (PCA):

"Prior to the analysis, sites with allelic states in statistical association were pruned with PLINK v.2.00a3LM (Chang et al. 2015) using the following settings: window size, 2000 SNPs; window step, 100 SNPs; r^2 threshold = 0.5 (argument "--indep-pairwise 2000 100 0.5"). This kind of linkage disequilibrium (LD) pruning is common in the population genetic literature since complex LD structure may confound PCA, especially higher PCs (Privé et al. 2020)."

and on lines 3382-3386 (ADMIXTURE):

"Prior to the analysis, sites with allelic states in statistical association were pruned with PLINK v.2.00a3LM (Chang et al. 2015) using the following settings: window size, 2000 SNPs; window step, 100 SNPs; r^2 threshold = 0.5 (argument "--indep-pairwise 2000 100 0.5"). This kind of LD pruning relying on r^2 is recommended since the underlying model assumes sites in linkage equilibrium (Alexander et al. 2009)."

We think that criticizing robustness of this LD pruning approach is beyond the scope of our study since only PCA and ADMIXTURE results on the AGS simulations are affected by LD pruning, but they are discussed overwhelmingly in the supplement (see **Fig. S6** and **Suppl. text 4**). No *qpAdm* results on AGS or SSL rely on LD-pruned data in our study.

February 22, 2025

RE: GENETICS-2025-307840

Dear Dr. Flegontov:

I am pleased to accept your manuscript titled "Performance of qpAdm-based screens for genetic admixture on admixture-graph-shaped histories and stepping-stone landscapes" for publication in GENETICS, pending minor revision. I am satisfied with your detailed response to the reviewer's concerns - this has resolved all the previous issues. However, the link to figshare did not work: you will need to upload and document the scripts used in the analysis before we can finally accept the MS.

I expect you should be able to provide the necessary data within 30 days. A suitably revised manuscript will be acceptable for publication; I don't expect to send it out for review.

Please ensure that the Data Availability Statement at the end of the Materials and Methods section fulfils our requirements. Details are available at <https://academic.oup.com/genetics/content/prep-manuscript>. The DAS should include the accession numbers or DOIs of any data you have placed in public repositories, describe supplemental material, include applicable IRB numbers, and may include specifications for how to properly acknowledge or cite the data.

Follow this link to submit the revised manuscript: Link Not Available

Thank you for submitting this story to Genetics.

Sincerely,

Nick Barton
Senior Editor
GENETICS

Approved by:
Howard Lipshitz
Editor in Chief
GENETICS

Reviewer #2:

General comments

The work is timely and important. This kind of analysis is extremely work intensive and the results are complex to express. I am reminded of the Nested Clade Analysis (NCA) debacle. Two research groups spent years of effort to measure NCA error rates (Beaumont and Panchal 2008, Knowles 2008) because an untested method had gone viral. That effort was greatly hampered because the NCA decision process could not be automated (cf L707-708 "this frequency of assumption violations was not quantified in our study since it requires another layer of manual topological analysis.").

I congratulate the authors on having done a good job of expressing complex results. I have a long list of very minor typos and suggestions for wording, but no issues sufficiently important that they should slow the publication of the work.

There are, however (as always) a few issues which the authors should try to resolve before publication, and that should not take much effort.

- 1) The description of the spatial lattice needs clarified*
- 2) The description of the qpAdm EAF estimator needs clarified*
- 3) There is one passage (L1804-1811) which appears self-contradictory.*

These first two crop up several times in the minor issues below.

Finally, in the Methods(L2122,L2131) it would be good to have some justification for data reduction by removal of sites found to be in admixture linkage disequilibrium, rather than data reduction by random subsetting. Admixture generates linkage disequilibrium genome-wide, and so it would seem this data reduction step is preferentially removing signal that should be available for inference. As with the contradictory passage, these are elaborated on below.

Stuart J.E. Baird

Studenec 18th November 2024

We are grateful to the reviewer for this positive assessment of our work. Since all the issues listed above are also mentioned in the specific comments, we respond to them below.

Minor issues

L36

"In this situation only a specific combination of landscape properties and feasibility criteria allows to efficiently reject highly asymmetric non-optimal models most abundant in random deme sets."
->

"In this situation only a specific combination of landscape properties and feasibility criteria allows highly asymmetric non-optimal models most abundant in random deme sets to be efficiently rejected."

["allows to" is missing a noun phrase before the infinitive phrase]

The Abstract where this sentence comes from was re-written and the sentence was removed.

L54

"only a few studies were devoted to testing" ->

"only a few studies have been devoted to testing"

The mistake in grammar was fixed (see line 100; all the line numbers here and below refer to the version with tracked changes).

L116

"interpretating deviations of D- and f4-statistics from 0 (or lack thereof) becomes hardly possible if both branch pairs are connected by detectable gene flows, a typical situation on isolation-by-distance landscapes"

[Here "hardly possible" is unclear. "Almost impossible" is different from "possible, given hard work"]

The sentence was clarified by choosing the first alternative, "almost impossible" (see line 206).

Box1

SSL entry:

"here we describe an implementation used in this study. A stepping-stone landscape is originally "unfolded" from a single founder deme via a serial founder process that can be represented by a bifurcating tree (Estavoyer and Francois 2022), or via multifurcation, i.e. star radiation. This stage is followed by a gene-flow era represented as an undirected graph connecting demes, where each edge represents a bidirectional gene flow, and gene-flow intensities are allowed to differ in the forward and reverse directions. Gene flows on this undirected graph happen in one or more epochs, with unidirectional gene-flow intensities sampled randomly from uniform or non-uniform distributions. In this study all landscapes are based on the triangular lattice, and node degree varies from 3 to 6."

See also Methods LL2173-2174

"The demes were arranged on a triangular lattice, forming an approximately circular landscape, where each deme was connected by bidirectional gene flows to 3 to 6 neighboring demes (Fig. 7)."

I am familiar with modelling geneflow on the triangular lattice (Baird and Santos 2010): it seems a strange choice here because indexing of the graph connecting demes in the gene flow era must be more complex than indexing of a traditional square lattice. The reader needs a line explaining why a triangular lattice was chosen, and how nodes of degree higher than 3 are connected (assuming degree 3 is nearest-neighbour). This last needs explanation especially as L844-845 "gene flows between non-neighboring demes were not allowed"... yet in the traditional notion of a triangular tiling of the plane, a node can only have 3 direct neighbours.

Nodes (demes) in the lattice were located at vertices of triangles tiling the plane, and in the unbounded case each node has 6 nearest neighbors. This lattice is sometimes called hexagonal, sometimes triangular, where the confusion probably comes from. We have clarified in **Box 1**, in Results (line 500) and in Methods (line 3387) that we used a type of lattice formed by triangles tiling the plane and that it should not be confused with hexagonal (honeycomb) tiling:

"In this study all SSL are based on finite hexagonal lattices where demes are located at vertices of triangles tiling the plane (not be confused with honeycomb tiling), and node degree varies from 3 to 6."

The lattice is also shown in **Fig. 4** presenting our SSL simulation setup. This type of lattice is frequently used in the spatial genetics literature (see, e.g., Petkova et al. 2016, Visualizing spatial population structure with estimated effective migration surfaces. *Nature Genetics*; the term "triangular grid" is used in that study).

Box1

proxy (ancestry) source entry:

"assumed to be cladal with one of true ancestry sources"->

"assumed to be cladal with one of the true ancestry sources"

The mistake in grammar was fixed.

L205

"This new type of qpAdm protocols, termed "rotating" protocol"->

"This new type of qpAdm protocol, termed the "rotating" protocol"

The mistakes in grammar were fixed, see line 326.

L222-226

"As an additional criterion of a fitting model, all inferred admixture proportions (see[...]), or proportions {plus minus} 2 standard errors (Narasimhan et al. 2019), may be required to lie between 0 and 1."

[This reads strangely. To require a proportion to lie between 0 and 1 is not a restriction].

Although it is possible to restrict admixture proportion estimates reported by the *qpAdm* algorithm to $[0, 1]$, this is not its default behavior, and this condition is usually included in model feasibility criteria instead: "Models are deemed implausible if their estimated admixture proportions fall outside the biologically relevant range (0–1) or if they are rejected statistically by having a small *P*-value." (Harney et al. 2021, Assessing the performance of *qpAdm*: a statistical tool for studying population admixture. *Genetics*, 217(4), iyaa045; this study is co-authored by Nick Patterson, the main developer of *qpAdm* and *Admixtools*). See also a justification for this unusual property of the *qpAdm* algorithm in the same study:

"This stepping-stone model can also be used to highlight an interesting feature of qpAdm, which is that admixture proportion estimates that fall outside the bounds of 0–1 may also be informative about the history of the population being modeled. It has previously been suggested that in cases where the estimated admixture proportion exceeds 1, this is indicative of the target population falling in a more extreme position along a genetic cline than either of the modeled source populations (Lazaridis et al. 2017). We confirm this to be true by attempting to model population 1 as the product of admixture between source populations 2 and 3 (Supplementary Figure S14). In this model, an estimated alpha of 1 would indicate that population 1 could be modeled as deriving 100% of its ancestry from population 2. Instead, we observe that all of the estimates of alpha all fall outside the bounds of 0–1, instead centering around 2, supportive of population 1's more extreme position along the genetic cline that also includes populations 2 and 3."

As another example from the literature, we quote Speidel et al. 2025, a study we re-visited in section 7: "we remove models with infeasible admixture proportions", which means that models with admixture fraction estimates outside $[0,1]$ were not shown in that paper (see lines 2494-2497). Thus, we followed the standard practice in our study. We've added a clarification in the Introduction and Results:

lines 355-358:

"We note that estimated admixture fractions (abbreviated as EAF) not restricted to $[0, 1]$ are reported by qpAdm by default, and this setting is used in a great majority of published studies (see Harney et al. 2021 for a justification of this unusual property of the algorithm)."

lines 810-812:

"Since estimated admixture fractions reported by qpAdm may vary from $\ll -1$ to $\gg 1$ (see a justification for this unusual property of the algorithm in Harney et al. 2021)..."

*Further: what admixture proportion is returned in the case of no information? When software is required to return an estimate irrespective of information available, the default is to return a mid-interval estimate: 1/2. For example, this is the case for STRUCTURE $K=2$ *q* estimates: Intermediate*

and Unsupported estimates can only be distinguished by the confidence intervals STRUCTURE places round them. In the no information case the confidence interval is 0-1.

If data quality is poor, EAF standard errors reported by *qpAdm* tend to grow indeed (Harney et al. 2021).

This suggests only the (Narasimhan et al. 2019) approach above has any hope of ensuring there is support for admixture

This is actually one of the main conclusions of our work: a requirement that $EAF \pm 2 SE$ is within (0, 1) following Narasimhan et al. 2019 is very important for reducing FPR of hence FDR of high-throughput *qpAdm* protocols, and our exploration of dozens of composite model feasibility criteria on various SSL simulations supports this statement. This point is mentioned in the Abstract (lines 46-47), and at several places in the text.

Lines 1721-1725:

"Since pre-study odds fall rapidly with increasing model complexity, to keep FDR for three-way and more complex models below 50%, both FPR and FNR must remain relatively low for these model classes (equation 1), but this is achievable only in the case of densely sampled landscapes and stringent conditions on EAF ($EAF \pm 2 SE \in (0, 1)$)"

Lines 1782-1787:

"A key observation is that the most optimal qpAdm setup (the densest, "10⁻³ to 10⁻²", landscape, $EAF \pm 2 SE \in (0, 1)$, $p\text{-value} \geq 0.001$, "trailing" models not checked) found in the previous section indeed results in the lowest fractions of positive experiments with "bad" outcomes, as compared to other conditions tested."

Lines 2755-2760:

"Furthermore, this optimal combination of sparsity ("10⁻³ to 10⁻²" landscapes) and feasibility criteria we found is very uncommon in practice: restricting both "right" and "left" deme sets to a sub-continental region, using a low $p\text{-value}$ threshold such as ≥ 0.001 , and requiring that $EAF \pm 2 SE \in (0, 1)$ are very rare approaches, to our knowledge. For example, in Speidel et al. 2025, a study we re-visited, the former condition was satisfied, but the latter two were not."

L271

"a particular type of SNP ascertainment: selecting sites" ->

"a particular type of SNP ascertainment bias: selecting sites"

Corrected, see line 405.

L287-290

"Finding a feasible two-way or more complex admixture model for a target is often interpreted as solid evidence for gene flow, especially if PCA and ADMIXTURE methods confirm the same signal. Thus, qpAdm protocols are used in fact as formal tests for admixture, whereas the latter two methods are not formal tests."

[Given the rather arbitrary and adhoc test criteria that have been described (egg L222-226 comment above), it seems a stretch to suggest qpAdm can be used to make a formal test... I note that your following paragraph goes no to say essentially this. Perhaps change the above text to: "... are used as supposed formal tests..."]

The text was corrected as suggested, see lines 427-428.

L330

"Below we explore on simulated AGS histories"->

"Below we explore simulated AGS histories"

We have rephrased as follows (lines 574-575): *"Below we explore, relying on simulated AGS histories, performance (mainly FDR) of qpAdm protocols representing the spectrum of protocols used in the literature."*

L352

"where target group's sampling date"->

"where the target group's sampling date"

The mistake in grammar was fixed, see line 595.

L397

"simulated graph six most ancient groups were selected"->

"simulated graph the six most ancient groups were selected"

The mistake in grammar was fixed, see lines 649-650.

L522-252

"admixture fractions estimated by qpAdm (abbreviated as EAF) [...]. Since qpAdm EAF vary from <<-1 to >>1...."

[This needs explained: how can an admixture proportion (by definition on (0,1) vary from <<-1 to >>1 ? To interpret the work done here, the reader needs to know what is actually being discussed]

Please see the discussion of this issue above (the comment for L222-226). We have added the following clarification here too (lines 810-812 in the updated version with tracked changes):

"Since estimated admixture fractions reported by qpAdm may vary from <<-1 to >>1 (see a justification for this unusual property of the algorithm in Harney et al. 2021)..."

L619

"in target's history"->

"in a target's history"

The mistake in grammar was corrected here (on line 884) and elsewhere in the text (the last entry in **Box 1**, lines 891, 3301).

L625-627

"occasionally fitting and false is significant too (Table S1b). Statistically significant trends in the same direction are observed for distal models too, but in this case median"->

"occasionally fitting and false is also significant (Table S1b). Statistically significant trends in the same direction are observed for distal models as well, but in this case median"

[too many toos]

The sentence cited here was removed in the process of shortening the paper.

L675-678

"A large fraction of false proximal models emerges as fitting due to false rejections of one-way models because of violations of the topological assumptions (Fig. 1), for instance, due to "left-to-right" gene flows, and that prompts the investigator to test more complex two-way models,"

"and that prompts the investigator" ->

"and these prompt the investigator"

[false rejections (plural) prompt the investigator]

The mistake in grammar was corrected as suggested, see lines 975-976.

L824

"but leave other landscapes out"->

"but leaving other landscapes out"

The mistake in grammar was corrected, see line 1175.

L889

"the current sampling situation for the Paleolithic humans"->

"the current sampling situation for Paleolithic humans"

The mistake in grammar was corrected, see line 1275.

L906-907

"Thus, there exists a limit to model complexity that makes sense to explore with qpAdm on SSL"

[While true, this sentence contains no information about qpAdm, as there exists a limit to model complexity that makes sense to explore with any tool and finite data. And data is always finite. What would be informative is an indication of where the complexity limit lies for people wish to use qpAdm].

We have clarified the sentence in the following way (lines 1292-1294):

"Thus, a limit to model complexity that makes sense to explore with high-throughput qpAdm on SSL is rather low in most of our simulations: just two or three sources (see more on this topic below in the paragraphs focused on FDR)."

In *section 5b* (lines 1682-1735, **Fig. 7**) and elsewhere in the updated manuscript we discuss that the limits to model complexity that makes sense to explore with high-throughput qpAdm protocols are due to growing pre-study odds and less precise EAF estimates for complex models (the latter leading to increased FNR, see lines 1614-1617, 2164, 2642-2644).

L987

"that part of the space is populated by most symmetric models" ->

(A) "that part of the space is populated by the most symmetric models"

OR

(B) "that part of the space is populated by most of the symmetric models"

[Inclusion of article "the" actually makes a difference in meaning here! (so rare!)]

In the current version some symmetric models populate other parts of the space.

In my suggested version (A) those models which are most symmetric, populate that part of the space, but my understanding is symmetry of the models is binary yes/no. 'most' then would not be appropriate to describe symmetry. So I favour (B)]

We disagree here: some admixture models in the SSL context are more symmetric than others (compare sources at a very small angular distance and those almost opposite to each other), and we try to capture that gradient with our “minimal source-target-source angle” metric (see **Box 1, Figs. 4 and 6**). That’s why we prefer the suggested version (A); see lines 532-534 in the supplement where this text was moved.

L995

"tests also became popular" -> "tests have also become popular" (I believe these tests still are popular).

The mistake in grammar was corrected, see line 540 in the supplement where this text was moved.

L1003

". And frequency..." -> ". Frequency"

Corrected as suggested, see line 548 in the supplement where this text was moved (no changes are tracked in SI).

L1055

"are populated by models highly unevenly" ->

"are highly unevenly populated by models"

The mistake in grammar was corrected, see line 1377.

L1100

"protocols on the real-world archaeogenetic sampling"->

"protocols on real-world archaeogenetic sampling"

The mistake in grammar was corrected, see line 1521.

L1148-1149

"following this approach, it is hardly possible to find a single best model complexity level for a target"

[Again "hardly possible" is ambiguous in English See L116 comment. See also "hardly achievable" on L1149 - difficult to achieve?]

The sentence was clarified by choosing the phrase "almost impossible" in the former case (see line 1624) and the phrase "difficult to achieve" in the latter case (see line 1626):

"We note that, following this approach, it is almost impossible to find a single best model complexity level for a target and a single best model at that complexity level, but that is difficult to achieve in principle on SSL as we discuss in the next section."

L1233

"the relaxed conditions on EAF do not allow to reject efficiently the highly asymmetric models" ->

"the relaxed conditions on EAF do not allow the highly asymmetric models to be efficiently rejected"

[cf comment L36, see also L1237, L1251, L1410, L1536, 1681]

All these mistakes in grammar were corrected: see lines 431-432, 1200-1201, 1802-1803, 2861-2862,

L1587-1590

"To be precise, on the "10⁻⁵ to 10⁻²" SSL the pre-qpAdm and post-qpAdm distributions of models in the optimality metric space are uncorrelated for the simplest, two-way, models (Figs. 9, 14, Table S11), but even this is not true in the case of the real data (Fig. 14)."

["but even this is untrue in the case of the real data"... even non-correlation is untrue for real data? ...rewording necessary]

We have rephrased in the following way (lines 2527-2531): "To be precise, on the "10⁻⁵ to 10⁻²" SSL the pre-qpAdm and post-qpAdm distributions of models in the optimality metric space are uncorrelated just for the simplest, two-way, models (**Figs. 6, 11, Table S11**), but in the case of the real data these distributions are correlated even for two-way models (**Fig. 11**)."

1611

"of chosen complexity classes only (one- and two-way)" ->

"of chosen complexity classes (only one- and two-way)"

The mistake in grammar was corrected, see line 3067.

1618-1620

"This problem was deliberately left out since in the literature more attention is paid to interpretation of "fitting" than of rejected qpAdm models."

[as this is in the Discussion, the authors should feel free to express an opinion on whether the literature's attention is well directed on the issue]

To make our reasoning more transparent, we have reorganized the whole section on limitations of our AGS-focused analyses, see lines 3061-3099 (now this section opens the Methods chapter). The sentence in question was removed since we skipped classification of universally rejected models into true and false ones also for another more important reason, and we highlight it now (lines 3064-3075):

"We ... have estimated FDR instead of FPR due to an important technical limitation: the process of model classification into true and false ones is difficult to automate fully since it requires careful interpretation of the simulated topology and simulated admixture proportions (this is illustrated

by the case studies in **Fig. S6**; note that we attempted to estimate FPR and FNR for *qpAdm* results on AGS histories indirectly, see the end of *section 3*). Due to this limitation, we did not attempt to classify universal rejections of two-way models by *qpAdm* or other methods into false rejections due to violations of the topological assumptions of *qpAdm* (**Fig. 1**) and true rejections when the true admixture history of the target does not fit a two-way model."

L1656-1657

*"compared to the real population history of humans and other species as we understand it" ->
"compared to the real population histories of humans and other species as we understand them"*

The mistake in grammar was corrected, see lines 3140-3141.

L1712-1716

"Another way of radically improving FDR of qpAdm protocols applied to AGS histories is combining qpAdm with an unsupervised ADMIXTURE analysis (Suppl. text 4), however we have not tested ADMIXTURE on low-quality data, and supervised ADMIXTURE protocols were not tested either. Unsupervised ADMIXTURE has known problems even on high-quality data and AGS histories (Lawson et al. 2018). That is why this recommendation is still tentative."

[It seems strange to describe a tentative recommendation as a "way of radically improving FDR"]

To soften the statement, we have replaced "Another way of radically improving FDR" with "A potential way of improving FDR", see lines 2611-2614.

L1764-1768

"EAF lie largely within (0, 1) and correlate with simulated gene-flow intensities only for sources located symmetrically around the target, but for asymmetric sources they vary widely both outside and, complicating the matters even further, inside (0, 1) Although this property of AF estimation is trivial from the mathematical perspective, it is not recognized in practice"

[cf comment L522-252: This needs explained. Further, returning 'proportions' outside the range (0,1) is NOT a general property of AF estimation, as there are many ways to estimate ancestry proportion that actually return estimates on (0,1) for two sources or the n-simplex for n sources (eg STRUCTURE and all its relatives and descendants). I suggest there is good reason for estimates outside(0,1) to be "not recognised in practice". I suggest it is quite possible for something to be both mathematically trivial and WRONG. Cf also Methods L1997-1998].

Please see the discussion of this issue above (the comments for L222-226 and L522). We have emphasized that this property of the default *qpAdm* algorithm is unusual and cited Harney et al. 2021 several times as a justification by the *qpAdm* developers (in Introduction on lines 355-358, Results on lines 811-813, and Methods on lines 3246-3247).

1780-1782

"efficient discrimination of the worst models from ideal ones is possible with a p-value threshold and the distal non-rotating (but not rotating!) protocol"

[Unclear text]

We have removed this sentence from Discussion in the process of streamlining it, and we have rephrased relevant sentences in Results to clarify our statements (see section 6b, lines 2153-2160):

"The systematic distal non-rotating protocol with the second feasibility criterion (the p-value threshold) demonstrates attractive properties in the case of three-, four-, and six-way models: there is strong correlation between the fraction of models passing the p-value threshold and model optimality, and the percentage of truly ideal models rejected equals the p-value threshold, as expected (Fig. 9b). In contrast, such a "clean" result was not achieved with the distal rotating protocol: the fraction of models passing the p-value threshold grows with increasing model complexity, but at the same time its correlation with model optimality disappears (Fig. 9b)."

L1804-1811

"We showed that in this situation robust rejection of highly asymmetric complex models with distant sources (with average ST distance greater or approximately equal to the landscape radius) is possible only for very specific combinations of landscape sparsity and model feasibility criteria, but models of that sort are by far the most common in random sets of demes drawn from both simulated landscapes (Figs. 9 and S20b, Table S11) and real archaeogenetic data (Fig. 14). Furthermore, this optimal combination of sparsity and feasibility criteria we found (EAF {plus minus} 2 SE \in (0, 1) and the "10⁻³ to 10⁻²" landscapes) is very uncommon in practice"

[A) robust rejection is possible only for very specific combinations

B) but models of that sort are by far the most common in sim and real data

C) Furthermore this optimal combination is very uncommon in practice

This text appears self-contradictory. Re-word.]

We agree that the text does not convey our message well: not admixture models as combinations of sources on the landscape, but combinations of landscape properties and model feasibility criteria are meant in statements A and C. We think that our emphasis on pre-study odds and FDR in the new manuscript version helps to improve the argument, see lines 2764-2773:

"Highly asymmetric models with distant sources are by far the most common in random sets of demes drawn from simulated landscapes (Figs. 6, S23) and real archaeogenetic data (Fig. 11): that is, pre-study odds are low (Fig. 7). Efficient rejection of these non-optimal models (with average ST distance greater or approximately equal to the landscape radius) is possible only for very specific combinations of landscape sparsity (influencing mostly power = 1 – FNR) and model feasibility criteria (influencing both FPR and FNR). Furthermore, this optimal combination of sparsity ("10⁻³ to 10⁻²" landscapes) and feasibility criteria we found is very uncommon in practice:

restricting both “right” and “left” deme sets to a sub-continental region, using a low p -value threshold such as ≥ 0.001 , and requiring that $EAF \pm 2 SE \in (0, 1)$ are very rare approaches, to our knowledge.”

1869-1870

"and at even higher levels of sparsity the method stops working completely (at least for our simulations where all demes arise via a multifurcation)."

[Is the suggestion here that all demes arising from a multifurcation is not realistic? - it does seem a strange way to start a lattice simulation - perhaps give the reader more info].

We did not mean in this sentence that the situation where all demes arise from a multifurcation is not realistic (although unrealistic it is!). We have added a clarification on lines 3438-3447 in the Methods:

*"There are so many ways of “unfolding” an initial lattice state via a serial-founder process represented as a bifurcating tree (Estavoyer and François 2022) that we decided to avoid this additional level of complexity in our study and initialized the lattice simulations with a multifurcation, but deep in the past, at 2,220 generations before the oldest sampling time point. In this way we focus on genetic structure generated by gene-flow networks on SSL instead of structure generated by phylogenetic processes. If gene-flow intensities on SSL are relatively high, traces of the initial multifurcation are lost by the beginning of the sampling epoch, but this is not true for simulations with the lowest per-generation gene-flow intensities (the “ 10^{-5} to 10^{-4} ” landscapes, see below), as demonstrated by our analysis of one-way $qpAdm$ models interpreted as cladality tests (**Suppl. text 5** and **Fig. S21**)."*

L1997-1998

"Inferred admixture proportions in our analysis were not constrained between 0 and 1."

[The reader needs to know why and how. See comments L522-252, L1764-1768]

Please see the discussion of this issue above (the comments for L222-226 and L522). We have emphasized that this property of the default $qpAdm$ algorithm is unusual and cited Harney et al. 2021 several times as a justification by the $qpAdm$ developers (in Introduction on lines 355-358, Results on lines 811-813, and Methods on lines 3246-3247).

L2122 *"linked sites were pruned with PLINK v.2.00a3LM (Chang et al. 2015)"*

[Sites on the same chromosome (linked) were pruned? That would be strange. Perhaps what is meant here is: sites with allelic states in statistical association were pruned. This would make more textual sense as the pruning has an r^2 argument (and sometimes in the literature statistical association (linkage disequilibrium) is confounded with being linked)... But would pruning sites in LD make any sense in terms of inference? What is the justification? Admixture generates LD genome wide (Baird 2015) and is therefore a potentially important part of the signal in the

simmed data... why prune this signal out? If a data reduction step is necessary before PCA, a random thinning would seem more appropriate]

L2131 "Prior to the analysis, linked sites were pruned with PLINK v.2.00a3LM (Chang et al. 2015)"

[See comment L2122... Admixture generates LD genome wide and is therefore a potentially important part of the signal in the simmed data... why prune this signal out? If a data reduction step is necessary before ADMIXTURE analysis, a random thinning would seem more appropriate]

We indeed pruned sites with allelic states in statistical association, following recommendations from key publications applying PCA to genomic SNP data (see, for example, Privé F, et al. 2020. Efficient toolkit implementing best practices for principal component analysis of population genetic data. *Bioinformatics*. 36(16):4449-4457). In the case of LD pruning prior to ADMIXTURE analyses, that practice is recommended by the software developers themselves: "Our model makes the further assumption of linkage equilibrium among the markers. Dense marker sets should be pruned to mitigate background linkage disequilibrium (LD). This can be done informally, by thinning the marker set according to a minimum separation criterion or by pruning markers observed to be in linkage disequilibrium on the basis of common LD summary statistics such as D' or r^2 . Neither pruning approach is a perfect remedy for linkage disequilibrium." (Alexander et al. 2009. Fast model-based estimation of ancestry in unrelated individuals. *Genome Res*. 19(9):1655-1664).

We have clarified this in Methods on lines 3360-3365 (PCA):

"Prior to the analysis, sites with allelic states in statistical association were pruned with PLINK v.2.00a3LM (Chang et al. 2015) using the following settings: window size, 2000 SNPs; window step, 100 SNPs; r^2 threshold = 0.5 (argument "--indep-pairwise 2000 100 0.5"). This kind of linkage disequilibrium (LD) pruning is common in the population genetic literature since complex LD structure may confound PCA, especially higher PCs (Privé et al. 2020)."

and on lines 3382-3386 (ADMIXTURE):

"Prior to the analysis, sites with allelic states in statistical association were pruned with PLINK v.2.00a3LM (Chang et al. 2015) using the following settings: window size, 2000 SNPs; window step, 100 SNPs; r^2 threshold = 0.5 (argument "--indep-pairwise 2000 100 0.5"). This kind of LD pruning relying on r^2 is recommended since the underlying model assumes sites in linkage equilibrium (Alexander et al. 2009)."

We think that criticizing robustness of this LD pruning approach is beyond the scope of our study since only PCA and ADMIXTURE results on the AGS simulations are affected by LD pruning, but they are discussed overwhelmingly in the supplement (see **Fig. S6** and **Suppl. text 4**). No *qpAdm* results on AGS or SSL rely on LD-pruned data in our study.

March 11, 2025

RE: GENETICS-2025-307840R1

Dr. Pavel Flegontov
University of Ostrava; Harvard University
Department of Biology and Ecology
Chittussiho 1077/10
Ostrava, N/A 71000
Czech Republic

Dear Dr. Flegontov:

Congratulations! We are delighted to inform you that your manuscript titled "Performance of qpAdm-based screens for genetic admixture on admixture-graph-shaped histories and stepping-stone landscapes" is acceptable for publication in GENETICS. Many thanks for submitting your research to the journal.

The data and scripts are now made available, and seem very well documented. I note that the figshare link is at present "shared privately", and will need to be updated on publication.

To Proceed to Production:

1. Format your article according to GENETICS style, as discussed at <https://academic.oup.com/genetics/pages/general-instructions>, and upload your final files at <https://genetics.msubmit.net>.
2. Your manuscript will be published as-is (unedited-as submitted, reviewed, and accepted) at the GENETICS website as an Advanced Access article and deposited into PubMed shortly after receipt of source files and the completed license to publish. Please notify sourcefiles@thegsajournals.org if you do not wish to publish your article via Advanced Access.
3. We invite you to submit an original color figure related to your paper for consideration as cover art. Please email your submission to the editorial office or upload it with your final files. You can submit a small-sized image for evaluation, and if selected, the final image must be a TIFF file 2513px wide by 3263px high (8.375 by 10.875 inches; resolution of 600ppi). Please avoid graphs and small type.

If you have any questions or encounter any problems while uploading your accepted manuscript files, please email the editorial office at sourcefiles@thegsajournals.org.

Sincerely,

Nick Barton
Senior Editor
GENETICS

Approved by:
Howard Lipshitz
Editor in Chief
GENETICS

note: Please add jnls.author.support@oup.com and genetics.oup@kwglobal.com (or the domains @oup.com and @kwglobal.com) to your email program's "safe senders" list. You will be contacted by both at various points during the production process.